# Protein mimetic 2D FAST rescues alpha synuclein aggregation mediated early and post disease Parkinson's phenotypes

Nicholas H. Stillman ®[1,2], Johnson A. Joseph[1,2], Jemil Ahmed[2,3], Charles Zuwu Baysah[1,2], Ryan A. Dohoney ®[1,2], Tyler D. Ball ®[1,2], Alexandra G. Thomas[1,2], Tessa C. Fitch[2], Courtney M. Donnelly[1,2] & Sunil Kumar ®[1,2,3] ✉

Abberent protein-protein interactions potentiate many diseases and one example is the toxic, self-assembly of α-Synuclein in the dopaminergic neurons of patients with Parkinson's disease; therefore, a potential therapeutic strategy is the small molecule modulation of α-Synuclein aggregation. In this work, we develop an Oligopyridylamide based 2-dimensional Fragment-Assisted Structure-based Technique to identify antagonists of α-Synuclein aggregation. The technique utilizes a fragment-based screening of an extensive array of non-proteinogenic side chains in Oligopyridylamides, leading to the identification of NS132 as an antagonist of the multiple facets of α-Synuclein aggregation. We further identify a more cell permeable analog (NS163) without sacrificing activity. Oligopyridylamides rescue α-Synuclein aggregation mediated Parkinson's disease phenotypes in dopaminergic neurons in early and post disease Caenorhabditis elegans models. We forsee tremendous potential in our technique to identify lead therapeutics for Parkinson's disease and other diseases as it is expandable to other oligoamide scaffolds and a larger array of side chains.

Abberent protein-protein interactions (aPPIs) are associated with a plethora of pathological conditions, including infectious diseases, cancer, neurodegenerative diseases, and amyloid diseases[1-11]. Consequently, modulation of aPPIs is considered to be a promising therapeutic intervention toward various pathologies. The pathological aPPIs are mediated via specific chemical interactions that often sample dynamic and transient conformations, which spread over a large and hydrophobic surface[1-7]. One such example is the aggregation of α-Synuclein (αS), which is a neuronal protein expressed at high levels in dopaminergic (DA) neurons in the brain and implicated in the regulation of synaptic vesicle trafficking and recycling, and neurotransmitter release[12-18]. The aggregation of αS is associated with impaired DA neurons, which is a pathological hallmark of Parkinson's disease (PD)[12-18]. Therefore, one of the potential disease-modifying therapeutic strategies for PD is the modulation of αS aggregation[19-29]. A few small molecules have been shown to inhibit αS aggregation[19-29] (ref. within 19); however, some of them have complex chemical structures, which might limit their ability for synthetic tuning and further optimization of the antagonist activity against αS aggregation. Also, protein mimetics have been identified to inhibit αS aggregation; however, the non-proteinogenic side chains on them was limited and there was no systematic optimization carried out against αS aggregation[23-29]. Moreover, most of these ligands were not tested against PD phenotypes in DA neurons in in vivo models to further assess their therapeutic potential[23-25,28,29]. Therefore, ligands with the ability to manipulate aggregation with a large array of functional

[1]Department of Chemistry and Biochemistry, F.W. Olin Hall, 2190 E Iliff Ave, University of Denver, Denver, CO 80210, USA. [2]The Knoebel Institute for Healthy Aging, 2155 E. Wesley Ave, Suite 579, University of Denver, Denver, CO 80208, USA. [3]Molecular and Cellular Biophysics Program, Boettcher West, Room 228, 2050 E. Iliff Ave, University of Denver, Denver, CO 80210, USA. ✉e-mail: sunil.kumar97@du.edu

groups and having the tendency for systematic optimization of the antagonist activity could lead to potent antagonism of αS aggregation.

Oligopyridylmides (OPs) are a class of synthetic protein mimetics that have been shown to manipulate the aggregation of multiple proteins, including islet amyloid polypeptide[30–32], Aβ peptide[33,34], and mutant p53[35], which are associated with type 2 diabetes (T2D), Alzheimer's disease (AD), and cancer, respectively. OPs have a large surface area and synthetically tunable side-chain functionalities that can complement the topography and side-chain residues of proteins, such as those present at the interfaces of aPPIs during protein aggregation[30–38]. The OP is an ideal scaffold for the fragment-based approach because the antagonist activity of OPs against their biological targets has been shown to increase with increasing side chains (monopyridyl<dipyridyl<tripyridyl)[32–35]. However, the OP (tripyridyl) library used in the screening to identify antagonists of protein aggregation was moderate in size ( ~ 30 OPs) with limited chemical diversity ( ~ 11 side chains), which may have precluded the opportunity for the optimization of the antagonist activity of OPs against the aggregation of various amyloid proteins[30–37]. There were several limitations with the previous method, due to which we were not able to generate OP libraries (tripyridyls) with a larger array of non-proteinogenic side chains to identify antagonists for the aggregation of proteins[30–37]. In the old method, we used only a small library of presynthesized tripyridyls with a very limited number of side chains (Fig. 1A, Table 1, old method). Additionally, the synthesis was very tedious, including several synthetic steps (Fig. 1A, 14 steps, Table 1, old method), several column chromatography steps (Fig. 1A, 11 steps, Table 1, old method), and very low total % yield to synthesize one tripyridyl (Table 1, old method). There was no systematic optimization of the antagonist activity of OPs against αS aggregation. Also, in the old approach, we never reported the optimization of cell permeability or any other pharmaceutical properties of the most potent antagonist OP. Here, we have developed a 2-dimensional Fragment-Assisted Structure-based Technique (2D-FAST) by combining fragment and structure-based techniques into the OP scaffold in order to systematically optimize the antagonist activity against αS aggregation (Fig. 1B, C). The fragment-based approach has emerged as a promising method for drug discovery to identify high-affinity ligands against various pathological targets, including aPPIs[39–46]. In 2D-FAST, the 2D consists of the side chains and the number of pyridyls groups in OPs (Fig. 1C). There are several innovative features of our 2D-FAST for OPs, including (1) Use of common precursors for the elongation of OP from mono- to di- to tripyridyl synthesis; (2) A significant improvement in the synthetic procedure of OPs, including a smaller number of synthetic steps (8 steps), much higher % yield ( >20 fold), and very few chromatography steps (Table 1, new method); (3) Introduction of a large chemically diverse library of side chains (21 side chains, two times more than the old method) on OPs that mimic to a higher number of amino acid side chains of proteins, which will aid in enhancing the affinity and specificity of OPs toward protein target; (4) Use of a fragment-based approach for systematic optimization of the antagonist activity of OPs against αS aggregation, (5) Enhancement of the cell permeability of the most potent ligand without sacrificing its antagonist activity. Collectivley, we have applied a fragment based technique to a protein mimetic scaffold.

Using the 2D-FAST and an array of biophysical and cellular assays, we have identified NS132 as the most potent antagonist of de novo and fibers-catalyzed aggregation of αS. NS132 was able to wholly inhibit αS aggregation, even at a substoichiometric ratio (αS:NS132, 1:0.2). In contrast, the peptidomimetic approaches without the innovative features of our 2D-FAST, have identified ligands that require 5-100 fold molar excess to inhibit the aggregation of αS[28,29]. This observation highlights the innovative aspects of our 2D-FAST approach, which entails a systematic fragment-based screening of a very large array of non-proteinogenic side chains against αS aggregation that allows the

identification of a very potent antagonist. A structure-activity relationship (SAR) study demonstrated that the side chains of NS132 are essential for its antagonist activity. The HEK293 cell-based assays demonstrated that NS132 potently rescues cytotoxicity and inhibits the formation of intracellular inclusions. The 2D HSQC NMR study demonstrates that NS132 interacts with specific sequences of αS, which have been previously suggested to be the key aggregation-prone sequences[24,47]. The study also led to the synthesis of an analog of NS132 (NS163, Supplementary Fig. 2) with improved cell permeability without sacrificing the antagonist activity against αS aggregation. The antagonist activity of NS163 and NS132 was tested against αS aggregation-mediated PD phenotypes in two *C. elegans* PD models. Both ligands (NS163 and NS132) were very effective in rescuing various PD phenotypes in two *C. elegans* PD models, including neuroprotective effect against degeneration of DA neurons, motility recovery, improved food-sensing behavioral deficits, and reduced reactive oxygen species (ROS) level. Moreover, the OPs were very effective in rescuing further progression of PD phenotypes in DA neurons when administered in a post-disease-onset PD model, a model that mimics the current therapeutic intervention strategies, where the treatment begins during post-diagnosis of PD.

In this work, we develop a 2D-FAST and demonstrate its utility in the identification of potent antagonists of αS aggregation, a process that is associated with PD. We use a comprehensive study to establish the synthetic protein mimetic-based 2D-FAST approach and identify potent ligands, which are very effective in rescuing αS aggregation mediated PD phenotypes in physiologically relevant PD *C. elegans* models.

## Results
### Design and synthesis of the 2D-FAST for OPs
We used a 2D FAST approach in OPs to identify potent antagonists of the aggregation of αS. In this approach, we start with a library of monopyridyls with different functional groups. The library of monopyridyls was synthesized using 2-chloro-6-methyl-3-nitropyridine as a common precursor, which was treated with primary alcohols with diverse side chain functionalities via a one-pot reaction (Fig. 1B, a and see Supplementary information for synthetic details). The synthesis of the monopyrodyls did not require any column chromatography as the pure monopyridyls were extracted via acid/base treatment or via lyophilization (see Supplementary information for synthetic details). We selected the side chains containing hydrophobic, polar, positively charged, and negatively charged functional groups, which mimic the side chains of the amino acids (Fig. 1C). We could not use a small selection of the monopyridyls for assays because of their poor solubility in solution conditions due to the side chains (in 1 × PBS buffer, pH 6.5). The monopyridyl library was screened against the aggregation of 100 μM αS at an equimolar ratio using Thioflavin T (ThT) dye-based aggregation assay (Fig. 1B, b). The most potent monopyridyl antagonist (OP1) of αS aggregation was used as the precursor for the synthesis of a chloro-dipyridyl using a recently developed chromatography-free amide coupling in our lab (Fig. 1B, OP2). The chloro-dipyridyl was reacted with a library of primary amines/thiols to synthesize a library of dipyridyls (Fig. 1B, e) using a one-pot reaction. All reactions went to completion and a large number of dipyridyl products did not require column chromatography as the excess primary amines/thiols were evaporated on rotovap or lyophilizer. However, a few dipyridyls required column chromatography to separate them from the starting material side chains because of their very high boiling point (7 out of 21 dipyridyls required column, Supplementary Fig. 1). The dipyridyls were screened against the aggregation of 100 μM αS at an equimolar ratio using ThT assay (Fig. 1B, f), which identified the most potent dipyridyl antagonist (OP3) of αS aggregation. Finally, we synthesized a library of tripyridyls, screened, and identified the most potent tripyridyl against the aggregation of 100 μM αS at an equimolar ratio (Fig. 1B, g–j).

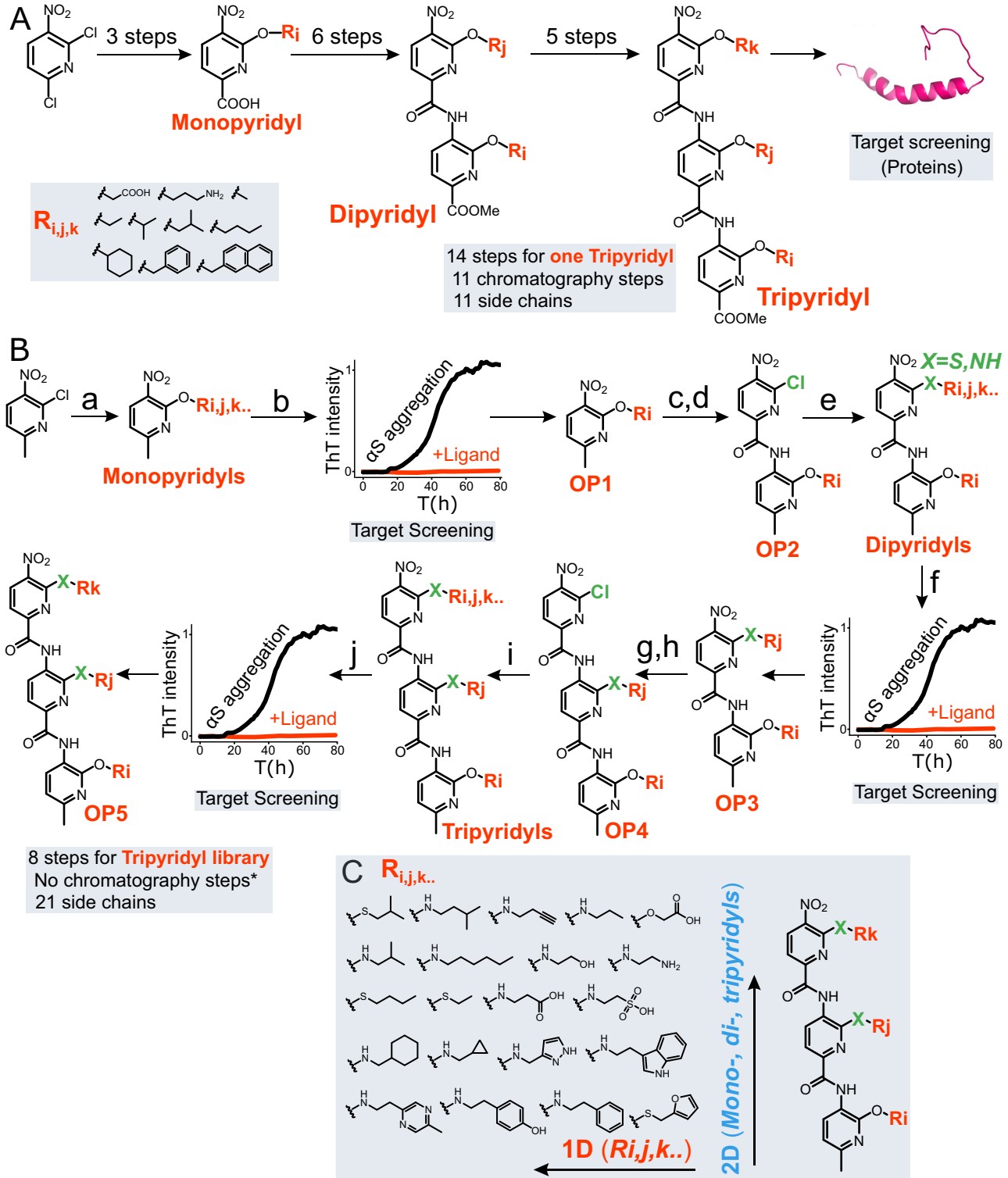

**Fig. 1 | A schematic for the comparison of the old method and the 2D-FAST.**
**A** Synthesis of the monopyridyls with various side chains ($R_{i,j,k}$) and the number of synthetic steps following the old method. (Inset) The chemical structures of the side chains on OPs. A flowchart for the synthesis of dipyridyls and tripyridyls and their testing against various biological targets. **B** The 2D-FAST schematic and conditions: **a** ROH, NaH or Na metal, toluene, 50 min. at 0 °C, then 5 h at r.t. **b** Screening and identication of the potent antagonist monopyridyls against αS aggregation using ThT aggregation assay. **c**, **g** Pd/C, H₂ (g), EtOAc, 3 h at r.t.

**d** 6-chloro-5-nitropicolinic acid, DCM (anhydrous), triethylamine, thionyl chloride, 0 °C to r.t., 45 min. **e**, **i** Primary amine/thiol, DIPEA, DCM, 3 h at r.t. **f** Screening and identication of the potent antagonist dipyridyls against αS aggregation using ThT aggregation assay. **h** 6-chloro-5-nitropicolinoyl chloride, dichloroethane (DCE), saturated sodium bicarbonate (NaHCO₃), 10 min at 0 °C. **j** Screening and identication of the potent antagonist tripyridyls against αS aggregation using ThT aggregation assay. **C** The representation of two dimensions in the 2D-FAST.

**Table 1 | Comparison of the properties to synthesize a tripyridyl with three distinct side chains using the old method and our new method (2D-FAST)**

| Properties | Old method | 2D FAST |
|---|---|---|
| *Total % Yield | 0.78%-3.61% | 17.57%-78.15% |
| Total fold yield | NA | 20 fold higher |
| Chromatography steps | 11 | 0# |
| Synthetic steps | 14 | 8 |
| Side chains mimicking natural amino acid side chains | 11 | 21 (~ 2 fold higher) |
| Total tripyridyls | One tripyridyl | Tripyridyl Library |

*The overall % yield was calculated by multiplying the yield of each synthetic step. # > 90% of side chains did not require column chromatography.

## Biophysical characterization of OPs against the aggregation of αS

The screening of the monopyridyls against 100 μM αS aggregation (1× PBS buffer, pH 6.5) identified NS41 as the most potent antagonist as it reduced the ThT signal to 29% and 17% at molar stoichiometries of 1:1 and 2:1 (NS41/αS), respectively (Fig. 2a–c). The inhibition of αS aggregation by NS41 was also confirmed by TEM images, which show an abundance (Supplementary Fig. 3a) and low amount (Supplementary Fig. 3b) of αS fibers in the absence and presence of NS41, respectively. We used SDS-PAGE (Sodium dodecyl sulfate polyacrylamide gel electrophoresis) as a complementary assay to validate ThT results and to further characterize the antagonist activity of the monopyridyls against αS aggregation. We used a total of four monopyridyls with varying antagonist activity against αS aggregation for the SDS-PAGE assay. The solutions of αS (±monopyridyls) were centrifuged and the soluble and insoluble fractions were subjected to SDS-PAGE analysis as reported earlier (Fig. 2d, Supplementary Fig. 4a, see details in methods section). The band intensities of the gels in SDS-PAGE assay were quantified using ImageJ software. In the absence of ligands, ~12% of αS was found in the soluble fraction and the rest was found in the insoluble fraction (Fig. 2d, Supplementary Fig. 4a, b). In the presence of NS41 at molar ratios of 1:1 and 2:1 (NS41:αS), 48% and 73% of αS protein were found in the soluble fraction, respectively, and the rest of αS protein was found in the insoluble fraction (Fig. 2d, Supplementary Fig. 4a, b). The most potent ligand from ThT assay (NS41) also demonstrated the highest amount of αS in the soluble fraction. Similarly, the least effective monopyridyl ligand (RD247) had the highest ThT intensity and demonstrated the highest amount of αS (87%) in the insoluble fraction, a value very close to the untreated αS protein (87% insoluble, Fig. 2d, Supplementary Fig. 4a, b). The antagonist activity of the monopyridyls corroborated well from both ThT and the SDS-PAGE assays (Fig. 2a–d, Supplementary Fig. 4a, b). The higher the antagonist activity of monopyridyls against αS aggregation, the lower the ThT intensity and the higher the amount of αS remained in the soluble fraction (Fig. 2a–d, Supplementary Fig. 4a, b). Together, NS41 was identified as the most potent monopyridyl antagonist of αS aggregation. It is important to note that NS41 contains a carboxyl (COOH) functional group side chain. We anticipated that the monopyridyl with the COOH functional group would be the most potent antagonist, because we have recently shown that a foldamer was a potent antagonist of αS aggregation and its antagonist activity predominantly relied on a negatively charged COOH side chain[24]. We have also shown in that work that the foldamer binds to the N-terminus of αS because of its negatively charged COOH functional group interaction with the positively charged lysine residues of αS[24]. Also, it has been suggested that the N-terminus of αS is important in facilitating its aggregation. Therefore, ligands that interact with the N-terminus of αS will likely inhibit the aggregation of αS. Additionally, we have also shown that the OPs with the COOH functional group are very effective inhibitors of aggregation of other amyloid proteins that contain lysine amino acid via the formation of salt bridges[30–34].

Subsequently, we synthesized dipyridyl library by keeping COOH acid as a functional group on first position. The dipyridyl library was screened against the aggregation of 100 μM αS (in 1×PBS buffer, pH 6.5) at an equimolar ratio using ThT aggregation assay (Fig. 2e)[48]. The screening led to the identification of NS55 as the most potent antagonist, as it was able to attenuate the ThT signal of αS aggregation by 96% (Fig. 2e–h). The inhibition of αS aggregation by NS55 was also confirmed by TEM images, which show an abundance (Fig. 2i) and no (Fig. 2j) αS fibers in the absence and presence of NS55, respectively. Similar to monopyridyls, we also used the SDS-PAGE assay to further validate the results of the ThT assay to determine the antagonist activity of the dipyridyls. We used six dipyridyls with varying antagonist activity (from ThT assay) for the SDS-PAGE assay. In the absence of dipyridyls, ~17% of αS was found in the soluble fraction and ~83% of αS was found in the insoluble fraction (Supplementary Fig. 5a–e). For the dipyridyls, the amount of insoluble fraction of αS (from SDS-PAGE, Supplementary Fig. 5b–e) was in close agreement with the ThT intensity (from ThT assay, Supplementary Fig. 5a). We observed that the higher ThT intensity in the presence of dipyridyls correlated with a higher amount of αS in the insoluble fraction. For example, in the presence of NS55 (most potent antagonist), the amount of αS the insoluble fraction (26%) and the ThT intensity (4%) were the lowest among the dipyridyls (Fig. 2e–h, Supplementary Fig. 5a–e). On the contrary, in the presence of NS119 (a poor antagonist), the amount of αS in the insoluble fraction (83%) and the ThT intensity ( >100%) were the highest among the dipyridyls (Fig. 2e, Supplementary Fig. 5a–e). Clearly, we demonstrated that the antagonist activity of dipyridyls determined from ThT assay was in close agreement with the SDS-PAGE assay.

We compared the antagonist activity of dipyridyls using the ThT assay (and gel shift assay) to carry out the SAR study between dipyridyls and αS. We observed various patterns between the antagonist activity and the chemical structure of the side chains on the second position of dipyridyls (Fig. 2e). We first compared the hydrophobicity of the side chains of the dipyridyls. For the most part, the antagonist activity of the dipyridyls was directly related to the hydrophobicity of the side chains. The propyl side chain has the lowest antagonist activity, and the antagonist activity for the most part was increased with the increase in the hydrophobicity of the side chains. The cyclohexyl group had the highest antagonist activity (NS55, Fig. 2e–h). The antagonist activity decreased with a higher hydrophobicity than cyclohexyl group, as demonstrated by the indole group (NS72, Fig. 2e). We also observed that the side chains with amines (aliphatic or aromatic) were detrimental to the antagonist activity against αS aggregation. The polar non-aromatic hydroxyl groups as side chains were effective antagonists of αS aggregation; however, the phenol group as a side chain was a moderate antagonist of αS aggregation. The negatively charged (carboxylic and sulfonic acid) side chains were very effective inhibitors of αS aggregation. The effective antagonist activity of the hydrophobic, negatively charged, and primary hydroxyl groups is because these dipyridyls might be interacting with different regions of αS and inhibiting the aggregation. We will further explore these different regions of αS that are important for the aggregation and that study will be part of a separate manuscript. For the current study, we chose the cyclohexyl group as the second side chain to synthesize the tripyridyls because it demonstrated the highest antagonist activity against αS aggregation (Fig. 2e–h). We envision that the negatively charged (COOH group) and hydrophobic (cyclohexyl) side chains on the dimer interact with the positively charged (lysine) and hydrophobic groups on the N-terminus of αS. Surprisingly, most dipyridyls synthesized using primary thiols were agonists of αS aggregation; therefore, we did not pursue primary thiols for the synthesis of the tripyridyl library (Supplementary Fig. 6).

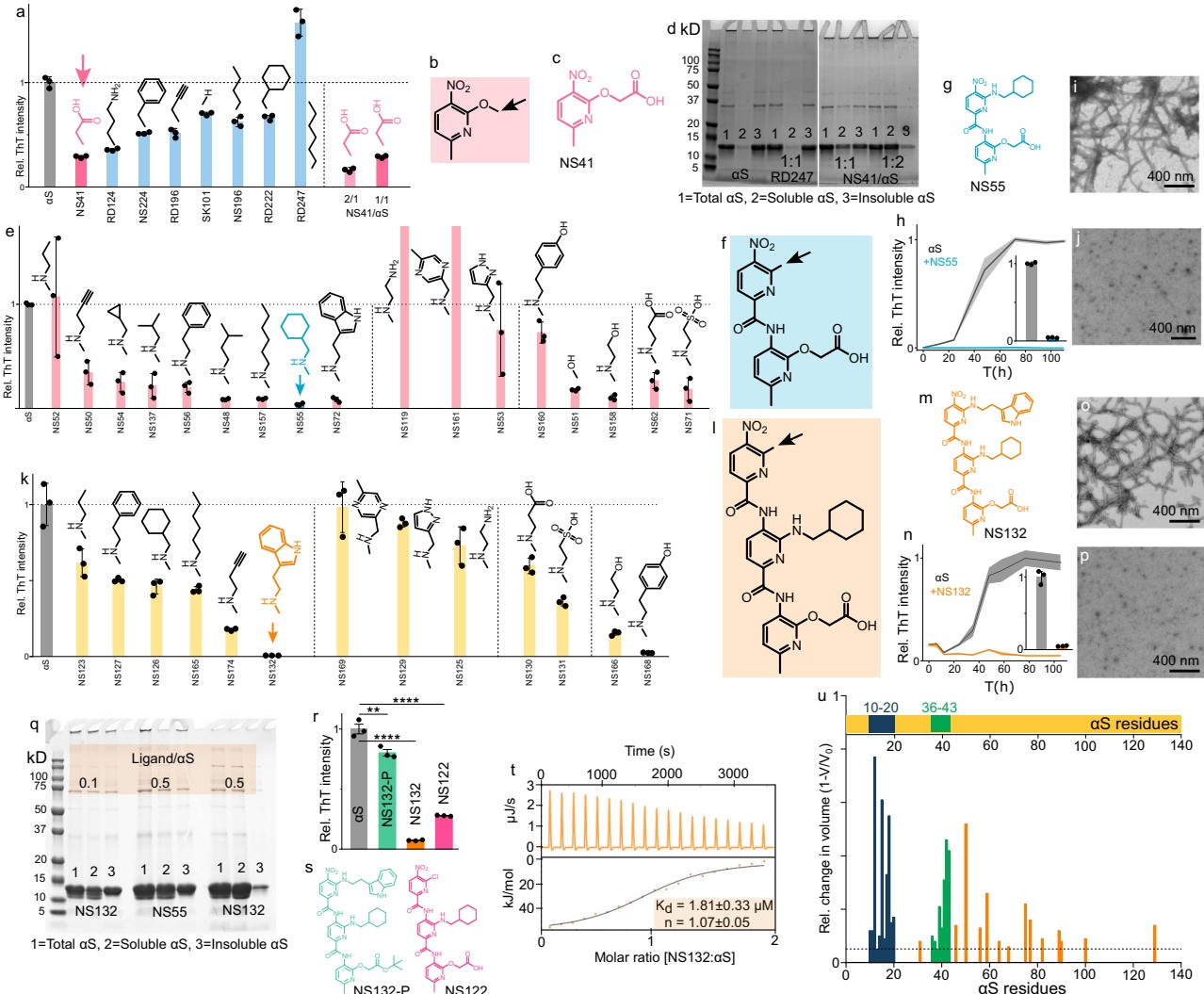

**Fig. 2 | Identification and characterization of the potent antagonist of αS aggregation using the 2D-FAST.** The graphical representation of the ThT intensity of 100 μM αS aggregation for four days in the absence and presence of mono-pyridyls (**a**), dipyridyls (**e**), and tripyridyls (**k**) at an equimolar ratio. The arrow indicates the most potent antagonist of αS aggregation. The generic chemical structures of monopyridyls (**b**), dipyridyls (**f**), and tripyridyls (**l**). The most potent monopyridyls (**c**), dipyridyls (**g**), and tripyridyls (**m**) antagonists of αS aggregation. SDS-PAGE gel shift assay analysis of 100 μM αS aggregation after four days in the absence and presence of monopyridyls (**d**) and dipyridyls/tripyridyls (**q**) at the indicated molar ratios. Gel image is representative of 3 individual experiments. The aggregation profile of 100 μM αS in the absence and presence of NS55 (**h**) and NS132 (**n**) at an equimolar ratio. TEM images of 100 μM αS aggregated for four days in the absence (**i**, **o**) and presence of equimolar NS55 (**j**) and NS132 (**p**). TEM images are representative of 3 individual experiments. **r** The ThT intensity of αS

aggregation after four days in the absence and presence of NS132 derivatives. **s** Chemical structures of NS132 derivatives. **t** The ITC thermogram for the titration of NS132 into αS where heat burst curves and the corrected injection heats upper and lower panel, respectively. **u** Graphical presentation of the relative volume changes (V = volume change, $V_0$ = total volume) of the backbone amide peaks of [15]N-labeled αS (70 μM) in the presence of equimolar NS132. The colored sequences (blue and green) are the potential binding sites of NS132 on αS. The dashed line represents the reported volume changes in [15]N-labeled αS residue peaks in the presence of NS132 above 5%. The ThT experiments were conducted three times and the reported change in the ThT intensity was an average of three separate experiments. The data were expressed as mean values ± SD. *P* values were determined by one-way ANOVA with Tukey's multiple comparisons test where relevant. *$p < 0.05$, **$p < 0.01$, ***$p < 0.001$, ****$p < 0.0001$. Source data are provided as a Source Data file.

Next, we used NS55 (dipyridyl) as a precursor to synthesize and generate a tripyridyl library because we have shown that tripyridyls are better antagonists than dipyridyls for various amyloid proteins[31,33-35,49]. For the synthesis of tripyridyls, we reduced the dipyridyl and used a chromatography-free amide coupling method to form the common precursor tripyridyl (OP4) (Fig. 1g, h). Subsequently, OP4 was treated with various primary amines to generate the library of tripyridyls with various functional groups (Fig. 1i, Fig. 2k, l). All reactions went to completion and most of the tripyridyls products did not require column chromatography (6 out of 15 tri-pyridyls required column, Supplementary information for synthetic details of tripyridyls).

The screening of tripyridyls against 100 μM αS aggregation (Fig. 1j) at an equimolar ratio using the ThT assay (in 1×PBS buffer, pH 6.5) led to the identification of NS132 as the most potent antagonist, which reduced the ThT fluorescence intensity from 100% (only αS) to 1% (Fig. 2k–n). The antagonist activity of NS132 was also confirmed with the TEM images, where we did not observe αS fibers in the presence of NS132 (Fig. 2o, p). Similar to monopyridyls and dipyridyls, we used the SDS-PAGE assay to further validate the antagonist activity of tripyridyls determined from the ThT assay. We used multiple tripyridyls with varying antagonist activity for the SDS-PAGE assay. Again, we observed that the higher the amount of the insoluble fraction of αS (gel shift assay) in the presence of tripyridyls, the higher their respective ThT

intensities. For tripyridyls, the amount of insoluble αS (from SDS-PAGE, Supplementary Fig. 7b–d) was in close agreement with the ThT intensity (from ThT assay, Fig. 2k, Supplementary Fig. 7a). For example, in the presence of NS132 (potent antagonist), the amount of αS in the insoluble fraction (14%) and the ThT intensity (1%) were the lowest among the tripyridyls (Fig. 2k, m, n, q and Supplementary Fig. 7b–d). On the contrary, in the presence of NS169 (poor antagonist), the amount of insoluble αS (88%) and the ThT intensity (98%) were the highest among the tripyridyls (Fig. 2k and Supplementary Fig. 7b–d). Clearly, we demonstrated that the antagonist activity of tripyridyls determined from ThT assay was in close agreement with the SDS-PAGE assay.

Both dipyridyl (NS55) and tripyridyl (NS132) were very effective inhibitors of αS aggregation at an equimolar ratio; however, NS132 was a far more effective antagonist than NS55 at a substoichiometric ratio of 1:0.5 (αS:ligand) and NS132 was almost equally effective at 1/5[th] of the concentration of NS55 in inhibiting the aggregation of αS (αS:ligand), reflected by SDS-PAGE and ThT assay (Fig. 2q, Supplementary Fig. 8a, b). In the case of NS132, αS was predominantly detected in the soluble fraction at a substoichiometric ratio (αS:ligand, 1:0.5, Fig. 2q, Supplementary Fig. 8a, b). In marked contrast, in the case of NS55, a significant amount of αS protein was found in the insoluble form (αS:ligand, 1:0.5, Fig. 2q). The soluble and insoluble amounts of αS were comparable when the concentration of NS132 was 5-fold less than NS55 (Fig. 2q). Collectively, both the ThT assay and SDS-PAGE analysis demonstrate that NS132 is a far better antagonist than NS55 of αS aggregation. These results highlight the validity of our 2D-FAST, where we were able to identify NS132 (tripyridyl) as a better antagonist than NS55 (dipyridyl) of αS aggregation.

We compared the antagonist activity of tripyridyls using the ThT assay (and gel shift assay) to carry out a SAR study between tripyridyls and αS. We observed various patterns between the antagonist activity and the chemical structure of the side chains on the third position of tripyridyls (Fig. 2k). We first compared the antagonist activity of the hydrophobicity of the side chains of the tripyridyls. The tripyridyl with propyl side chain (NS123) demonstrated the lowest antagonist activity as it decreased the ThT signal from 100% to 72.7% (Fig. 2k). The tripyridyls with more hydrophobic groups, including NS127, NS126, and NS165 demonstrated moderate antagonist activity against αS aggregation as they decreased the ThT signal to 50.3%, 45.9%, 43.9%, respectively. The tripyridyl with the indole group (NS132) was the most potent antagonist as it decreased the ThT signal to 1% (Fig. 2k). The antagonist activity, for the most part, was increased with an increase in the hydrophobicity of the side chains in the tripyridyls except the trimer with an alkyne side chain (NS174), which decreased the ThT signal to 17.9% (Fig. 2k). One of the reasons could be that NS174 might be interacting with a different αS region than NS132 and inhibiting αS aggregation. It has been shown recently by us and others that there are multiple αS sequences that facilitate αS aggregation[24,47]. We also observed that the tripyridyls with side chains as amines (aliphatic or aromatic) did not demonstrate any noticeable effect on αS aggregation, a pattern similar to the dipyridyls (Fig. 2e, k). The tripyridyls with carboxylic (NS130) and sulfonic (NS131) acids were moderate antagonists of αS aggregation as they decreased the ThT signal to 59.6% and 36.1%, respectively (Fig. 2k). The tripyridyls with aliphatic hydroxyl (NS166) and phenol (NS168) groups were potent antagonists of αS aggregation as they decreased the ThT signal to 15.5% and 2.7%, respectively (Fig. 2k). The tripyridyl, NS168 was a potent antagonist of αS aggregation as it decreased the ThT signal to 2.5%, close to NS132 (Fig. 2k). We speculate that NS168 might be interacting with a different sequence of αS than NS132. We will investigate the interaction of these tripyridyls with αS in detail and it will be presented in the near future.

To further confirm that the side chains of NS132 are essential for its antagonist activity, we used various analogs of NS132 and compared their antagonist activity for αS aggregation. The ThT signal of αS

aggregation was decreased from 100% to 80%, 30%, and 7% in the presence of NS132-P (Protected COOH group), NS122 (Chloro side chain), and NS132 at an equimolar ratio, respectively (Fig. 2r, s). The SDS-PAGE analysis also validated the ThT results, where the insoluble fraction of αS for NS132 and NS132-P were 11% and 84%, respectively at an equimolar ratio (ligand:αS), suggesting that NS132-P was a poor antagonist of αS aggregation (Supplementary Fig. 9a–d). Collectively, both ThT assay and SDS-PAGE analysis demonstrate that NS132 is a far better antagonist than NS132-P and NS122 and the side chains are important for the antagonist activity of NS132 against αS aggregation. Under matching conditions of the ThT aggregation assay, we did not observe any significant quenching of the ThT fluorescence signal by NS132 (Supplementary Fig. 10). We also characterized the binding interaction between αS and NS132 using the isothermal calorimetry titration (ITC) (Fig. 2t). The ITC titration yielded the dissociation constant ($K_d$) of $1.81 \pm 0.33 \,\mu M$ with a binding stoichiometry of 1:1 (αS:NS132) (Fig. 2t). We utilized two-dimensional heteronuclear single quantum coherence NMR spectroscopy (2D NMR HSQC) to gain insights into the binding site of NS132 on αS. We collected the HSQC NMR of $70 \,\mu M$ $^{15}$N-$^{1}$H-uniformly labeled αS in the absence (Supplementary Fig. 11, red) and presence of NS132 (Fig. 2u, Supplementary Fig. 11, blue) and compared the volumes of the amide peaks. In the presence of NS132, we observed noticeable volume changes in the amide peaks for specific residues toward the N-terminus, indicative of the interaction and binding site of NS132 on αS, especially continuous residue sequences from 10-20 and 36-43 (Fig. 2u, blue and green). NS132 also moderately or weakly interacted with a few distinct residues on the N-terminus of αS (up to 100 residues), represented by small changes in the peak volumes (Fig. 2u). These might be weak secondary binding sites of NS132 on αS. From the 2D HSQC NMR, we have identified two potential binding sites of NS132 on αS. NS132 contains a negatively charged side chain (COOH) and two hydrophobic side chains (cyclohexyl and indole). For the first binding site (from 10-20 residues of αS), we posit that the COOH and the two hydrophobic functional groups of NS132 likely interact with the lysine (K12) and the hydrophobic residues patch (14-16) of αS. Similarly, for the second binding site of NS132 on αS, we envision that the side chain functional groups of NS132 interact with the lysine (K43) and the hydrophobic residues patch (36-40) of αS. We have shown these interactions using 2D-NMR, where the overall volume intensities of these residues of αS were affected in the presence of NS132. The binding sites of NS132 have been suggested to be the essential sequences for αS aggregation (residue 10-20 and 36-43) and these sequences have been considered to be the potential therapeutic targets for the effective inhibition of αS aggregation and associated PD phenotypes[24,47]. Our study supports the hypothesis that the targeting of these sequences will effectively inhibit αS aggregation.

To confirm that NS132 did not generate fiber-competent cytotoxic structures during αS aggregation inhibition, we utilized a well-established model of HEK293 cells, which stably express YFP-labeled αS-A53T mutant (αS-$_{A53T}$-YFP, Fig. 3a, control)[23,24,50]. The endogenous monomeric αS-$_{A53T}$-YFP have been shown to template into fibers when transfected with exogenously added αS fibers in the presence of Lipofectamine 3000[23,24,50]. The aggregation of endogenous monomeric αS-$_{A53T}$-YFP into fibers can be detected by the intracellular fluorescent puncta (Fig. 3a, + αS). A solution of $100 \,\mu M$ αS was aggregated in the absence and presence of NS132 at an equimolar ratio for four days. The resulting solutions of αS fibers ($5 \,\mu M$ in monomeric αS, ±NS132) were introduced to HEK cells in the presence of Lipofectamine 3000 for 24 h. In contrast to the control (no fibers), we observed a significant number of fluorescent inclusions in the presence of αS fibers (Fig. 3a, white arrows, +αS), which was a consequence of the templating of endogenous monomeric αS-$_{A53T}$-YFP by the exogenously added αS fibers. The αS inclusions were colocalized in the cytoplasm of HEK cells as suggested by others as well[24,51–54]. In

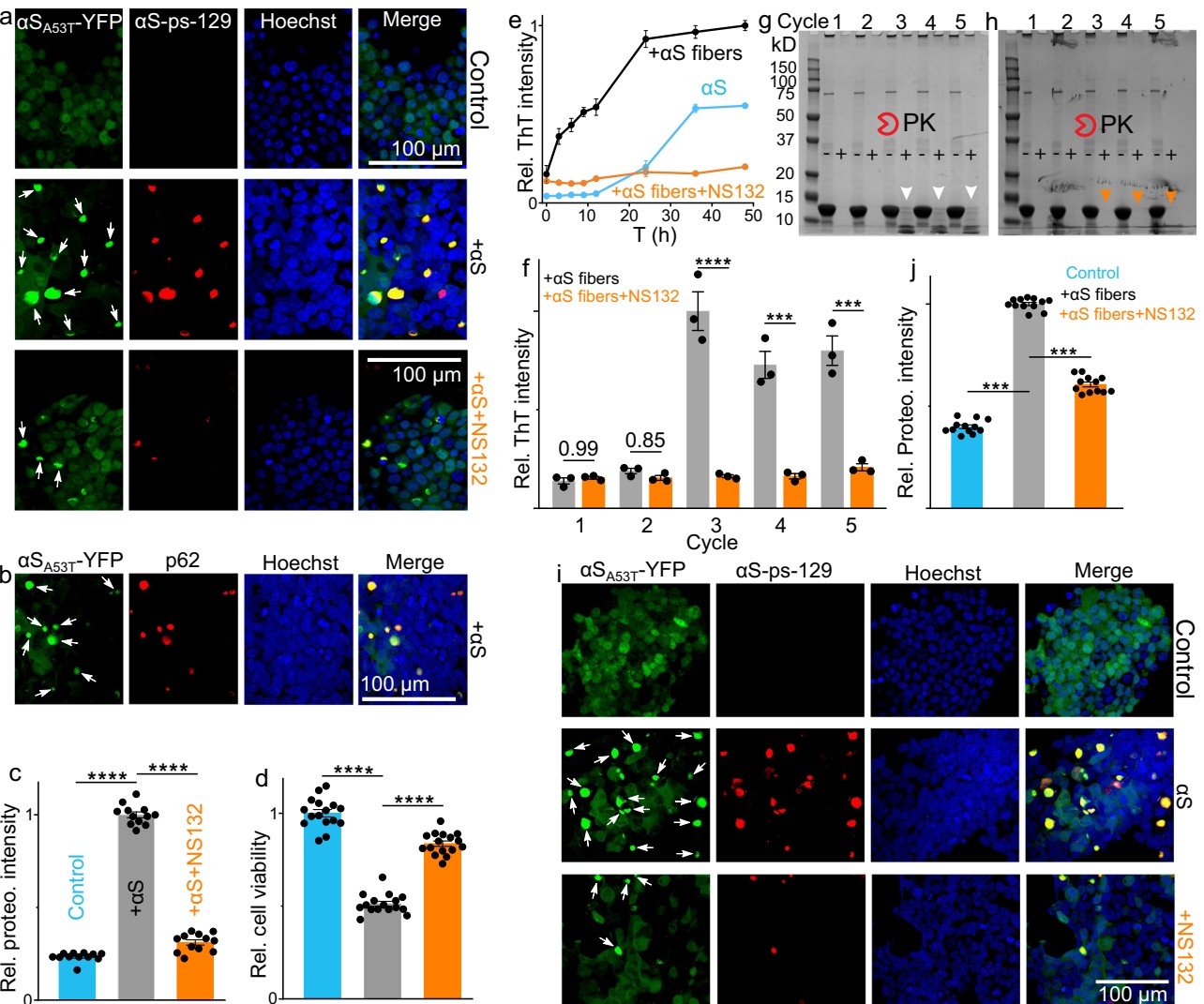

**Fig. 3 | The effect of NS132 on the fibers-catalyzed aggregation of αS.** For all confocal images, inclusions of αS$_{A53T}$-YFP = white arrows, Hoechst = blue. **a** Confocal images of HEK cells treated with 5 μM αS aggregated in the absence and presence of equimolar NS132. αS-pS-129 = red, merge = Hoechst, αS-pS-129, and αS$_{A53T}$-YFP. **b** Confocal images of HEK cells treated with aggregated solution of 5 μM αS. p62 = red, merge = Hoechst, p62, and αS$_{A53T}$-YFP. The relative intensity of Proteostat dye-stained aggregates of αS$_{A53T}$-YFP inclusions (**c**) and relative viability (**d**) of HEK cells treated with 5 μM αS aggregated in the absence and presence of equimolar NS132. **e** The ThT kinetics assay of 100 μM αS catalyzed by αS fibers (20% monomer conc.) in the absence and presence of equimolar NS132. **f** The statistical analysis of the relative ThT intensity of PMCA samples in the absence (gray) and presence (orange) of NS132, before PK treatment. The Bis-tris gels of PMCA samples from all five cycles in the absence (**g**) and presence (**h**) of NS132. (−) and (+) signs = without and with PK treatment, respectively; arrows indicate effect of PK on PMCA

samples from indicated cycle (n = 3 independent experiments). **j** Confocal images of HEK cells after treatment with 5th cycle PMCA samples under the indicated conditions. αS-pS-129 = red, merge = Hoechst, αS-pS-129, and αS$_{A53T}$-YFP. **i** The relative intensity of Proteostat dye-stained aggregates of αS$_{A53T}$-YFP inclusions in HEK cells treated with fifth cycle PMCA samples (2 μM αS in monomer) in the absence and presence of NS132. Confocal images are representative of 10 images from each of 4 independent experiments. The ThT intensity data (e-f) are reported as mean values ± SD, where n = 3 independent experiments. The cell viability and ProteoStat data (**c**, **d**, **j**) are reported as mean values ± s.e.m., where n = 4 biological replicates, each consisting of four technical replicates. P values were determined by one-way ANOVA with Tukey's multiple comparisons test where relevant. *p < 0.05, **p < 0.01, ***p < 0.001, ****p < 0.0001. Source data are provided as a Source Data file.

addition, we also observed that the αS-pS-129 protein (αS phosphorylated at Serine residue 129) colocalized in the aggresome of αS inclusions in HEK cells (Fig. 3a, +αS, red). Moreover, we observed the colocalization of an adaptor protein, p62, in the aggresome of αS inclusions in HEK cells (Fig. 3b, +αS, red) [24,51–54]. It has been suggested that during αS aggregation, p62 recruits the autophagy machinery to the αS inclusions[51–54]. The autophagy machinery regulates many vital cellular processes and its impairment due to αS aggregation can lead to PD and other neurodegenerative disorders[51–54]. In marked contrast, in HEK293 cells transfected with αS aggregated in the presence of NS132, there was a significant decrease in the intracellular αS$_{A53T}$-YFP inclusions (Fig. 3a, +αS + NS132). We also quantified the inclusions

(αS$_{A53T}$-YFP) using a ProteoStat dye-based high throughput 96-well plate reader-based assay recently developed in our lab[24]. The ProteoStat dye-based intensity of HEK cells treated with αS fibers was ~4-5 fold higher than the control (no fibers) (Fig. 3c, +αS, black). In marked contrast, we did not observe a significant difference in the ProteoStat intensity of HEK cells treated with αS fibers+NS132 and the control conditions (Fig. 3c, control and +αS + NS132, blue and orange). Both proteins, including αS-pS-129 and p62 have been shown to be key pathological biomarkers for forming Lewy-body-like aggregates[24,51–54]. The presence of both proteins in the inclusions suggests that these inclusions mimic some features of Lewy-body-like structures that are important events in inducing cytotoxicity in HEK cells[24,51–54]. Therefore,

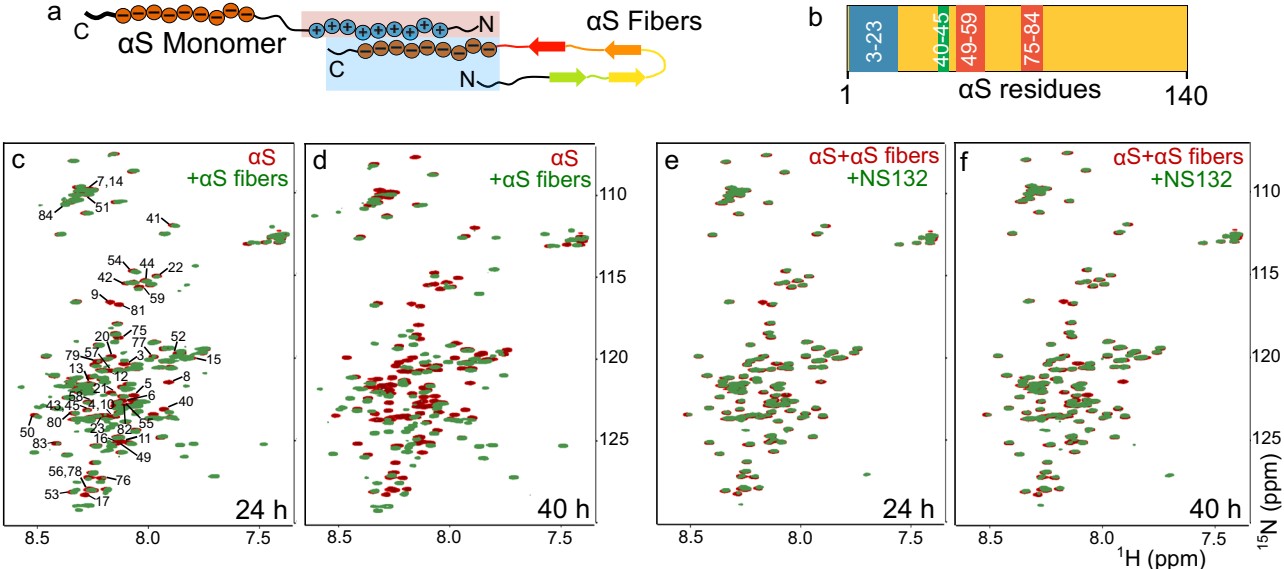

**Fig. 4 | The effect of NS132 on HSQC NMR of $^{15}$N-labeled αS after incubation with αS fibers. a** A model for the proposed interaction of αS monomer with αS fibers. **b** The proposed binding interaction sites of αS fibers on the αS monomer residues. The comparison of the HSQC NMR of the 70 μM $^{15}$N-labeled αS in the absence (red) and presence (green) of αS fibers after 24 h (**c**) and 40 h (**d**). The comparison of the HSQC NMR of the premixed solution of 70 μM $^{15}$N-labeled αS + αS fibers in the absence (red) and presence (green) of equimolar NS132 after 24 h (**e**) and 40 h (**f**).

we used HEK cells to test the cytotoxicity of the aggregated solution of αS in the absence and presence of NS132 (Fig. 3d). The viability of HEK cells was measured using the (3-(4,5-dimethylthiazol-2-yl)-2,5-diphenyltetrazolium bromide) (MTT) reduction based assay. The viability of HEK cells decreased to 52% in the presence of the aggregated solution of αS (Fig. 3d, black); however, the viability of HEK cells was improved to 85% in the presence of the αS aggregated solution with an equimolar ratio of NS132 (Fig. 3d, orange). This data suggest that NS132 doesn't promote the formation of seed competent αS assemblies and the higher order aggresome.

**Antagonist effect of OPs against fibers-catalyzed aggregation of αS**

In addition to the spontaneous accumulation of αS via de novo aggregation, another crucial mechanism for inducing pathology in PD is αS fibers-catalyzed aggregation of αS[20,24,55–61]. Therefore, we monitored the effect of NS132 on the αS fibers-catalyzed aggregation of αS using ThT kinetic assay. The aggregation kinetics of 100 μM αS in the presence of preformed αS fibers (20%, monomer concentration) resulted in the acceleration of monomeric αS aggregation (Fig. 3e, αS, blue line), which is reflected by a significant increase in the ThT signal and the disappearance of the lag phase (Fig. 3e, +αS fibers, black line). The αS fibers-catalyzed aggregation of αS was wholly suppressed by NS132 at an equimolar ratio, as evidenced by a significantly lower ThT signal (Fig. 3e, +αS fibers+NS132, orange line). The TEM data also corroborate well with the ThT assay. We used the solutions from the ThT assay at 60 h for the TEM experiment. We observed an abundance of αS fibers in the fibers-catalyzed aggregation of αS (Supplementary Fig. 12a). In marked contrast, significantly fewer αS fibers were observed in the presence of NS132 at an equimolar ratio (Supplementary Fig. 12b). The antagonist activity of NS132 was also assessed on more robust αS fibers generated using the protein misfolding cyclic amplification (PMCA) technique[20,24,57–60]. The PMCA technique is used to amplify the aggregation of proteins from a small number of fibers from the previous cycle, which generates robust fibers via a nucleation-dependent polymerization model[20,24,57–60]. In the PMCA assay, the fibers of αS are amplified for five cycles using αS monomer and αS fibers from the previous cycle. The ThT intensity for αS aggregation via

PMCA assay increases significantly after cycle two and the intensity stays consistent up to cycle five (Fig. 3f, gray bar). Additionally, PMCA samples (from αS aggregation) from cycle three to cycle five were proteinase K (PK) resistant (Fig. 3g, white arrow), which indicates these αS fibers are very robust and non-degradable after cycle three. In marked contrast, in the presence of NS132 at an equimolar ratio, we did not observe any significant change in the ThT intensity up to cycle five (Fig. 3f, orange bar). More importantly, the PMCA assay samples of αS aggregation in the presence of NS132 from cycle three to cycle five were completely degraded by PK treatment (Fig. 3h, orange arrow). The data suggests that NS132 interacts with αS and generates fiber-incompetent structures, which are easily degradable with PK treatment. We also used TEM images to validate the ThT results and characterize the morphology of αS fibers in the absence and presence of NS132. We took TEM images of the solution from the 5$^{th}$ cycle of the PMCA experiment of αS in the absence and presence of NS132 at an equimolar ratio (Supplementary Fig. 13a, b). We observed that there was an abundance of fibers in the solution from the 5$^{th}$ cycle of the PMCA experiment of αS (Supplementary Fig. 13a). In marked contrast, we observed a very small number of fibers from the 5$^{th}$ cycle of the PMCA experiment of αS in the presence of NS132 at an equimolar ratio (Supplementary Fig. 13b). The morphology of the fibers was amorphous in nature for the PMCA sample (5$^{th}$ cycle) of αS in the presence of NS132 (Supplementary Fig. 13b). We have also shown that αS treated with NS132 did not template αS monomer into fibers from the 1$^{st}$ through the 5$^{th}$ cycle. Together, the data suggests that NS132 interacts with αS and generates fiber-incompetent structures in all five cycles of the PMCA experiments.

To further validate that NS132 generates fiber incompetent structures from the fibers-catalyzed aggregation, we utilized HEK cells that stably express αS-$_{A53T}$-YFP[24,51–54]. The solutions of αS aggregation in the absence and presence of NS132 from the 5$^{th}$ cycle of PMCA (2 μM in monomeric αS, ±NS132) were introduced to the HEK cells in the presence of Lipofectamine 3000 for 24 h. In contrast to the control (no fibers) (Fig. 3i, control), we observed a significant number of inclusions of αS-$_{A53T}$-YFP protein in the presence of αS fibers (Fig. 3i, αS, white arrows). In addition, we also observed the colocalization of αS-pS-129 in the aggresome of αS inclusions in HEK cells (Fig. 3i, αS-pS-129, red).

In contrast, in the presence of the PMCA assay sample from the 5th cycle (+NS132), there was a significant reduction in the number of αS-$_{A53T}$-YFP inclusions (Fig. 3i, +NS132). The ProteoStat intensity of inclusions in HEK cells treated with the PMCA sample from cycle five was ~2–3 fold higher than the control condition (no fibers) (Fig. 3j, black). In marked contrast, we observed a significantly lower ProteoStat intensity in the presence of the PMCA sample from cycle five in the presence of NS132 (Fig. 3j, orange). Clearly, NS132 was a potent antagonist of the fibers-catalyzed aggregation of αS and it generates fiber incompetent off-pathway structures.

We also employed 2D NMR HSQC for atomic-level insights into fibers-catalyzed aggregation of αS and its inhibition by NS132. It has been suggested that the negatively charged flexible C-terminal tail of αS (in fibers) interacts and recruits the positively charged N-terminal segment of αS (in monomer) during the fibers-catalyzed aggregation of αS (Fig. 4a)[62]. We have also shown that NS132 specifically interacts with the N-terminal residues of αS. Therefore, we hypothesize that NS132 will be able to inhibit the interaction of αS (monomer) with αS fibers, which is suggested to be the prerequisite interaction to initiate the seed-catalyzed aggregation of αS. We incubated preformed fibers of αS with 70 μM $^{15}$N-$^{1}$H-uniformly labeled αS monomer and used HSQC NMR to characterize the kinetics of fibers-catalyzed aggregation of αS on a molecular level. The total changes in the volume intensity of the amide peaks of $^{15}$N αS monomer (Fig. 4c, red) in the presence of αS fibers (Fig. 4c, green) suggest that the binding interaction was predominantly toward the N-terminus of αS (Fig. 4b,c). Also, αS fibers interact specifically with four αS sequences on the N-terminus, including residues 3-23, 40-45, 49-59, and 75-84 (Fig. 4b,c). There was an induction of a secondary structure in the αS monomer, indicated by the spreading of the amide peaks toward the $^{1}$H resonances (Fig. 4c, green). At 40 h, more pronounced changes were observed in the location and volume intensity of the amide peaks of the αS monomer, suggesting a much stronger interaction with αS fibers and further induction of a secondary structure in $^{15}$N-$^{1}$H αS monomer (Fig. 4d, green). In marked contrast, no significant change in the volume of the amide peaks of $^{15}$N-$^{1}$H-uniformly labeled αS monomer was observed in the presence of NS132 at an equimolar ratio (Fig. 4e). Even after 40 h, there were fewer and smaller changes in the volume intensity of the amide peaks of $^{15}$N-$^{1}$H-uniformly labeled αS monomer in the presence of NS132 (Fig. 4f). The NMR study suggests that NS132 inhibits the αS monomer-αS fibers interaction by potentially interacting at the N-terminus of αS. Clearly, NS132 is a potent inhibitor of fibers-catalyzed aggregation of αS.

### Optimization of the cell permeability of OPs

To act as a potent antagonist of intracellular αS aggregation in various cellular or in vivo PD models, one of the prerequisite properties of NS132 is that it should permeate the cell membrane. We used the parallel artificial membrane permeation assay (PAMPA) to test the cell permeability of NS132 and compare it with various PAMPA standards of cell permeabilities (Fig. 5a). The cell permeability of NS132 was lower in comparison to the PAMPA medium standard (Fig. 5a), which was most likely a consequence of the COOH functional group. We have also shown that COOH is a very important side chain for the antagonist activity of NS132 against αS aggregation (NS132 vs NS132-P). We surmise that we may under achieve the overall antagonist effect of NS132 against intracellular αS aggregation due to its less than moderate cell permeability. Therefore, to enhance the cell permeability without sacrificing the antagonist activity of NS132, we synthetically converted the carboxylic acid to a hydroxamic acid (NS163, Fig. 5a, b), which is considered to be one of the most common and successful carboxylic acid isosteres in the pharmaceutical industry and has shown higher cell permeability than the former[63]. The cell permeability of the hydroxamic acid analog (NS163) was higher than NS132 (Fig. 5a, b). NS163, similar to NS132, was a potent antagonist of αS aggregation, confirmed

by ThT assay (Fig. 5c) and TEM images (Supplementary Fig. 14a, b). We used ITC and HSQC experiments to characterize and compare the binding affinity and binding site of NS163 with NS132 against αS. Under exact conditions to NS132, the K$_d$ of NS163 for αS was 2.11 ± 0.36 μM with a binding stoichiometry of 1.03 ± 0.04, which was in close agreement of NS132 (K$_d$ = 1.81 ± 0.33 μM) (Fig. 5d). We collected the HSQC NMR of 70 μM $^{15}$N-$^{1}$H-uniformly labeled αS in the absence (Fig. 5e, Supplementary Fig. 15, red) and presence of 70 μM NS163 (Fig. 5e, Supplementary Fig. 15, blue) and compared the change in the signal intensity (volume) of the amide peaks of NS163 with NS132. In the presence of NS163, we observed noticeable volume changes in the amide peaks for specific residues toward the N-terminus, including 9-21 and 36-43 residues (Fig. 5e, blue and green). NS163 also demonstrated changes in the volume intensity of some residues toward the N-terminus of αS (up to 100 residues) (Fig. 5e). It is important to note that the binding site of NS163 was in close proximity to the binding site of NS132 on αS (Fig. 2u and Fig. 5e). Clearly, the PAMPA assay, ThT assay, TEM, ITC, and HSQC NMR study demonstrate that we have improved the cell permeability of NS132 (in NS163) without sacrificing the antagonist activity, binding affinity, or binding site for αS.

We also tested the effect of NS163 on the preformed fibers of αS. A solution of preformed fibers of αS (100 μM, monomer concentration) in 1 × PBS buffer (pH 6.5) was treated with NS163 at an equimolar ratio. The solution was incubated for 24 h, followed by the addition of ThT solution (50 μM) and then checked the fluorescence intensity (ThT dye) of αS fibers in the absence and presence of NS163. We did not observe any significant difference in fluorescence intensity (ThT dye) of αS fibers in the absence and presence of NS163 at an equimolar ratio (Supplementary Fig. 16a). Additionally, we used TEM to further validate the results from the ThT assay. The TEM images of the preformed fibers of αS demonstrated that the morphology and the amount of αS fibers were similar in the absence (Supplementary Fig. 16b) and presence of NS163 (24 h incubation, Supplementary Fig. 16c) at an equimolar ratio. These results indicate that NS163 does not have any noticeable effect on the preformed fibers of αS.

### OPs were more potent antagonists than the reported ligands in the literature

We also compared the antagonist activity of NS163 and NS132 with various reported ligands using ThT aggregation assay. We used multiple reported ligand inhibitors of αS aggregation, which were commercially available, including Bexarotene[64], Tyrosol[65], Valproic acid[66], and EGCG[67]. The ligands were screened against 100 μM αS aggregation (for four days) at an equimolar ratio using ThT aggregation assay. We identified that both NS163 and NS132 were the most potent antagonist of αS aggregation as they completely suppressed the ThT intensity of the αS aggregation (Fig. 5f, Supplementary Fig. 17). Most of these ligands were not very effective antagonists of αS aggregation at an equimolar ratio, as reported by others as well. EGCG was the only ligand which demonstrated a moderate inhibition of αS aggregation as it decreased the overall ThT intensity by ~25% after 96 h. Clearly, NS163 and NS132 are far more potent antagonists than the reported antagonists of αS aggregation.

### Effect of OPs on intracellular αS aggregation in a HEK cell PD model

To test the antagonist activity of NS163 against intracellular αS aggregation, we utilized HEK293 cells, which stably express αS-$_{A53T}$-YFP[23,24,50]. The endogenous monomeric αS-$_{A53T}$-YFP template into fibers when transfected with exogenously added αS fibers in the presence of Lipofectamine 3000 (Fig. 3a)[23,24,50]. In this assay, preformed fibers of αS (5 μM monomer concentration) were introduced to the HEK cells, followed by the introduction of NS163 to the cells after 8 h, which is the time required for the internalization of aS fibers, as we have shown recently. The HEK cells were incubated for an additional

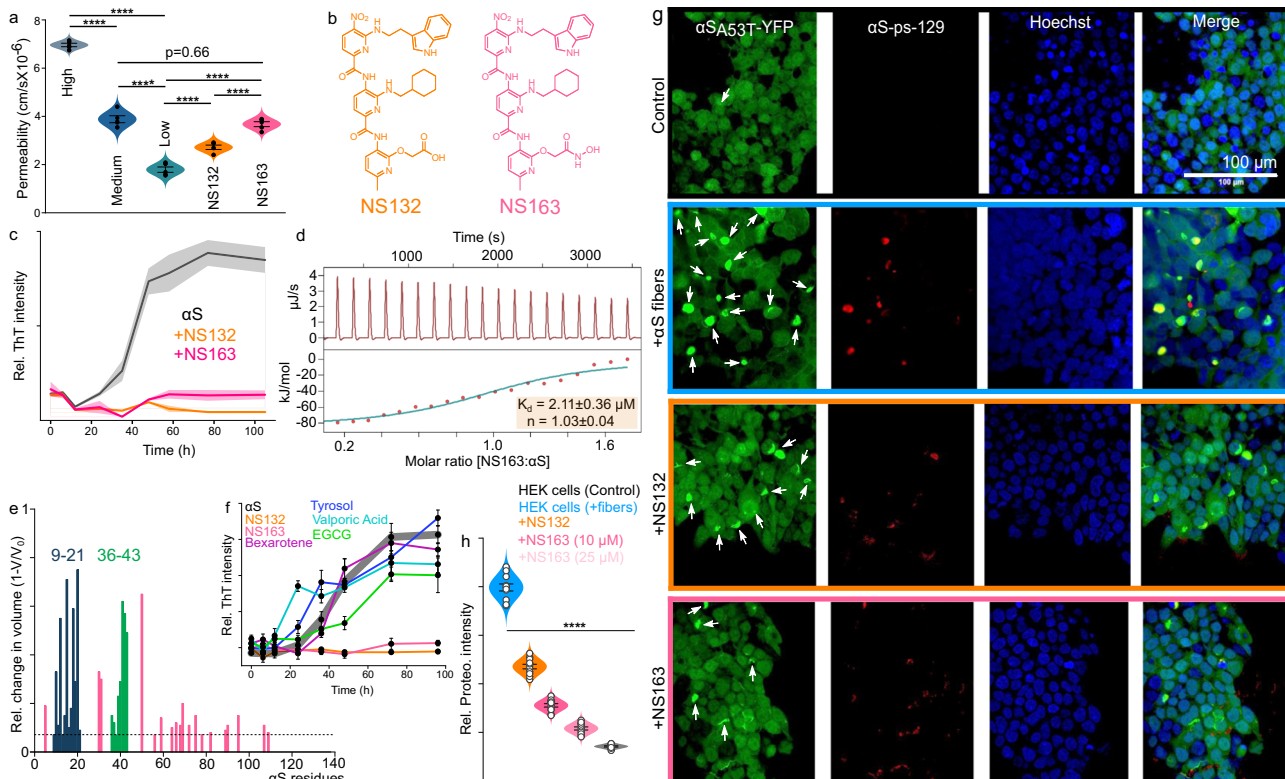

**Fig. 5 | The intracellular inhibition of αS aggregation by OPs in HEK cells.**
**a** Assessment of cell permeability of the indicated ligands using the PAMPA. **b** The chemical structures of NS132 and NS163. **c** The aggregation profile of 100 μM αS in the absence and presence of the indicated ligands at an equimolar ratio. **d** The ITC thermogram for the titration of a solution of NS163 into αS where heat burst curves and the corrected injection heats are represented by the upper panel and the lower panel, respectively. The fit for the curve (teal line) yielded the binding and thermodynamic parameters between NS163 and αS. **e** The relative volume changes (V=volume change, $V_0$ = total volume) of the backbone amide peaks of [15]N-labeled αS (70 μM) in the presence of equimolar NS163. The dashed line represents the reported volume changes in [15]N-labeled αS residues peaks in the presence of NS132 above the value of 5%. The colored sequences (blue and green) are the potential binding sites of NS163 on αS. **f** The comparison of the aggregation profile of 100 μM αS in the absence and presence of the indicated ligands at an equimolar ratio. Confocal images (**g**) and statistical analysis (**h**) of HEK cells treated with the aggregated solution of 5 μM αS, followed by the treatment with the indicated ligands (10 μM). Inclusions of $αS_{A53T}$-YFP = white arrows, Hoechst = blue, αS-pS-129 = red, merge = Hoechst, αS-pS-129, and $αS_{A53T}$-YFP. The ThT intensity data (**c**, **f**) are reported as mean values ± SD, where $n = 3$ independent experiments. Confocal images are representative of 10 images from each of 4 independent experiments. $P$ values were determined by one-way ANOVA with Tukey's multiple comparisons test where relevant. $*p < 0.05$, $**p < 0.01$, $***p < 0.001$, $****p < 0.0001$. Source data are provided as a Source Data file.

18 h, which is the total time (24 h) required for the complete maturation of aS inclusions in the HEK cells. To test the effect of the NS163 on the intracellular inclusions ($αS_{-A53T}$-YFP), we used confocal imaging and a ProteoStat dye-based assay recently developed in our lab[24]. In comparison to the control (Fig. 5g, control), there was an abundance of intracellular inclusions in the presence of αS fibers (Fig. 5, +αS fibers). In contrast, there was a significant reduction in inclusions in the presence of 10 μM NS163 (Fig. 5g, +NS163). The number of inclusions was even further reduced by NS163 at 25 μM concentration, very close to the control (Fig. 5g, h, +NS163). The ProteoStat dye-based intensity of HEK cells treated with αS fibers was ~5 fold higher than the control (no fibers) (Fig. 5h, +αS fibers). NS163 was able to reduce the intracellular inclusions of αS by 76% and 90% at 10 μM and 25 μM, respectively, measured using the ProteoStat dye-based assay (Fig. 5h, +NS163). NS163 inhibits the formation of αS inclusions in HEK cells in a dose dependent manner. NS132 was also able to inhibit the formation of inclusions by 47%, which is less than NS163 (Fig. 5g, h + NS132). It is likely to be a consequence of the better cell permeability of NS163 than NS132. We also compared other tripyridyls with NS163 for their ability to inhibit intracellular αS aggregation using this assay. We used multiple tripyridyls (NS127, NS131, NS132, and NS163) with varying antagonist ability to inhibit αS aggregation. To test their antagonist activity against intracellular αS aggregation, we first determined the cell permeability of the tripyridyls using the PAMPA (Fig. 5a,

Supplementary Fig. 18a). The increasing order of the cell permeability was NS131 < NS127 < NS132 < NS163 (Fig. 5a, Supplementary Fig. 18a). The cell permeability of NS131 was the lowest, most likely because of the sulfonic acid side chain. The cell permeability of NS127 and NS132 was very similar because of the same COOH group and hydrophobic groups (Fig. 5a, Supplementary Fig. 18a). The cell permeability of NS163 was highest because the COOH group was replaced with a hydroxylamine group (Fig. 5a). We utilized the HEK293 cells-based assay as we used for NS163 and NS132 to determine the antagonist activity of other tripyridyls against intracellular αS aggregation. To test the tripyridyls, we first incubated the HEK cells with preformed αS fibers (5 μM, monomer concentration) for 8 h. Subsequently, we added tripyridyls (10 μM) to the HEK cells and then we incubated the cells for additional 18 h, which is the total time (24 h) required for the complete maturation of αS inclusions in the HEK cells. We used confocal imaging and the ProteoStat dye-based assay to monitor the effect of tripyridyls on intracellular αS inclusions (Supplementary Fig. 18a–c). We observed a weak antagonist effect of NS131 (8% decrease in inclusions) on the intracellular αS aggregation even though it was a good antagonist of αS aggregation (from ThT, Fig. 2k, Supplementary Fig. 18b, c). The poor intracellular antagonist activity of NS131 is most likely because of very poor cell permeability (Supplementary Fig. 18a). We observed a noticeable inhibition (~20% decrease in inclusions) of the intracellular aggregation of αS in the presence of NS127 (Supplementary Fig. 18b, c).

NS132 (47% decrease in inclusions) was a much better antagonist than NS127 of the intracellular αS aggregation (Fig. 5g,h, Supplementary Fig. 18b, c). It is worth noting that the cell permeability of NS127 was comparable to NS132 (Fig. 5a, Supplementary Fig. 18a). The ThT assay also validated these results that NS132 is a far more potent antagonist of αS aggregation (Fig. 2k). NS163 was the most potent antagonist of the intracellular αS aggregation (76% decrease in inclusions) among the tripyridyls (Fig. 5g, h, Supplementary Fig. 18b, c). Both NS163 and NS132 were comparable in the inhibition of αS aggregation from ThT and gel shift assays; however, the cell permeability of NS163 is better than NS132 (Fig. 5a). These results demonstrate that higher cell permeability and antagonist activity together allow for better antagonist activity against intracellular aggregation of αS.

We further tested NS163 against intracellular αS aggregation using same HEK293 cells (express αS-$_{A53T}$-YFP)[23,24,50]. In this assay, we added NS163 (10 μM) to the HEK cells and incubated the cells for 16 h, washed the cells and treated with the preformed fibers of αS (5 μM monomer) and incubated for an additional 24 h. The ProteoStat dye-based intensity of HEK cells treated with αS fibers was ~4 fold higher than the control (no fibers) (Supplementary Fig. 19). NS163 was a very potent antagonist as it was able to reduce the intracellular inclusions by 80% at 10 μM, measured using the ProteoStat dye-based assay (Supplementary Fig. 19). Under matched conditions, 10 μM NS132 was able to reduce the intracellular inclusions by 57% (Supplementary Fig. 19). The higher antagonist activity of NS163 than NS132 is likely due to the better cell permeability of the former. Collectively, NS163 is a potent inhibitor of αS fibers-catalyzed aggregation of αS in the cellular milieu. It is important to point out that our potent ligands (NS163 and NS132) were able to inhibit αS preformed fibers templated aggregation of both monomeric WT αS and A53T αS mutant. It has been shown that the WT αS fibers templated aggregation of WT αS and its mutants is predominantly dependent on the interaction of the fibers with the N-terminus of the monomeric WT αS and its mutants[62,68,69]. Therefore, we surmise that the inhibition of WT αS fibers templated aggregation of intracellular monomeric A53T αS mutant is likely due to the interaction of our ligands (NS163 and NS132) toward the N-terminus of A53T αS mutant (similar to WT αS).

### Effect of OPs on intracellular αS aggregation in a *C. elegans* PD model

Next, we investigated the antagonist activity of NS163 against intracellular αS aggregation in a well-established *C. elegans* PD model. The *C. elegans* models have been extensively used to study the underlying mechanisms and therapeutic interventions for neurodegenerative diseases associated with protein aggregation because of the short lifespan (2–3 weeks), tractability to genetic manipulation, distinctive behavioral and neuropathological defects, and high degree of genetic relevance compared to humans[20,24,70]. We utilized a well-established disease model (NL5901 worms), which expresses the αS-fused yellow fluorescent protein (αS-YFP) in the body wall muscle cells[20,24,70]. The PD phenotypic readouts in the NL5901 worms include a gradual increase in inclusions (αS-YFP) in body wall muscle cells and a decline in motility during the aging of the worms[20,24,68]. We tested the antagonist activity of NS163 (50 μM, treatment on day two and four) against αS aggregation-mediated PD phenotypes in NL5901 worms (for details see methods section). We observed a gradual increase in the number of inclusions from day five (~15 inclusions/worm) to day eight (~34 inclusions/worms) and a slight decrease on day nine (~32 inclusions/worm), suggesting a saturation in the number of inclusions after day eight in the worms (Fig. 6a, d, white arrow, blue bar, Supplementary Fig. 20). In marked contrast, there was a substantial decline in the number of inclusions in the presence of NS163 from day five (~4 inclusions/worm) to day nine (~2 inclusions/worm) (Fig. 6c, d, red bar, Supplementary Fig. 20). We also carried out a dose-dependent study of NS163 against the intracellular αS inclusions in NL5901 worms. The

number of αS inclusions in NL5901 worms decreased from ~34/32 (day 8/9) to ~19/20 (day 8/9) and ~9/10 (day 8/9) at 10 and 25 μM NS163, respectively (Supplementary Fig. 21). The data suggest that NS163 inhibits αS inclusions in NL5901 worms in a dose-dependent manner (Supplementary Fig. 21). We observed a decrease in the number of inclusions in the presence of NS132 (lower effect than NS163) as the number of inclusions/worm were ~5 and ~6.6 on day eight and nine, respectively (Fig. 6b, d, orange bar, Supplementary Fig. 20). As we predicted earlier, the cell permeability of NS132 was lower than NS163, which is the likely reason for the lower effect of NS132 (Fig. 6d). It has been shown that the motility of the NL5901 worms decreases during the aging process due to an increase in intracellular αS aggregation, which impairs the muscle cells. We utilized the WMicroTracker ARENA plate reader[24,71] to measure the motility of NL5901 worms in the absence and presence of 50 μM NS163. The relative motility of the worms (under various conditions; see details in methods section) is calculated by using the first data point of the well of the control worms (N2, healthy *C. elegans* strain, Fig. 6e, green bar) as the highest value of one. We observed a significant decline in the motility of NL5901 worms compared to N2 worms (Fig. 6e,f, blue bar). The decline in the motility of NL5901 worms was 70% on day four (Supplementary Fig. 22); however, it is worth noting that we started the motility experiment on day four because of the treatment of worms with ligands on day two and four. However, the decline in the motility of NL5901 worms was much slower (~40%) in comparison to the N2 worms on day three (Supplementary Fig. 22, Supplementary Movie 1 = N2, Supplementary Movie 2 = NL5901 + NS163, Supplementary Movie 3 = NL5901), which further decreased to ~13% on day eight (Supplementary Fig. 22, Supplementary Movie 4 = N2, Supplementary Movie 5 = NL5901 + NS163, Supplementary Movie 6 = NL5901). In marked contrast, the NL5901 worms treated with NS163 on day two and day four resulted in a significant improvement in their motility (Fig. 6e,f, red bar, Supplementary Movie 5). The *C. elegans*-based study suggests that NS163 permeates the body wall muscle cell membrane of the worms and inhibits αS aggregation. We also carried out a dose-dependent study of NS163 against the motility of NL5901 worms. We observed a noticeable increase in the motility of NL5901 worms from 10 to 25 μM NS163 (Supplementary Fig. 23). The data suggests that NS163 improves the motility of NL5901 worms in a dose-dependent manner (Supplementary Fig. 23). As a control, we also checked the effect of NS163 on the motility of the N2 worms. Under matched conditions, NS163 (50 μM) did not demonstrate any significant toxic effect on the motility of N2 worms (Supplementary Fig. 24). Clearly, NS163 did not display any inherent toxicity toward the N2 worms. NS132 also demonstrated an increase in the motility of NL5901 worms, albeit with a less pronounced effect than NS163 (Supplementary Fig. 25).

### Effect of OPs on the degeneration of DA neurons in a *C. elegans* PD model

The aggregation of αS is associated with the neurodegeneration of DA neurons, which is a pathological hallmark of PD. Next, we investigated the neuroprotective effect of NS163 on the intraneuronal αS aggregation-mediated degeneration of DA neurons in a well-established *C. elegans* PD model (UA196)[20,72–75]. The UA196 worms express both human αS and GFP in DA neurons under the control of the dopamine promoter genotype (Pdat-1::GFP; Pdat-1::α-SYN). The expression and aggregation of αS in six DA neurons that are located within the anterior region of worms lead to progressive neurodegeneration characteristics during the aging of UA196 worms. This strain has been used to gain insights into the PD-associated mechanisms and to assess the neuroprotective effect of ligands against the intraneuronal αS aggregation[20,72–75]. During the aging of worms, αS protein aggregates and degenerates the DA neurons as represented by the gradual loss of cell bodies as observed from day 3 (Fig. 7b, c, filled red arrow) to day 5 (Fig. 7d, e, empty red arrow), to day 15 (Fig. 7f, g, empty

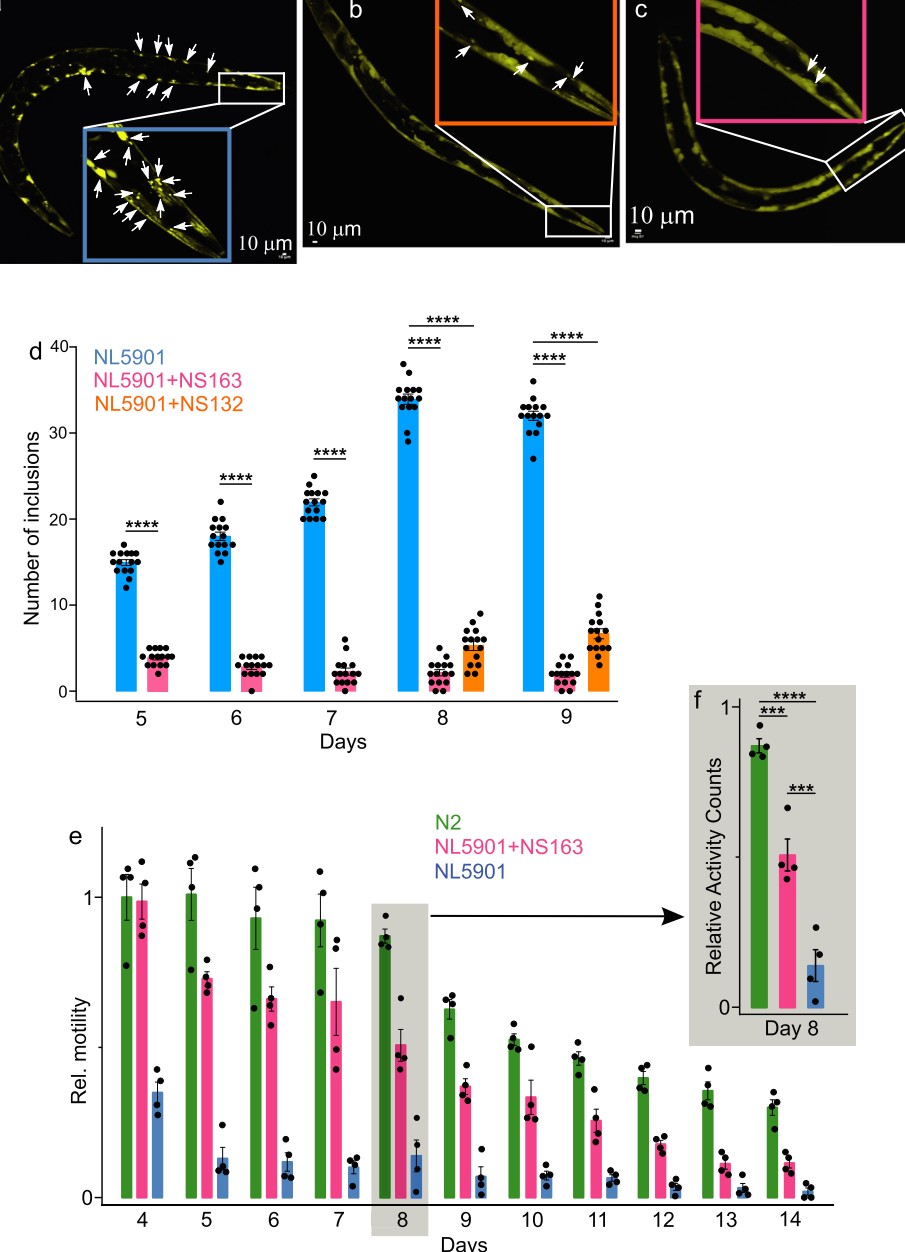

**Fig. 6 | The effects of OPs on αS inclusions and motility in *C. elegans*.** The confocal images of NL5901 worms with αS inclusions (white arrows) in the body wall muscle cells (Day 8) in the absence (**a**) and presence of 50 μM NS132 (**b**) and 50 μM NS163 (**c**). Confocal images are representative of 10 images from each of 4 independent experiments. **d** The number of inclusions for experiment in NL5901 in the absence (blue bar) and presence of 50 μM NS132 (orange bar) and 50 μM NS163 (red bar) from day four to day nine. **e** The motility of N2 (green bar) and NL5901 and statistics (Day 8, **f**) in the absence (blue bar) and presence (red bar) of 50 μM NS163

during the aging process. For each confocal imaging experiment (**a**–**c**), at least 10 worms were used, and the inclusions were counted manually each day. For motility experiments (**e**, **f**), a total of 50 worms were used for each experiment and each condition. The number of inclusions and relative motility data were expressed as mean values ± s.e.m., where *n* = 4 biological replicates and each n consists of four technical replicates. *P* values were determined by one-way ANOVA with Tukey's multiple comparisons test where relevant. \**p* < 0.05, \*\**p* < 0.01, \*\*\**p* < 0.001, \*\*\*\**p* < 0.0001. Source data are provided as a Source Data file.

red arrow). Additionally, there was fragmentation and blebbing of neurites from day 3 (Fig. 7b, c, filled yellow arrow) to day 5 (Fig. 7d, e, empty yellow arrow), which has been observed by others as well[72–75]. The maximum number of healthy DA neurons in one UA196 worm is six on day 3 (no degeneration) and we used 10 worms/biological replicate, which makes a total of 60 healthy DA neurons/biological replicate. Therefore, for each study, the maximum number of healthy DA neurons in UA196 worms is 60 as an average for each biological replicate. There was a gradual decline in the total number of healthy intact DA neurons from day 3, through days 5, 10, and 15, which were 59.5, 43.3,

19.8, and 13.5, respectively (Fig. 7j, blue line). None of the UA196 worms had all six healthy DA neurons on day 5 (Fig. 7k, -NS163) and there was a gradual decline in the total number of healthy DA neurons in UA196 worms (Supplementary Fig. 26a). In the presence of NS163 (50 μM, treatment on day two and four, Fig. 7a), we did not observe any significant decline in the number of DA neurons (Fig. 7h, i, filled red arrow) or any change in the morphology of neurons and the neurites (Fig. 7h, i, filled yellow arrow) up to day 15. The average number of intact DA neurons in UA196 worms treated with NS163 was 59.5, 58.7, and 57.3 on day five, 10, and 15, respectively (Fig. 7j, red line).

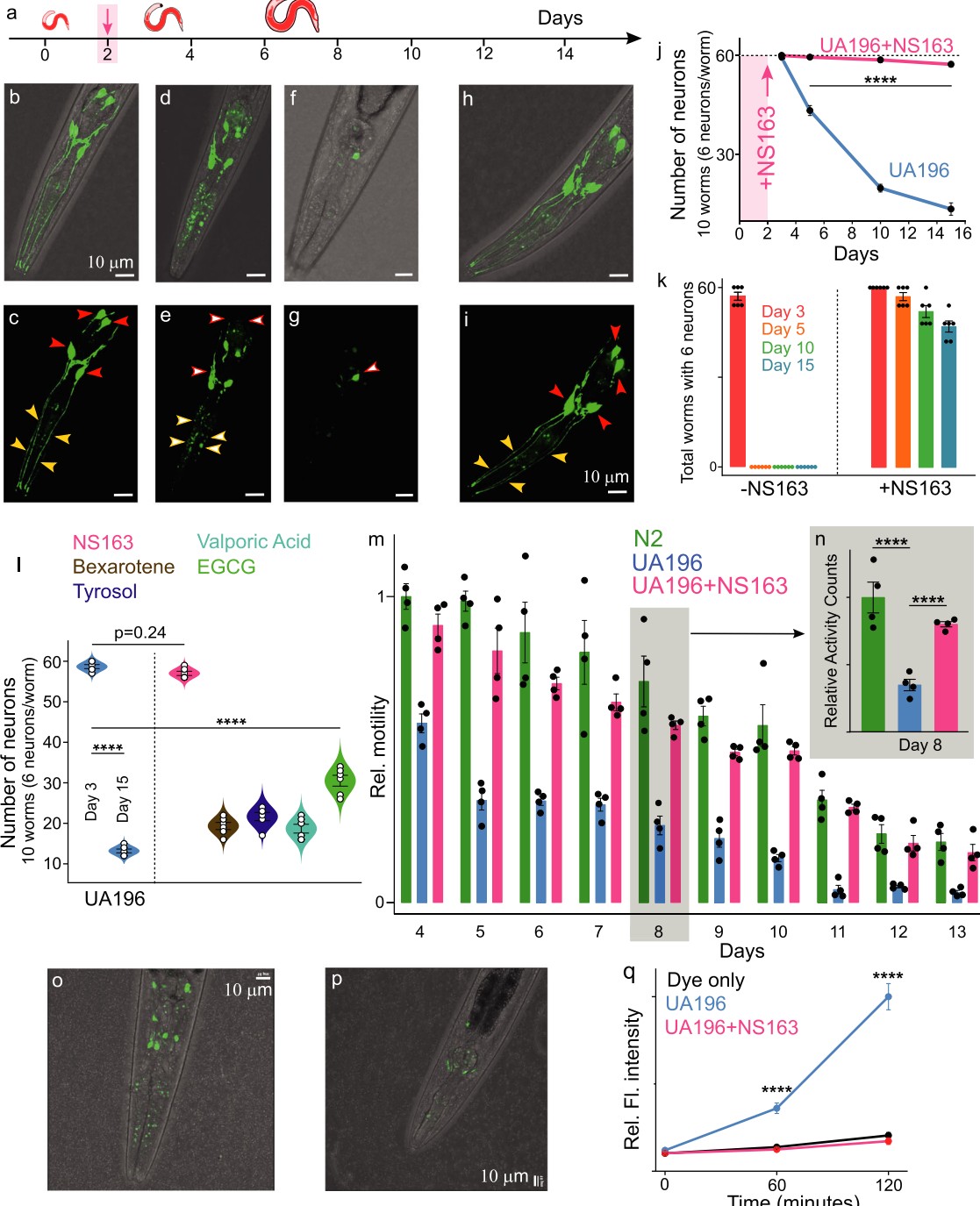

**Fig. 7 | Neuroprotective effect of OPs on the degeneration of DA neurons.**
**a** Schematic of the aging process of UA196 worms and their treatment with ligands. Representative combined images (confocal and white field) of UA196 worms on day 3 (**b**, **c**), day 5 (**d**, **e**), and day 15 (**f**, **g**) and in the presence of 50 μM NS163 (**h**, **i**) on day 15. Filled red and yellow arrows = healthy neurons and neurites, empty red and yellow arrows = degenerated neurons and neurites. **j** Statistics for the total number of healthy neurons in UA196 worms during the aging process in the absence and presence of 50 μM NS163. **k** Number of worms with six healthy neurons in UA196 worms during the aging process in the absence and presence of 50 μM NS163. **l** Statistics for the total number of healthy neurons in UA196 worms during the aging process in the absence and presence of indicated ligands (50 μM). **m** The comparison of the motility for 13 days of N2 and UA196 and statistics (for day eight, **n**) in the absence and presence of 50 μM NS163. Representative confocal images of

UA196 worms (day eight) treated with CM-H2DCFDA dye to quantify the ROS (green) in the abence (**o**) and presence of 50 μM NS163 (**p**, day two treatment with NS163). **q** Statistical analysis of the quantification of the ROS for experiment **o**, **p** For the ROS quantification, at least 50 worms were used, and each condition consisted of *n* = 3 independent experiments, each with five technical replicates. For each confocal imaging experiment, at least 10 worms were used, and the healthy neurons (GFP signal) were counted manually for *n* = 6 independent experiments. Confocal images are representative of 10 images from each of 6 independent experiments. For motility experiments, 50 worms were used in duplicate for each experiment and each with *n* = 4 independent experiments. The data were expressed as mean values ± s.e.m. *P* values were determined by one-way ANOVA with Tukey's multiple comparisons test where relevant. *$p < 0.05$, **$p < 0.01$, ***$p < 0.001$, ****$p < 0.0001$. Source data are provided as a Source Data file.

Additionally, the total number of treated worms with all six intact DA neurons were 57, 52, and 47 on day 5, 10, and 15, respectively (Fig. 7k, Supplementary Fig. 26b). The data suggest that NS163 was very effective in the rescue of the degeneration of DA neurons in UA196 worms.

We compared NS163 and NS132 with other reported ligand inhibitors of αS aggregation in the literature for their ability to rescue the degeneration of DA neurons in UA196 worms under matched conditions (50 μM concentration). The number of intact DA neurons in UA196 worms on day 15 in the presence of Bexarotene, Tyrosol, Valporic acid, and EGCG was 19.3, 21.7, 18.7, and 30.5, respectively (Fig. 7l, Supplementary Fig. 27). Most of the ligands were ineffective in rescuing the degeneration of DA neurons in UA196 worms (Fig. 7l, Supplementary Fig. 27). EGCG was moderate in rescuing the degeneration of DA neurons (Fig. 7l, Supplementary Fig. 27). Our results corroborate well with the published data as some of these molecules demonstrate noticeable rescue effect at one mM concentration[64-66]. Under matched conditions, in the presence of NS132 (50 μM), the total number of intact DA neurons on day 15 was 51, which confirms that NS132 also effectively rescues the degeneration of DA neurons in UA196 worms (Supplementary Fig. 27, 28a–e). The neuroprotective effect of NS163 was better than NS132, indicated by a higher number of intact healthy neurons (Fig. 7k, Supplementary Fig. 28a–e), most likely due to the former's better ability to permeate the cell membrane.

### Effect of OPs on the motility of UA196 worms

The degeneration of DA neurons has been directly linked with the loss of motor functions, resulting in slow motility[13-18,73]. Therefore, we assessed the motility of UA196 in the absence and presence of NS163 using WMicroTracker ARENA plate reader (Fig. 7m, n). There was a significant decline in the motility of the UA196 worms (Fig. 7m, n, blue) during the aging process in comparison to the control worms (Fig. 7m, n, N2, green). In marked contrast, the motility of UA196 worms treated with 50 μM NS163 (day two and day four) was significantly improved during the aging process (Fig. 7m, n, red). The improvement in motility is likely due to the rescue of the degeneration of DA neurons by NS163. NS132 also displayed a neuroprotective effect; therefore, we also assessed its effect on the motility of UA196 worms. Under matched conditions to NS163, we observed a noticeable rescue of the motility of UA196 worms in the presence of NS132 during the aging process (Supplementary Fig. 29). The effect of NS163 was better than NS132 in rescuing the motility of UA196 worms, which is most likely due to the better cell permeability of the former.

### Effect of OPs on the ROS level in UA196 worms

One of the causal agents associated with the etiology of PD is the generation of ROS, which oxidizes lipids, proteins, and DNA[72,74,75]. The neurodegeneration in UA196 worms due to αS aggregation leads to the production of intraworm ROS. The ROS level was determined using a fluorescent probe (CM-H2DCFDA), which reacts with ROS in UA196 worms (day eight) and produces a green-fluorescent signal (Fig. 7o), with intensity increasing for up to 2 h (Fig. 7o, q, blue line)[75]. In marked contrast, UA196 worms treated with 50 μM NS163 (on day two and four) displayed a significant decrease in the intracellular ROS level on day eight (Fig. 7p, q, red line). The signal intensity was very similar for the dye control sample and UA196 worms treated with NS163 (Fig. 7q, red and black line). The data suggest that the decrease in the ROS level in UA196 worms in the presence of NS163 is a consequence of the rescue of degeneration of DA neurons. Similarly, in the presence of NS132, the ROS level was low, which suggests that NS132 was also effective in reducing the ROS level in UA16 worms (Supplementary Fig. 30). It has been shown earlier that the GFP signal in DA neurons does not interfere significantly with the detection of the ROS level by the green fluorescent dye because of the weak signal of the former in comparison to the later[75].

### Effect of OPs on Behavioral deficits in UA196 worms

It has been shown that the lack of dopamine synthesis in DA neurons in *C. elegans* leads to behavioral deficits, including food-sensing behavior[72,73,76,77]. The DA neurons in UA196 worms degenerate over time, which leads to a decrease in the amount of dopamine and NS163 rescues the degeneration of DA neurons. Therefore, we hypothesized that NS163 would be able to rescue behavioral deficits of UA196 worms. We used a chemotaxis assay to assess the effect of NS163 on the behavioral deficits of UA196 worms by following a published protocol (Fig. 8a, see details in methods)[76-79]. We used the ARENA plate reader to measure the chemotaxis index (CI) over time with values from -1.0 to +1.0. The CI value closer to +1 and -1 represents the total time spent by the worms in quadrants with attractant or repellent, respectively. The kinetics of the CI over time was monitored on day three for 2 h for various worms, including N2 (Fig. 8b, light and dark green lines, Supplementary Movie 7), UA196 (Fig. 8b, light and dark blue lines, Supplementary Movie 8), and UA196 + 50 μM NS163 (Fig. 8b, light and dark red lines, Supplementary Movie 9). The kinetic data for the CI suggest that all the worms spent most of their time in the *E. coli* quadrants (attractant), reflected by a value of -1 and a value of ~-1 for ethanol (repellent) during the course of 2 h (Fig. 8b). The data suggest that none of the worms displayed behavioral deficit on day three (Fig. 8b, Supplementary Movie 7, 8, 9), because all the DA neurons in UA196 worms were intact on day three (Fig. 7b, c, j). Therefore, we anticipate a similar behavioral response of UA196 (Fig. 8b, light and dark blue lines, Supplementary Movie 8) and N2 (Fig. 8b, light and dark green lines, Supplementary Movie 7) worms. In marked contrast, on day 10, the CI value of UA196 worms was close to zero for the whole course of the experiment (Fig. 8c, d, light and dark blue lines), which suggests that UA196 worms do not display any preference for ethanol (repellent) or *E. coli* (attractant) (Fig. 8c, d, light and dark blue lines, Supplementary Movie 10). The lack of preference of UA196 worms is due to the behavioral deficits caused by the decrease in dopamine as a result of the degeneration of DA neurons. However, the UA196 worms treated with NS163 strongly favored *E. coli* (Fig. 8c, e, light and dark red lines, Supplementary Movie 11) over the ethanol, similar to the control worms (Fig. 8c, f, light and dark green lines, Supplementary Movie 12), indicated by the CI values. Clearly, NS163 was able to rescue the behavioral deficits of UA196. As a control, we also checked the effect of NS163 on the behavior of the N2 worms. Under matched conditions to the UA196 worms, we treated the N2 worms with NS163 (50 μM) and used ARENA plate reader to measure the CI over time with values from −1.0 to +1.0 (Supplementary Fig. 31). The N2 worms did not display any behavioral deficit up to day 10 as the kinetic data for the CI suggest that all the worms spent most of their time in the *E. coli* quadrants (attractant) reflected by a value of -1 and a value of ~-1 for ethanol (repellent) during the course of 2 h (Supplementary Fig. 31, light and dark green lines). Similarly, we did not observe any behavioral deficits in N2 worms after treatment with NS163 (50 μM) for 10 days as their CI was in very close proximity to that of the N2 worms (Supplementary Fig. 31, light and dark red lines). Both motility and chemotaxis assays suggest that NS163 did not have any inherent toxicity toward the N2 worms. Similarly, NS132 was also able to rescue behavioral deficits in UA196 worms under the matched conditions to NS163 [Supplementary Fig. 32a, b, Supplementary Movie 13 = UA196 + NS132 (day 3), Supplementary Movie 14 = UA196 + NS132 (day 10)].

### Effect of OPs on the production of dopamine in UA196 worms

We have shown that a decrease in the motility in UA196 worms is a consequence of the loss of DA neurons, which is potentially associated with a substantial decrease in dopamine synthesis[72,73]. Therefore, we hypothesized that the motility could be enhanced by administrating

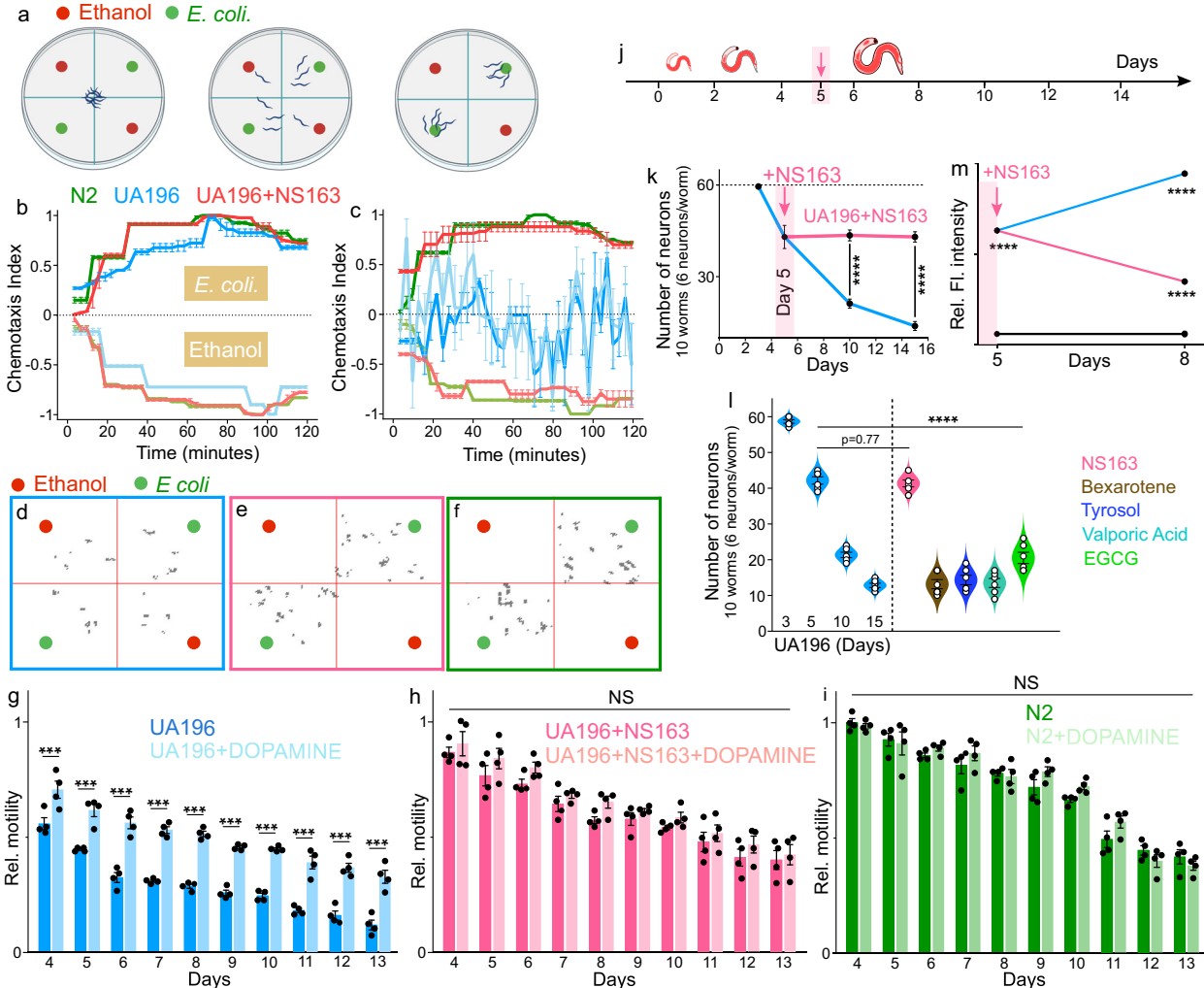

**Fig. 8 | Effect of OPs on the PD phenotypes in UA196 worms. a** Schematic to assess the behavioral deficits in UA196 worms in the presence of ethanol and *E. coli* as a function of time. The CI graph for N2, UA196 worms, and UA196 worms treated with 50 µM NS163 under the indicated conditions on day three (**b**) and day 10 (**c**). Snapshots of the animated videos (at 60 min) collected for the CI for UA196 (**d**), UA196 + 50 µM NS163 (**e**), and N2 (**f**) under the indicated conditions on day 10. The motility of UA196 worms ( + 2 mM Dopamine) in the absence (**g**) and presence (**h**) of 50 µM NS163 and N2 worms (**i**, + 2 mM Dopamine). **j** Schematic of the aging process of UA196 worms and their treatment with ligands (day five) in a post-disease onset PD model. **k** The relative number of healthy neurons in UA196 worms treated with 50 µM NS163 on day five. **l** Statistics for the total number of healthy neurons in UA196 worms when treated on day five with the indicated ligands (50 µM). **m** Statistical analysis of the ROS level on day 8 in UA196 worms when

treated with 50 µM NS163 on day five. For the ROS quantification, at least 50 worms were used, and each condition consisted of $n = 3$ independent experiments, each with 2 technical replicates. For motility experiments, a total of 50 worms were used in duplicate for each experiment and each condition consisted of $n = 4$ independent experiments, each with 2 technical replicates. For chemotaxis assays, a total of 50 worms were used for each experiment and each condition consisted of $n = 3$ independent experiments, each with 2 technical replicates. For confocal imaging, the healthy neurons of 10 worms were counted manually, and each condition (day) consisted of $n = 6$ independent experiments, each with 2 technical replicates. The data represent the mean values ± s.e.m. $P$ values were determined by one-way ANOVA with Tukey's multiple comparisons test where relevant. *$p < 0.05$, **$p < 0.01$, ***$p < 0.001$, ****$p < 0.0001$. Source data are provided as a Source Data file.

dopamine in UA196 worms. The UA196 worms treated with dopamine (2 mM, day two and four) displayed much higher motility during the aging process (Fig. 8g). In marked contrast, we did not observe any significant difference in the motility for 50 µM NS163 (day two and four) treated UA196 worms in the absence and presence of 2 mM dopamine (Fig. 8h). Similarly, we did not observe any significant difference in the motility of the N2 worms in the absence and presence of 2 mM dopamine (Fig. 8i). The data suggest that the improvement in the motility of the dopamine treated UA196 worms is the compensation for the decrease in the dopamine synthesis due to the loss of DA neurons. However, we did not observe any change in the motility for N2 and NS163 treated UA196 worms in the absence and presence of dopamine because of the intact DA neurons and dopamine synthesis. Therefore, the additional dose of dopamine did not affect the motility of worms.

### Effect of OPs on the neurodegeneration in a post-disease onset PD model

NS163 rescues the degeneration of DA neurons in UA196 worms when it was added on day two and day four; however, the effect of NS163 has not been tested in a post-disease onset PD model. For PD, most of the current therapeutic intervention strategies predominantly rely on the post-disease-onset model and the treatment begins after the diagnosis of PD[75,80,81]. Also, during the post-disease onset of PD, the aggregation of αS is facilitated by multiple mechanisms, including the de novo αS aggregation and fibers-catalyzed αS aggregation (via the prion-like spread of αS fibers)[18,24]. Moreover, we have identified that NS163 is a potent inhibitor of the de novo aggregation (Fig. 2g–m), but we had not tested the antagonist activity of NS163 against the fibers-catalyzed αS aggregation. We tested the effect of NS163 on the αS fibers-catalyzed aggregation of αS using ThT kinetic assay. The aggregation

kinetics of 100 µM αS in the presence of preformed αS fibers (20%, monomer eq.) resulted in the acceleration of monomeric αS aggregation (Supplementary Fig. 33, blue line), which is reflected by a significant increase in the ThT signal and the disappearance of the lag phase (Supplementary Fig. 33, black line). The αS fibers-catalyzed aggregation of αS was suppressed by 78% by NS163 at an equimolar ratio, as evidenced by a lower ThT fluorescence signal (Supplementary Fig. 33, red line). The data demonstrate that NS163 is a potent inhibitor of de novo and fibers-catalyzed aggregation of αS. Therefore, we envision that NS163 will be effective in rescuing the degeneration of DA neurons in the post-disease onset model of PD in UA196 worms. We have already shown a gradual decline in the number of DA neurons in UA196 worms from day 3 to day 15 (Fig. 7j, Fig. 8k, blue line, Supplementary Fig. 34). Also, it has been suggested that ~30% neuronal loss or day five in *C. elegans* is a post-disease-onset PD model[75,80,81]. Therefore, we chose day five of the UA196 worms as the post-disease onset PD model, where ~28% of the total DA neurons were degenerated (Fig. 7j, Fig. 8k, blue line, Supplementary Fig. 34). The percent of the degenerated DA neurons (day 3 = 59.5 to day 5 = 44.5 neurons) was calculated based on the lowest number of neurons (day 15 = 13.8) (Fig. 8k, blue line, Supplementary Fig. 34). The UA196 worms were treated with 50 µM NS163 on day five (Fig. 8j, arrow) and the number of intact healthy DA neurons was counted on day 10 and day 15 (Fig. 8k, blue line, Supplementary Fig. 34). The number of intact healthy neurons in the untreated UA196 worms was 44.5, 21.2, 13.8 on day 5, day 10, and day 15, respectively (Fig. 8k, blue line, Supplementary Fig. 34). In marked contrast, in the presence of NS163, the number of intact healthy DA neurons was 43.5 and 43 on day 10 and day 15, respectively (Fig. 8k, red line. Supplementary Fig. 34). NS163 was very effective in rescuing the degeneration of DA neurons as we did not observe any significant loss of DA neurons in the presence of NS163 (Fig. 8k, red line, Supplementary Fig. 34). NS163 was also effective in rescuing the degeneration of DA neurons at lower doses as the number of intact healthy DA neurons was 34/27 (25 µM/10 µM) and 27/17 (25 µM/10 µM) on day 10 and day 15, respectively (Supplementary Fig. 35). Clearly, NS163 was potent in stopping the further degeneration of DA neurons when added on day five to UA196 worms as a post-disease onset PD model.

We also compared the antagonist activity of NS163 and NS132 with other reported inhibitors of αS aggregation for their ability to rescue the degeneration of DA neurons in UA196 worms in the late-stage onset PD model. The ligands (50 µM) were added on day five and the number of intact DA neurons was counted on day 15. The number of intact DA neurons decreased from 58.7 (day 3) to 12.8 (day 15). At this dose, the number of intact DA neurons in UA196 worms on day 15 in the presence of NS163, Bexarotene, Tyrosol, Valporic acid, and EGCG was 41.3, 13.2, 14.3, 13.5, and 20.5, respectively (Fig. 8l). All other ligands except EGCG were not able to rescue the degeneration of DA neurons in UA196 worms (Fig. 8l). EGCG was moderate in rescuing the degeneration of DA neurons. NS132 was also effective (less than NS163) in rescuing degeneration of DA neurons in a post-disease onset PD model (Supplementary Fig. 36a–f, 37). The number of intact DA neurons in UA196 worms on day 15 in the presence of NS132 was 32, which indicates that it was a far better ligand than other reported ligands from literature in rescuing the degeneration of DA neurons (Supplementary Fig. 36a-f, 37). Overall, NS163 was the most potent ligand in rescuing degeneration of DA neurons in the post-disease onset PD model.

The effect of NS163 was also tested on the intraworm ROS level in the post-disease model of UA196 worms. The ROS level of UA196 worms was measured on day five, followed by the addition of 50 µM NS163 and the ROS level was assessed on day eight in the absence and presence of NS163 (Fig. 8m). In UA196 worms, the ROS level increased from day five to day eight due to the increase in the degeneration of DA neurons mediated by αS aggregation (Fig. 8m, blue line). However, the ROS level was significantly decreased in UA196 worms treated with

NS163 (Fig. 8m, red line). Clearly NS163 is a potent ligand in rescuing degeneration of DA neurons (and ROS level) when added to a post-disease onset PD model. NS132 was also very effective in decreasing the ROS level when added on day five to UA196 worms (Supplementary Fig. 38). It is important to note that NS163 and NS132 are able to rescue PD phenotypes in a post-disease onset PD model. Most of the ligands reported in the literature rescue PD phenotypes when added in the early stages of PD models, which does not mimic the clinical landscape for the current therapeutic interventions that rely on the post-diagnosis of PD[75,80,81].

Our data suggest that NS163 and NS132 are potent inhibitors of αS aggregation both in vitro and in vivo models. NS163 and NS132 displayed potent efficacy to rescue PD phenotypes in two *C. elegans* PD models, including intracellular/intraneuronal αS aggregation, degeneration of DA neurons, impaired motility, decreased dopamine synthesis, behavioral deficits, and increased ROS level. More importantly, OPs were very effective in rescuing PD phenotypes in a post-disease onset model. Overall, we have developed a unique and innovative technique to identify potent ligands with tremendous therapeutic potential for the treatment of PD.

## Discussion

The aPPIs are elusive targets as they are of pathological significance and their modulation approaches are directly linked to the discovery of lead therapeutics[1–11]. One of the most effective approaches to modulate aPPIs is the design of synthetic protein mimetics with diverse array of non-proteinogenic side chains, which can complement the sequence and structural topography of aPPI's interfaces[1–11]. OPs are a class of synthetic protein mimetics that imitate the secondary structure of proteins and have been shown to modulate the aggregation of multiple proteins[30–38]. Despite the overall success of OPs as antagonists, no attention has been directed towards enhancing their antagonist activity, which is predominantly dependent on the extension of the chemical diversity of the side chains on OPs[30–38] and the onus is on the tedious synthetic route for the generation of OP libraries as the synthesis requires multiple chromatography steps to add individual side chains on OPs.

We have developed a fragment-based approach (2D-FAST) to append a large number of side chains with a very diverse array of non-proteinogenic side chains on OPs using a highly efficient synthetic method. We used this approach to modulate a dynamic and transient target, which is the self-assembly of αS, a process linked to the onset of PD[12–18]. PD is the second most common neurodegenerative disorder, affecting more than 10 million people worldwide and there is no cure for the disease[20,24]. Therefore, there is a pressing need to identify therapeutic strategies that can prevent or slow down the progression of the disease. We have used the 2D-FAST approach to identify ligands that can modulate the aggregation of αS, which led to the identification of a potent antagonist of de novo and fibers-catalyzed aggregation of αS, and in vitro model of the prion-like spread (PMCA) of αS fibers.

We have also demonstrated that OPs are synthetically tunable ligands with the ability to improve their cell permeability (from NS132 to NS163) without sacrificing their antagonist activity. Under matched conditions, NS163 was a better antagonist than NS132 in rescuing αS aggregation-mediated PD phenotypes in muscle cells and DA neurons in two *C. elegans* PD models, most likely due to the better cell permeability of the former. Moreover, we have applied a post-disease onset PD model that partly resembles and templates the current therapeutic models to identify potent disease-modifying ligands for PD. Both OPs have shown an effective rescue of the degeneration of DA neurons in the post-disease onset PD model of UA196 worms. One of the main reason is that the PD phenotypes in a post-disease onset model is most likley is a consequence of both de novo and fibers-catalyzed aggregation of αS in DA neurons. The degeneration level is not the same for all six DA neurons during the aging of UA196

worms[20,73–76]. We envision that the manifestation of neurotoxicity in DA neurons due to αS aggregation varies in individual DA neurons, which is the reason for different time intervals for the degeneration of individual DA neurons. It is a stepwise process as some DA neurons degenerate earlier and the degeneration of the DA neurons slows down with the aging of worms, which is evidenced by the fact that the number of degenerated DA neurons is 16.2 in the first two days (from day 3 to day 5), 23.5 in the next five days (day 5 to day 10), and 10 in the next five days (day 10 to day 15) (Fig. 8k, blue line). Since the degeneration of DA neurons is a sequential event, there is a possibility that the degeneration of DA neurons at the later stage is a consequence of both the de novo and fibers-catalyzed aggregation of αS. Therefore, ligands that can inhibit both aggregation pathways of αS (de novo and fibers-catalyzed aggregation), could completely rescue the degeneration of DA neurons in a post-disease onset PD model. We have shown that NS163 and NS132 are potent antagonists of both the de novo and fibers-catalyzed aggregation of αS; therefore, both ligands were very effective in rescuing PD phenotypes in the post-disease onset model of UA196 worms. Moreover, the inhibition of αS aggregation (de novo and fibers catalyzed) and prevention of PD phenotypes in vitro *and* in vivo models is a consequence of the interaction of OPs toward the N-terminus of αS and, more specifically, to the suggested aggregation-prone αS sequences[24,47]. The study further highlights the targeting of these sequences as a potential therapeutic approach for the treatment of PD.

We surmise that NS163 will not likely disrupt the physiological function of αS and potently inhibit αS aggregation multiple mechanisms (de novo and fibers-catalyzed). It has been suggested that αS exists in multiple native forms including monomeric, tetrameric, and a multimeric membrane-bound state on the synaptic vesicles[13,16,17]. Moreover, it has been suggested that the interaction between αS and the synaptic vesicles is a key feature in regulating its proposed biological functions, which are neurotransmitter release and synaptic plasticity[12–18]. Also, the binding affinity between the synaptic vesicles and the monomeric αS protein (to regulate its function) is ~300–500 nM[22,24,47]. We have shown that NS163 binds to the monomeric form of αS with a $K_d$ of 1.8 μM. Therefore, the binding affinity between synaptic vesicles and αS is ~five fold higher than the binding affinity between NS163 and αS. Also, we have shown that NS163 not only inhibits the de novo aggregation of αS, but also inhibits the fibers-catalyzed aggregation of αS, which is another important mechanism to potentiate toxicity in neurons. Collectively, NS163 inhibits the aggregation (de novo and fibers-catalyzed) of αS via multiple mechanisms. Therefore, we surmise that NS163 will not disrupt the monomeric αS-synaptic vesicle interaction nor disrupt the physiological function of αS.

We have also shown in the past and the current study that OPs are enzymatically stable in the biological milieu[34,35,37]. Therefore, in the near future, the most potent OPs will be tested for their ability to cross the blood-brain barrier and their pharmacokinetics and pharmacodynamic properties. Subsequently, we will use the most potent OPs in PD mouse models to further assess their pharmaceutical properties and the antagonist activity against PD phenotypes mediated by αS aggregation. The overall neuroprotective effect of OPs on the degeneration of DA neurons (both early and post-disease onset PD models) promises the identification of lead therapeutics for the treatment of PD.

We envision that the OP scaffold-based 2D-FAST could be used to modulate numerous pathological targets, including aPPIs and RNA-protein interactions. This approach is expandable as a greater number of side chains can be appended on OPs because of a very convenient synthetic pathway. We envisage that the 2D-FAST will have a broader impact as both the chemistry and the fragment-based approach could be used for various foldamers (aromatic oligoamides, synthetic protein mimetics, and hybrid macrocycles of peptide and synthetic ligands) for the development of potent antagonists for various pathological protein and nucleic acid targets. The combination of efficient synthetic methodology and the fragment-based approach allows a systematic screening of a large array of non-proteinogenic side chains in a succinct time, which will aid in the identification of high affinity and specificity ligands for various pathological targets. To the best of our knowledge, this is the only example of a fragment-based approach for synthetic protein mimetics to successfully identify potent antagonists of a pathological protein target. We envision that the 2D-FAST will have a profound impact on the development of lead therapeutics for various diseases.

## Methods

The synthesis and characterization of OPs are described in the Supplementary Information.

### Protein expression and purification

The plasmid construct pET11-αS (Addgene, Watertown, MA) was chemically transformed into competent BL21(DE3) cells and plated on ampicillin selection plate. A single colony was used to inoculate a 5 mL starter culture which was kept overnight at 37 °C while shaking at 200 rpm. The following day, the starter culture was used to inoculate 4 L of autoclaved Luria Broth (LB). The culture was incubated at 37 °C and shaken at 200 rpm until the optical density ($OD_{600nm}$) reached 0.8. Then, protein expression was induced for 5 h by adding isopropyl β-D-thiogalactoside (IPTG) at a final 1 mM concentration. Cells were then harvested by centrifugation at $10,600 \times g$ and 4 °C for 15 min. αS purification was carried out using a previously described Osmotic Shock protocol[82]. Briefly, harvested cells were resuspended in Osmotic Shock Buffer (30 mM Tris pH 7.2, 30% sucrose, 2 mM EDTA) and stirred for 15 min at 4 °C. The cells were centrifuged again at $17,000 \times g$ for 10 min at 4 °C and resuspended in cold Milli-Q $H_2O$. The cell suspension was spiked with $MgCl_2$ at a final 5 mM concentration. The suspension was stirred for an additional 5 min and centrifuged at $4,300 \times g$ for 10 min to remove cells. The supernatant was boiled at 95 °C for 15 min to precipitate proteins other than αS. The resulting precipitate was removed by centrifugation at $6000 \times g$ for 20 min and the solution was loaded on Bio-Scale Macro-Prep High Q ion-exchange column (Bio-Rad, Hercules, CA) equilibrated with 20 mM Tris pH 8.0, 25 mM NaCl, 1 mM EDTA, and αS was eluted using 20 mM Tris pH 8.0, 1 M NaCl, 1 mM EDTA. Pure fractions were collected, buffer exchanged to Milli-Q water using Amicon ultra 3 K filters (MilliPoreSigma, Burlington, MA), before lyophilizing and storing the dried protein at −80 °C.

### Aggregation kinetics measurement

Aggregation Kinetics were measured according to a previously described protocol[24]. Briefly, 100 μM αS solutions (250 μL, with or without ligands) were prepared in phosphate buffer saline (PBS, 137 mM NaCl, 2.7 mM KCl, 8 mM $Na_2HPO_4$, 2 mM $KH_2PO_4$, pH 7.3). The solutions were placed in a ThermoMixer (Eppendorf, Hamberg, Germany) at 37 °C with constant shaking at 1300 rpm (rpm). At different time points, 5 μL aliquots of each αS solution were added to 95 μL PBS containing 50 μM Thioflavin T (ThT) dye in a Costar black 96 well plate (Corning Inc., Kennebunk, ME). ThT fluorescence intensity was measured ($\lambda_{ex} = 450$ nm and $\lambda_{em} = 490$ nm) on an Infinite M200PRO plate reader (Tecan, Männedorf, Switzerland), and plotted against time to give a sigmoidal curve typical of amyloid aggregation. For the single point ThT aggregation, 100 μM αS, in the absence and presence of ligands, was aggregated under the described conditions and the ThT fluorescence was reported at 96 h. The reported ThT intensity values are the average of each experiment conducted in triplicate, with the error bars representing the standard deviations or standard error of the mean (s.e.m.) as described in the main manuscript.

## SDS-PAGE assay

An αS solution (100 μM) was prepared in PBS with or without ligands and shaken (13,000 rpm, 37 °C) until the fluorescence plateaued, as observed using the ThT aggregation assay, indicative of the aggregation of αS as a control (four days for most experiments unless otherwise mentioned). The solutions were separated into soluble and insoluble fractions by centrifugation at 22,000 × g for 20 min. The fractions were diluted in 2×Laemmli protein sample loading buffer (Containing 2.1% SDS, Bio-rad, Hercules, CA), boiled at 95 °C for 5 min, and ran on a 12% Mini-PROTEAN precast protein gel (Bio-rad, Hercules, CA). The gel was visualized using ChemiDoc MP Imaging System (Bio-rad, Hercules, CA) and band intensity was quantified using Image Lab image acquisition and analysis software (Bio-rad, Hercules, CA). Densitometry values were averaged from three technical replicates and reported with the error bars representing their standard deviations.

## Transmission electron microscopy (TEM)

Transmission Electron Microscopy was used to visualize protein assemblies after aggregation experiments as previously described[24]. Briefly, aliquots (5 μL) of αS solutions (with or without ligands) were applied on glow-discharged carbon-coated 300-mesh copper grids for 2 min and dried using Kimwipes (Kimberly-Clark, Irving, TX). The copper grids were negatively stained for 1 min with uranyl acetate (0.75%, w/v). The micrographs were taken on an FEI Tecnai G2 Biotwin TEM at 80 kV accelerating voltages.

## Isothermal titration calorimetry (ITC)

The ITC experiments were performed in a NANO-ITC (TA Instruments, New Castle, DE). For each ITC experiment, a solution of 250 μM NS132 in PBS was serially titrated (2 μL injections every 10 sec via rotary syringe, stirring speed at 300 rpm) into a sample cell containing 350 μL of 15 μM αS in the same buffer conditions at 300 sec intervals. The heat associated with each injection was calculated by integrating each heat burst curve using NanoAnalyze software (TA Instruments, New Castle, DE). The associated heat for each injection of NS132 in αS solution was corrected by subtracting heat resulted from the titration of NS132 into buffer (in 1 × PBS) under identical conditions. The corrected heat values were plotted as a function of the molar ratio of NS132 to αS and the plot was fitted using a one binding site independent model available in NanoAnalyze software. The fitting was carried out using 10,000 iterations in the software without any data constraints during the fitting. The ITC titrations were conducted in triplicate and the reported thermodynamic parameters were calculated as an average of three independent experiments.

## Protein misfolding cyclic amplification assay (PMCA)

The PMCA assay was performed according to a previously described protocol[24]. Briefly, an αS solution (100 μM) was prepared in PBS and 60 μL of the solution was placed in a 200 μL Polymerase Chain Reaction (PCR) tube and the mixture was subjected to five 24 h cycles of 1 min shaking (1200 rpm) and 29 min incubation at 37 °C. Every 24 h, 1 μL of αS solution was passaged to seed 60 μL of fresh soluble monomeric αS solution. The seeding cycle was repeated for five days, after which the ThT signal was measured for all five passages. For the preparation of PMCA samples of αS in the presence of NS132, a molar ratio of 1:1 (αS:NS132) was maintained. All experiments were performed in triplicate. The reported ThT intensity values are the average of three separate experiments with the error bars representing the standard deviations.

## Proteinase K digestion of PMCA samples

A Protease K (PK) solution at 50 μg/mL (IBI Scientific, Dubuque, IA) in digestion buffer (10 mM Tris, pH 8.0, 2 mM CaCl₂) was diluted (0.1 eq.) into 30 μL of PMCA solution (with or without NS132) and incubated for 30 min at 37 °C. Subsequently, the samples were diluted (1:1, v/v) in

SDS Protein Gel Loading Dye 2× (Quality Biological, Gaithersburg, MD) and loaded on Mini-PROTEAN TGX Stain-Free Protein Gel (BioRad, Hercules, CA). The gel was stained using the Fairbanks staining method and imaged using ChemiDoc MP (BioRad, Hercules, CA).

## Parallel artificial membrane permeability assay (PAMPA)

The permeability of various ligands was assessed with a PAMPA kit (BioAssay Systems, Hayward, CA) according to a previously described protocol[24]. Briefly, 4% lecithin solution (LS) was prepared in dodecane and 5 μL was applied to the donor plate membrane. Then, 300 μL PBS was added to the acceptor plate. Subsequently, 200 μL of ligand solutions (500 μM) and various permeability standards (high, medium, and low, 500 μM) were added to the donor plate membrane. The donor plate was placed in the acceptor plate and incubated at r.t. for 18 h. The solutions were then moved from the acceptor plate into a clear-bottom 96-well plate (Corning Inc., Corning, NY) and absorbance was recorded at 360 nm for various ligands and 275 nm for permeability standards. The permeability was calculated using Eq. (1) below, where the permeability rate, C, equals $7.72 \times 10^{-6}$, $OD_A$ is the absorbance of the acceptor solution, and $OD_E$ is the absorbance of the equilibrium standard.

$$P_e = C \times -\ln\left(1 - \frac{OD_A}{OD_E}\right) cm/s \qquad (1)$$

## Heteronuclear single quantum coherence (HSQC) NMR spectroscopy

A solution of ¹⁵N αS was prepared by resuspending 1 mg of uniformly labeled ¹⁵N αS (rpeptide, Bogart, GA) in 1 mL Milli-Q water to yield a 70 μM αS solution in 20 mM Tris-HCl pH 7.4, 100 mM NaCl. The protein was buffer exchanged to Milli-Q water, lyophilized, and stored at −80 °C. For the preparation of NMR experiments, the lyophilized ¹⁵N αS powder was dissolved in the NMR buffer (300 μL, 20 mM NaPO₄ pH 6.4, 5% D₂O, v/v) to make a final concentration of 70 μM. The protein concentration was confirmed with a NanoDrop One (Thermo Scientific, Waltham, MA) at 280 nm using an extinction coefficient of 5960 $M^{-1}cm^{-1}$. The solutions were transferred into Shigemi BMS-005 NMR tube (Shigemi, Tokyo, Japan) and the two-dimensional ¹H-¹⁵N HSQC NMR experiments were performed on a 600 MHz Bruker Avance Neo (CU Anschutz, NMR core facility, CO) instrument equipped with a triple resonance HCN cryoprobe. Data was collected at 12 °C on Topspin 4.9.0 software (Bruker, Billerica, MA). The HSQC spectra were recorded using a data matrix consisting of 1024 (t₂, ¹H) × 160 (t₁, ¹⁵N) complex points. Resonance assignments were determined utilizing a previous publication[83] and the data analysis was performed using MestReNova software.

For the HSQC experiments titrating NS132 into ¹⁵N αS, the same experimental conditions were used as above. Incrementally, NS132 was added to a single ¹⁵N αS solution, mixed, kept at r.t. for 10 min and HSQC was recorded under the conditions stated above. For HSQC experiments containing pre-formed fibrils (PFF) of αS, 5.4 eq. of PFF solution (500 μM) was added to monomeric ¹⁵N αS solution, and the spectra was recorded as previously stated.

## Preparation of lipofectamine solution

The Lipofectamine solution was made fresh for each transfection to avoid possible denaturation of reagents in a stored stock solution. For each transfection protocol, the total volume of lipofectamine solution required is equivalent to 30% of the total volume of αS/lipofectamine solution needed. The solution was made by mixing OptiMEM media (Thermo Fisher Scientific, Waltham, MA), Lipofectamine, and P3000 reagents (Invitrogen, Carlsbad, CA) at a ratio of 50:1:1 (v/v/v), homogenized, and incubated at r.t. for 20 min. To prepare a 5 μM αS/Lipofectamine solution, 250 μL of an aggregated solution of αS (100 μM)

was diluted with OptiMEM to a final volume of 3.5 mL and sonicated for 10 min. Subsequently, the Lipofectamine solution (1.5 mL) was combined with the αS protein solution for a total volume of 5 mL. The resulting solution (5 μM) was homogenized and added into their respective wells for transfection of cells.

## MTT (3-(4,5-dimethylthiazol-2-yl)-2,5-diphenyl-2H-tetrazolium bromide) reduction Assay

This assay was performed based on a previous protocol with slight modifications[24]. Unless stated otherwise, the density of Human Embryonic Kidney (HEK) cells (Generous gift by Prof. Mark Diamond's lab) expressing αSA53T-YFP was 150,000 cells/mL for each experiment. This HEK cell line was developed and authenticated by Woerman et al. [50]. The HEK cells were plated (100 μL/ well) and incubated (24 h, 37 °C, 5% $CO_2$) in a Costar 96-well transparent plate (Corning, Kennebunk, ME) with complete DMEM (Thermo Fisher Scientific, Waltham, MA) supplemented with 10% FBS (Hyclone, Logan, UT) media containing 1% penicillin/streptomycin (Life Technologies Corporation, Grand Island, NY). The cell solution replaced with OptiMEM (Life Technologies Co., Grand Island, NY) containing αS aggregated solution in the absence and presence of NS132 at an equimolar ratio, containing the lipofectamine solution. A solution of MTT dye (10 μL, 5 mg/mL in PBS) was added to each well and the plate was covered with aluminum foil and incubated for 3 h. Afterward, the solution was carefully aspirated out without disturbing the formazan crystals and replaced with DMSO (100 μL/well). The crystals were dissolved with gentle shaking of the plate, and subsequently, the absorbance was measured at 570 nm using an Infinite M200 Pro Plate Reader (Tecan, Grödig, Austria). For all MTT assays, four biological replicates were performed, and each biological assay consisted of four technical replicates.

## ProteoStay-Dye assay

This assay was performed based on a previous protocol with slight modifications[24]. The HEK cells (αSA53T-YFP) were plated in 35 mm dishes (Celltreat, Pepperell, MA) with a density of 300,000 cells/dish. The cells were then transfected with αS fibrils (aggregated in the absence and presence of NS132 at an equal molar ratio) using lipofectamine and incubated for 24 h. The Petri dishes were checked for puncta using an Axio Observer Microscope (Carl Zeiss Microscopy, Göttingen, Germany) before measuring intracellular aggregation using the PROTEOSTAT Protein Aggregation Assay Kit (Enzolifesciences, Farmingdale, NY). The media containing dead cells was then transferred into 15 mL Falcon tubes (Corning Science Mexico, Tamaulipas, Mexico). Subsequently, a solution of Detachin (Genlantis, San Diego, CA) was added to detach the live cells, which were then transferred into their respective 15 mL Falcon tubes. After the cells were transferred, the solution was centrifuged for 10 min at 2,000 × g, and the detachin was aspirated out and replaced with 500 μL PBS. The resulting cell pellets were then homogenized, transferred into an autoclaved 1.7 mL microcentrifuge tube, and counted to normalize the data. The microcentrifuge tubes were then centrifuged for 5 min at 1,800 × g and 4 °C (unless stated otherwise, centrifugation parameters remained the same throughout the assay). The process was repeated two more times with 500 μL PBS to properly wash the cell pellets and remove any excess of detachin or FBS media. The cells were treated with 4% paraformaldehyde (Electron Microscopy Sciences, Hatfield, PA), homogenized, and incubated on ice for 30 min. The tubes were then centrifuged for 5 min, the paraformaldehyde solution was replaced with 100 μL PBS, and centrifuged for another 5 min. The 100 μL PBS was replaced with 500 μL PBS containing 0.15% (v/v) Triton X-100 (Oakwood Chemical, Estill, SC), homogenized, and incubated on ice for 20 min. The tubes were then centrifuged again, and the supernatant was replaced with 100 μL PBS. Following another round of centrifugation, the supernatant was replaced with 375 μL PBS

containing the ProteoStat buffer (10%, v/v) and ProteoStat Dye (1%, v/v), according to the manufacturer's guidelines. The cell solutions were homogenized, protected from ambient light, and incubated at r.t. for 20 min. The tubes were centrifuged, and the dye solution was replaced with 400 μL PBS. The cell solution was homogenized and equally aliquoted into the desired number of wells in a Costar 96-well flat black plate and fluorescence was measured ($\lambda_{ex} = 550$ nm, $\lambda_{em} = 600$ nm) on an Infinite M200 Pro Plate Reader. For each condition, four biological replicates were performed, each consisting of three technical replicates.

## Antibodies

The antibodies used in this research are as follows: Phospho-alpha Synuclein (Ser129) Polyclonal Antibody (Biolegend, 825702, 1:1000), Rabbit polyclonal pAb p62/SQSTM1 (Novus Biologicals, NBP1-42821SS, 1:1000). The following secondary antibodies were used: Donkey anti-Mouse IgG (H + L) Highly Cross-Adsorbed Alexa Fluor Plus 647 (ThermoFisher Scientific, A32787TR, 1:1000), Donkey anti-Rabbit IgG (H + L) Highly Cross-Adsorbed Alexa Fluor Plus 647 (ThermoFisher Scientific, A32795, 1:1000).

## Immunofluorescence staining and confocal imaging

The staining and confocal imaging of the HEK cells followed a slightly modified protocol that has been previously reported[24]. The HEK cells expressing YFP-tagged A53T mutant αS were plated onto a μ-slide 8-well plate (Ibidi, Gräfelfing, Germany) at 45,000 cells per well and incubated at 37 °C and 5% $CO_2$ for 24 h. The cells were transfected with fibrils of αS aggregated in the absence and presence of NS132 using Lipofectamine and incubated for 24 h under the same conditions. After confirmation of the transfection by the observation of puncta within the cells under an Axio Observer microscope, the cells were fixed with a solution of 4% paraformaldehyde for 10 min followed by 3 washes with PBS. The paraformaldehyde solution was then replaced with PBS containing 0.15% Triton X-100. After 10 min, the cells were again washed with PBS 3 times. The cells were then treated with PBS containing 1% w/v BSA (Thermo-Fischer Scientific, Rockford, IL) and 0.1% (v/v) Tween-20 (Sigma-Aldrich, St. Louis, MO) for 30 min, followed by 3 more washes with PBS. For imaging, the cells were stained with anti-α-Synuclein Phospho Ser129 (αS phosphorylated at serine residue 129) mouse antibody (BioLegend, San Diego, CA) or anti-p62 rabbit antibody (Millipore Sigma, Burlington, MA) 1 h and washed with PBS 3 times. The cells were then treated with a Donkey anti-mouse tagged with Alexa Fluor Plus 647 (Invitrogen, Rockford, IL) or Donkey anti-rabbit tagged with Alexa Fluor Plus 647 (Invitrogen, Rockford, IL) secondary antibody for 1 h and washed 3 more times with PBS. The cells were then incubated with a 1 mg/mL solution of Hoechst (Cayman Chemical Company, Ann Arbor, MI) in PBS for 10 min and washed 2 times with PBS. The confocal images were obtained using an Olympus Fluoview (FV3000 confocal/2-photon microscope) and analyzed on OlympusViewer in the ImageJ processing software.

## ProteoStat-Dye Assay and confocal imaging to test antagonist activity of ligands (post treatment) against αS fibers templated intracellular aggregation in HEK cells

To test the antagonist activity of ligands against intracellular αS aggregation, we utilized HEK293 cells, which stably express αS-A53T-YFP. In this assay, preformed fibers of αS (5 μM monomer concentration+lipofacomine 3000) were introduced to the HEK cells as mentioned above, followed by the introduction of ligands at various concentrations mentioned in the main manuscript (stock conc. = 10 mM DMSO, DMSO > 0.25%, v/v) to the cells after 8 h. The HEK cells were incubated for an additional 18 h, which makes it a total time of 24 h. To test the effect of the ligands on the intracellular inclusions (αS-A53T-YFP), we used confocal imaging and a ProteoStat dye-based

assay as mentioned earlier. We used the exact conditions as we used in the earlier assays.

## ProteoStat-Dye Assay to test antagonist activity of ligands (pre-treatment) against αS fibers templated intracellular aggregation in HEK cells

To test the antagonist activity of ligands against intracellular αS aggregation, we utilized HEK293 cells, which stably express αS-$_{A53T}$-YFP. In this assay, we added ligands (10 μM, stock conc. = 10 mM DMSO, DMSO > 0.25%, v/v) to the HEK cells and incubated the cells for 16 h to allow the internalization of ligands. Subsequently, the cells were washed three times with 1×PBS (pH 6.5) and treated with the preformed fibers of αS (5 μM monomer concentration+lipofacomine 3000). The HEK cells were incubated for an additional 24 h, which is the total time (24 h) required for the complete maturation of aS inclusions in the HEK cells. To test the effect of the ligands on the intracellular inclusions (αS-$_{A53T}$-YFP), we used a ProteoStat dye-based assay in the absence and presence of ligands as mentioned earlier. We used the exact conditions as we used in the earlier assays.

## C. elegans experiments

**Media and buffers for various worm assays.** The worms were maintained at standard conditions (20 °C at all times) on nematode growth media (NGM) agar in 60 mm plates (CytoOne, Ocala, FL) using *E. coli* OP50 strain as the food source following the previous protocols[24,84,85]. To ensure that worm's colonies don't starve, they were always transferred on new plates with *E. coli* OP50 strain as the food source. All worm strains were replaced with new worm colony after every six months to avoid any genetic mutation, which might affect the disease phenotypes. NGM agar plates, M9 buffer (3 g KH$_2$PO$_4$, 6 g Na$_2$HPO$_4$, 5 g NaCl, 1 mL 1 M MgSO$_4$, milli-Q H$_2$O to 1 L), Chemotaxis (CTX) media plates (2% Agar, 5 mM KHPO$_4$, 1 mM CaCl$_2$, and 1 mM MgSO$_4$), and CTX buffer (5 mM KH$_2$PO$_4$ 1 mM CaCl$_2$, and 1 mM MgSO$_4$) were prepared using previous protocols[24,78,85,86].

NGM agar plates are prepared from an autoclaved solution containing 15 g agar, 2.4 g NaCl, 2 g Tryptone, and 2.72 g KH$_2$PO$_4$ in 1 L of Milli Q water. After cooling to 60 °C in warm water, the following are added per liter: 0.8 ml of 1 M CaCl$_2$, 1 mL cholesterol (5 mg/ml in ethanol), and 1 mL 1 M MgSO4, and (to prevent bacterial and fungal contamination) 1 mL streptomycin (100 mg/ml) and 1 mL nystatin (10 mg/ml). The 60 mm plates are then filled to 2/3 of their volume (10 mL) and left completely still to dry.

(CTX) media 6-well plates are prepared by adding 3 mL per well of the following: 2 g of agar in 100 mL of Milli Q water is autoclaved, before adding 1 M KH$_2$PO$_4$ (0.5 mL), 1 M CaCl2 (0.1 mL), and 1 MgSO$_4$ (0.1 mL) and the solution mixing by shaking.

## Strains

The N2 (wild-type *C. elegans* Bristol strain), NL5901 (*C. elegans* model of PD), and *Escherichia coli* OP50 (*E. coli*, a uracil requiring mutant) strains were obtained from Caenorhabditis Genomics Center (CGC, Minneapolis, MN). The UA196 strain was generously donated by the laboratory of Dr. Guy Caldwell (Department of Biological Science, The University of Alabama, Tuscaloosa, AL, United States)[87]. The sex ratio for all strains is 99.5% female (or hermaphroditic), with the remaining 0.5% being facultative male. The age and number of worms used varies by experiment and are indicated in their respective methods sections.

**NL5901 strain (pkIs2386 [unc-54p::alphasynuclein::YFP+unc-119(+)]).** In NL5901 strain, αS-YFP is expressed in the muscle cells of worms. The aggregates of αS-YFP are visible from day 4 and the maximum number of αS-YFP aggregates are visible round day 8 and day 9.

**UA196 strain [P$_{dat-1}$::α-syn+P$_{dat-1}$::GFP].** In UA196 strain, human αS and GFP are expressed in DA neurons under control of a dopamine transporter-specific promoter [P$_{dat-1}$::α-syn+P$_{dat-1}$::GFP], which results in age-dependent neurodegeneration of six DA neurons.

## Culture methods for *C. elegans* strains

The standard worm conditions were used for the culturing of various worm strains. Briefly, the worms were bleached and synchronized using hypochlorite solution, followed by the incubation of the eggs (at 20 °C) on NGM plates (35 mm, CellTreat Scientific, Pepperell, MA), which were seeded with OP50 (350 μL, 0.5 OD$_{600nm}$) as a food source, **referred to as PLATE 1**. The cultures of OP50 were prepared by incubating 50 mL of LB medium with OP50 18 h at 37 °C and the final OD value was adjusted to 0.5 at 600 nm. The NGM plates were prepared by treating them with 350 μL OP50 and leaving the plates at 20 °C for 3 days. On day two, the worms were transferred (using M9 buffer) to NGM plates (35 mm, CellTreat Scientific, Pepperell, MA) containing 75 μM Fluorodeoxyuridine (FUDR; to prevent worm reproduction and ensure that equal ages of worms were used for the experiment)[24,83] and OP50 as food source, **referred to as PLATE 2**. These NGM plates were prepared using autoclaved NGM media supplemented with 75 μM FUDR (2 mL total liquid). After 12 h, the NGM plates were seeded with 350 μL of OP50 (with OD = 0.5 at 600 nm) at 20 °C. For treating various worm strains with ligands, the worms were transferred on day 2 to the NGM plates treated with ligands (please see below to prepare the NGM plates treated with ligands).

## Preparation of NGM plates with ligands

The NGM plates (with NGM media) were used for the treatment of different worm strains with ligands. The NGM media containing 75 μM FUDR was autoclaved and poured in NGM plates (2 mL total liquid). After 12 h, the NGM plates were seeded with 350 μL of OP50 (with OD = 0.5 at 600 nm) at 20 °C. After 12 h, different doses of ligands (10–50 μM, stock conc. = 10 mM in DMSO, DMSO from 0.1 to 0.5%, v/v) were dissolved in M9 buffer (total volume = 300 μL) and were spotted atop the NGM plates at 20 °C, **referred to as PLATE 3**. The NGM plates treated with the ligands were placed in sterile laminar flow hood at 20 °C for 1 h. The NGM plates treated with ligands were prepared and used within 24 h. We used these plates to treat various disease strains for different assays.

## Motility assay for *C. elegans* (N2, NL5901 and UA196)

Briefly, all worm strains were bleached at the same time and the eggs were incubated on **PLATE 1**. On day two, the worms were divided into two batches. One half of the worms were transferred (using M9 buffer) on **PLATE 2 (without ligand)** and the second half of the worms were transferred to **PLATE 3 (with ligand)**. The concentration of the ligands varied from 10–50 μM as described in the main manuscript. The worms were incubated on **PLATE 2 or PLATE 3** up to day 4 at 20 °C in an incubator with constant humidity. On day four, various worm strains (with and without ligands from PLATE 2 or PLATE 3) were transferred to sterile 24 well plate (CellTreat Scientific, Pepperell, MA) containing liquid media (500 μL/well), **referred to as PLATE 4 A** and containing liquid media with ligands (500 μL/well with 10–50 μM ligand in M9 buffer with 0.1%-0.5% DMSO, v/v), **referred to as PLATE 4B**. The liquid media for **PLATE 4 A/B** was prepared with 67.28% (v/v) of M9 buffer, 75 μM FUDR, 0.1% of 1 M magnesium sulfate (v/v), 0.1% of 1 M calcium chloride (v/v), 2.5% of 1 M potassium phosphate solution (pH 6, v/v), and 30% (v/v) of OP50 (0.5 OD$_{600nm}$). A total of 50 worms per well were manually transferred into **PLATE 4 A/B** at 20 °C and a total of four wells (4 technical replicates) were used for each condition. Subsequently, the worms were incubated for 6 h at 20 °C with constant shaking (rpm = 100) to get acclimated with the solution conditions before starting the motility assay experiment.

To test the effect of each ligand on disease worm strains (NL5901 or UA196), there were four conditions were used in **PLATE 4 A/B:** (1) N2 worms, (2) N2 worms treated with ligand, (3) NL5901 (or UA196) worms, and (4) NL5901 (or UA196) worms treated with the ligand (10–50 μM). The assay was started on day four using the WMicro-Tracker ARENA plate reader (Phylumtech, Santa Fe, Argentina) at 20 °C for 1 h per day over a 14-day period at intervals of 24 h. The WMicro-Tracker ARENA plate reader uses a large array of infrared light microbeams in each well of the plate to detect the interference caused by the movement of the worms. The output by the ARENA plate reader is an average of the overall movement of all worms present in each well, which is denoted as the motility of the worms in each well. Each day, a total of 20 activity scores per well were collected in 1 h. After collecting the data each day, the **PLATE 4 A/B** was again placed on the shaker (100 rpm) at 20 °C in the incubator. The **PLATE 4 A/B** was on the on the shaker (100 rpm) at 20 °C in the incubator the whole duration of the experiment except only when the reading was collected on the WMicroTracker ARENA plate reader for 1 h. For each condition, four biological replicates were performed, and each biological replicate consisted of four technical replicates (four wells). For each condition, the data were expressed as mean and the error bars report the s.e.m. ($n = 4$ independent experiments and each n consisted of a total of four technical replicates).

### Motility assay for *C. elegans* (N2 and UA196) in the presence of dopamine and NS163

This motility assay was performed exactly as the previous assay with slight adjustment in the preparation of the plates for the treatment of UA196 worms. For this assay, the only difference is the 24 well plates were prepared with 2 mM dopamine (**referred to as PLATE 5**) and both 2 mM dopamine and 50 μM NS163 (**referred to as PLATE 6**). To prepare **PLATE 5**, the NGM media containing 75 μM FUDR was autoclaved and poured in NGM plates (2 mL total liquid). After 12 h, the NGM plates were seeded with 350 μL of OP50 (with OD = 0.5 at 600 nm) at 20 °C. After 12 h, 2 mM dopamine dissolved in M9 buffer (total volume = 300 μL) was spotted atop the NGM plates at 20 °C. To prepare **PLATE 6**, the NGM media containing 75 μM FUDR was autoclaved and poured in NGM plates (2 mL total liquid). After 12 h, the NGM plates were seeded with 350 μL of OP50 (with OD = 0.5 at 600 nm) at 20 °C. After 12 h, NS163 (50 μM, stock conc. = 10 °mM in DMSO) and dopamine (2 mM in M9 buffer) were dissolved in M9 buffer (total volume = 300 μL) and were spotted atop the NGM plates at 20 °C. The NGM plates treated with dopamine and NS163 and dopamine were placed in sterile laminar flow hood at 20 °C for 1 h. The NGM plates treated with dopamine and NS163 and dopamine were prepared and used within 24 h.

Similar to the previous section, the motility assay was performed for N2 and UA196 worms in the absence and presence of NS163+dopamine in sterile 24 well plate (CellTreat Scientific, Pepperell, MA) containing liquid media (500 μL/well), liquid media + 2 mM dopamine, and liquid media + 2 mM dopamine + 50 μM NS163. There were six conditions for this experiment: (1) N2 worms, (2) N2 worms treated with 2 mM dopamine, (3) UA196 worms, (4) UA196 worms treated with 2 mM dopamine, (5) UA196 worms treated with 50 μM NS163, and (6) UA196 worms treated with 2 mM dopamine and 50 μM NS163. For each condition, four biological replicates were performed, and each biological replicate consisted of four technical replicates (four wells). For each condition, the data were expressed as mean and the error bars report the s.e.m. (n = 4 independent experiments and each n consisted of a total of four technical replicates).

### Confocal microscopy imaging of early stage treated *C. elegans* (NL5901 and UA196) with ligands

This experiment was performed based on previously described protocols with slight modifications[24,65,88]. Briefly, the worm strains (NL5901 or

UA196) were bleached at the same time and the eggs were incubated on **PLATE 1**. On day two, the worms were divided into two batches. One half of the worms were transferred (using M9 buffer) on **PLATE 2 (without ligand)** and the second half of the worms were transferred to **PLATE 3 (with ligand)**. The concentration of the ligands varied from 10–50 μM as described in the main manuscript. The worms were incubated on **PLATE 2 or PLATE 3** up to day 3 at 20 °C in an incubator with constant humidity. On day four, the ligand (10–50 μM, stock conc. = 10 mM in DMSO, DMSO from 0.1–0.5%, v/v) was added again to **PLATE 3**. For confocal imaging, at least 10 worms per condition (from **PLATE 2 or PLATE 3**) were transferred to a cover slide containing an anesthetic (40 mM sodium azide), and mounted on glass microscope slide containing 2% agarose pads using a reported protocol[13]. The images of the worms were collected using an Olympus Fluoview FV3000 confocal/2-photon microscope (40 x Plan-Apo/1.3 NA objective with DIC capability) and processed using the OlympusViewer in ImageJ software[24]. For NL5901 worms, the inclusions of the aggregated αS (GFP puncta in the muscle cells) were manually counted (10 worms per condition) for the five-day imaging period (day 5 to day 9) for **PLATE 2 or PLATE 3**. For UA196 strain, the number of healthy DA neurons (fluorescence due to GFP in DA neurons) were counted (10 worms per condition) using confocal imaging of the worms on day 3, day 5, day 10, and day 15. For each condition, six biological replicates were performed and at least 10 technical replicates were used for each biological replicate. The data were expressed as mean and the error bars report the s.e.m. (n = 6 independent experiments and each n consisted of a minimum of ten technical replicates).

### Chemotaxis Assay for *C. elegans* (N2 and UA196) in the presence of ligands

For chemotaxis assay, the Chemotaxis (CTX) media plates (2% Agar, 5 mM $KH_2PO_4$, 1 mM $CaCl_2$, 1 mM $MgSO_4$, 75 μM FUDR), **referred to as PLATE 7**, and CTX buffer (5 mM $KH_2PO_4$, 1 mM $CaCl_2$, and 1 mM $MgSO_4$) were prepared using previous protocols[24,78,85,86]. The **PLATE 7 (without ligand)** were treated with ligand (50 μM, stock conc. = 10 mM in DMSO) that was dissolved in CTX buffer (total volume = 300 μL) and was spotted atop the CTX media plates at 20 °C, **referred to as PLATE 8**. For CTX assay, we prepared another plate **referred to as PLATE 9**, which was **PLATE 7** divided into four equal quadrants and designated as A and C (diagonally opposite), B and D (diagonally opposite). A solution of *E coli* (50 μL of 0.5 OD at 600 nm, as an attractant) was placed at ˜ 0.4 cm from the edge of quadrants B and D and the plate was allowed to dry for 1 h at 20 °C. Subsequently, ethanol (10 μL, repellant) was added at ˜0.4 cm from the edge of quadrants A and C at 20 °C. To ensure that the ethanol did not dry, the worms (N2 or UA196) were transferred to this plate within 10 min and the lid of the plate was closed. The experiment on the WMicroTracker ARENA plate reader (Phylumtech, Santa Fe, Argentina) started within 1 h of the transfer of the worms at 20 °C. To carry out the experiment, the worms (N2 or UA196) were bleached at the same time and the eggs were incubated on **PLATE 1**. On day two, the worms were divided into two batches. One half of the worms were transferred (using CTX buffer) on **PLATE 7 (without ligand)** and the second half of the worms were transferred to **PLATE 8 (with 50 μM ligand)** and kept at 20 °C in an incubator. On day 3, a total of 50 worms were transferred to the center of the **PLATE 9** using CTX buffer ( ˜20 μL) and the number of worms were counted under an Olympus microscope (SZ-6145, Waltham, MA). Subsequently, the lid of the plate was closed with parafilm (Bemis Company, Inc., Neenah, WI). The worm activity was monitored using the WMicro-Tracker ARENA plate reader (Phylumtech, Santa Fe, Argentina) at 20 °C for 2 h. We used the exact conditions for day 10 experiment except the ligand was added again (50 μM and 300 μL in CTX buffer, stock conc. = 10 mM in DMSO, DMSO = 0.5%, v/v) to **PLATE 8**. For day 10, 50 worms (N2 or UA196) from **PLATE 7 (without ligand)** or **PLATE 8 (with 50 μM ligand)** were transferred to **PLATE 9** using CTX buffer ( ˜ 20 μL) and

the number of worms were counted under an Olympus microscope (SZ-6145, Waltham, MA). Subsequently, the lid of the plate was closed with parafilm (Bemis Company, Inc., Neenah, WI). The worm activity was monitored using the WMicroTracker ARENA plate reader (Phylumtech, Santa Fe, Argentina) at 20 °C for 2 h. The report was generated using MapPlot option on the WMicroTracker ARENA plate reader. The experiment was conducted for both N2 and UA196 worms (in the absence and presence of 50 μM NS163) on day three and day 10 of the aging process. For each condition, three biological replicates were performed and at least two technical replicates were used for each biological replicate. The data were expressed as mean and the error bars report the s.e.m. (n = 3 independent experiments and each n consisted of two technical replicates).

### Measurement of the ROS level in UA196 worms in the presence of ligands (Quantification of ROS and confocal imaging)

The worm strains were bleached at the same time and the eggs were incubated on **PLATE 1**. On day two, the worms were divided into two batches. One half of the worms were transferred (using M9 buffer) on **PLATE 2 (without ligand)** and the second half of the worms were transferred to **PLATE 3 (with ligand)**. The concentration of the ligands was 50 μM as described in the main manuscript. On day four, the ligand (10–50 μM in M9 buffer with 0.1%-0.5% DMSO, v/v) was spotted again atop the **PLATE 3** at 20 °C. The worms were incubated on **PLATE 2 or PLATE 3** up to day eight at 20 °C in an incubator with constant humidity. On day eight, the worms were transferred into 1.7 mL microcentrifuge tubes using M9 buffer (1 mL) and washed with M9 buffer (1 mL and three times) by centrifugation for 2 min at 700xg and 20 °C. Subsequently, a total of 100 worms/well were transferred to a Costar 96-well black plate (Corning, Kennebunk, ME) containing 100 μL solution, including M9 buffer (89 μL), worm solution (10 μL and 100 worms), and 2′,7′-dichlorofluorescein diacetate (H$_2$DCFDA) dye (1 μL, 50 μM, stock solution = 5 mM in cell culture grade DMSO, 99% pure). As a control, 99 μL of M9 buffer and 1 μL of 5 mM H$_2$DCFDA reaction solution was placed in the wells. Subsequently, the Costar 96-well plate was gently shaken for 30 sec at 20 °C and the fluorescence intensity was measured ($\lambda_{ex}$ = 485 nm and $\lambda_{em}$ = 530 nm) at multiple time points (0 to 120 min) using the Infinite M200 Pro Plate Reader (Tecan, Männedorf, Switzerland)[65,89].

For confocal imaging, the same UA196 worms (in the absence and presence of ligands) used for the quantification of ROS, were transferred from the 96-well plate into **PLATE 2**, air-dried into the sterile laminar flow hood. The worms (~10 worms) were transferred to a cover slide containing an anesthetic (40 mM sodium azide) and mounted on a glass microscope slide containing 2% agarose pads for the confocal imaging as described in the previous sections. This experiment consisted of three biological replicates and each biological replicate consisted of three technical replicates. The data were expressed as mean and the error bars report the SD (n = 3 independent experiments and each n consisted of three technical replicates).

### Confocal imaging of a post-disease onset PD model of UA196 worms in the absence and presence of ligands

The UA196 worms were bleached, and the eggs were incubated on **PLATE 1**. On day two, the UA196 worms were transferred (using M9 buffer) on **PLATE 2 (without ligand)**. On day five, 10 UA196 worms were used to determine the number of healthy DA worms using confocal imaging (as described earlier) prior to the treatment with ligands on day five. In tandem, ~50 worms were transferred to **PLATE 3 (with ligand)** using M9 buffer (10-20 μL) and the plate was incubated at 20 °C in the incubator. On days 10 and 15, 10 UA196 worms from **PLATE 2 (without ligand)** and **PLATE 3 (with ligand)** were used to determine the number of healthy DA worms using confocal imaging (as described earlier). For each condition, six biological replicates were performed and at least 10 technical replicates were used for each

biological replicate. The data were expressed as mean and the error bars report the s.e.m. ($n$ = 6 independent experiments and each n consisted of a minimum of ten technical replicates).

### Measurement of intracellular ROS level in a post-disease onset PD model of UA196 worms in the absence and presence of ligands

The UA196 worms were bleached, and the eggs were incubated on **PLATE 1**. On day two, the UA196 worms were transferred (using M9 buffer) on **PLATE 2 (without ligand)**. On day five, 100 UA196 worms were used to determine the ROS level (as described earlier) prior to the treatment with ligands on day five. In tandem, ~200 worms were transferred to **PLATE 3 (with ligand)** using M9 buffer (10-20 μL) and the plate was incubated at 20 °C in the incubator. On day 8, 100 UA196 worms from **PLATE 2 (without ligand)** and **PLATE 3 (with ligand)** were used to determine the ROS level (as described earlier). For each condition, three biological replicates were performed, and 3 technical replicates were used for each biological replicate. The data were expressed as mean and the error bars report the s.e.m. ($n$ = 3 independent experiments and each n consisted of 3 technical replicates).

### Reporting summary

Further information on research design is available in the Nature Portfolio Reporting Summary linked to this article.

## Data availability

All the datasets generated and analyzed during the current study are also available from the corresponding author. Source data is available for Figs. 1–6 and Supplementary Figs. 4–10, 16–19, and 21–38 in the associated source data file. Source data are provided with this paper.

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

## Acknowledgements

The authors would like to thank the department of chemistry and biochemistry, The Knoebel Institute for Healthy Aging, and the Uni-versity of Denver for the startup funds. The authors also thank the PinS program (University of Denver) for awarding summer undergraduate fellowship to C.M.D. and T.C.F. We would also like to thank Prof. Marc Diamond's lab for the wonderful gift of the HEK cells that stably express YFP-labeled A53T αS mutant (αS-$_{A53T}$-YFP). We would also like to thank the American Parkinson Disease Association Research Grant (S.K.) to support the research conducted in this manuscript. We would also like to thank the Parkinson's Foundation for the summer student fellowship to C.M.D. We would also like to acknowledge the Cancer Center Support Grant (P30CA046934) and the Office of Research Infrastructure Grant (1S10OD025020-01) which allowed us to conduct all HSQC NMRs on the University of Colorado Anschutz Medical Campus.

## Author contributions

S.K. designed and conceived the project with assistance from N.H.S., J.A.J., and J.A. The synthesis of the OP libraries was carried out by N.H.S. and R.A.D. The biophysical study was carried out by J.A., N.H.S., and T.C.F. The NMR study was carried out by J.A., R.A.D., and N.H.S. The HEK cell-based cytotoxicity and confocal microscopy imaging were carried out by T.D.B. and C.M.D. The *C. elegans*-based in vivo experiments (for both PD models), including the confocal imaging, behavioral experi-ments, dopamine study, and the motility study were carried out by J.A.J., C.B., and A.G.T. The paper was written by S.K. with assistance from N.H.S., J.A., J.A.J., T.D.B., and R.A.D.

## Competing interests

This work has been filed for a US provisional patent application by the University of Denver. S. Kumar, N.H. Stillman, and R.A. Dohoney and The University of Denver are the inventors of the patent. The remaining authors declare no competing interests.
