## [Peer Review File · Nature Communications]

Protein Mimetic 2D FAST Rescues Alpha Synuclein Aggregation Mediated Early and Post Disease Parkinson's PhenotypesREVIEWERS' COMMENTS:

Reviewer #1 (Remarks to the Author):

Nicholas H. Stillman et al. conducted a unique study using the 2D-FAST approach for the design and synthesis of oligopyridylamides (OPs), and an array of biophysical assays for the screening and evaluation of the activity of these OPs against the aggregation of alpha synuclein (aSyn). Although this study comprises a monumental work of synthesis and spectroscopic characterization of OPs, the authors have previously conducted similar study synthesizing OPs for aggregation inhibition of other amyloid proteins, and hence a lack of novelty for this screening system and study.

In addition to the lack of novelty, subsequent validation has not adequately demonstrated the rationale of their screening with ThT. The authors did not perform a secondary screening using, for example, a cell membrane permeability assay or a proper cell-based assay to acquire better compounds for the biological context; indeed, they noticed later that their compound was not suitable because of the lack of membrane permeability.

Moreover, there have been many reports of compounds that inhibit aSyn aggregation. It has also not been verified whether the hit compounds obtained by the authors' strategy are superior to existing compounds.

Furthermore, although they have acquired numerous data using a nematode model system, there are concerns about their cultural methods and substantial questions about some of the data.

Specific comments that should be addressed are listed below.

Major points

1. The authors started the evaluation of dipyrindylamides with carboxylic acid functional group in the first side chain considering previous reports, and then they screened the second and third side chains by their capability of inhibiting the aggregation using the ThT assay. Although they synthesized and screened several compounds, the OPs seem to have been created randomly in a blinded manner, without a specific strategy and without, apparently, considering the specific structural characteristics of aSyn. If there is a strategy for the compound deployment, it should be described clearly.

Also, there is missing a deeper and rational discussion of the results that explains the relation between the structural properties of the side groups of the compounds with good activity against the fibril formation. For example, considering the compound NS132, there is a clear tendency, from the first to the third pyridyl side chains, to show an increase in the size of the side group. Is this tendency observed in the other compounds synthesized in this study? How do the authors explain the interaction between NS132, that possesses several hydrophobic moieties, with the regions in the N-terminus domain, positively charged, of aSyn?

In addition, it would be good to discuss from the result of ThT which kind of side chains of the second and third pyridyls effectively suppress the aggregation of aSyn. In other words, it might be useful to speculate the structure-efficacy correlations.

2. The authors indicate that “the data suggest that NS132 interacts with aS and generates off-pathway fiber-incompetent structures”. From the experimental data of the Fig 3, we can only see the absence of fibrillar species. How the authors define the term “off-pathway structures”? Can the authors actually demonstrate the existence of these structures?

3. In the Figs 2p and 3i-l, could the authors discuss why in one case they focus on the chemical shift perturbation (Fig. 2p), but on peak intensities in Fig 3i-l? Also, the manuscript would benefit from showing the data of peak intensities and chemical shift perturbation of the NMR spectra as supplementary material.

4. The screening is limited to the in vitro thioflavin T assay. Certainly, ThT is useful as a first screening, but a major issue is that the degree of reduction in ThT does not always correspond with the effect in cells or in vivo situations. In fact, the authors realized the importance of cell membrane permeability later and needed to modify the structure of the first pyridylamide, which was fixed at the very beginning of the screening. Changing the structure of the first pyridylamide raises suspicion throughout the library creation stage, subsequent screening, and the relevance of the hit compounds obtained.

5. Although the author used a cellular system for the validation, they failed to show the compound efficacy in cells because the authors only treated cells with aSyn pretreated with

the compounds during the fibril formations. It would be more important to see if the compound inhibits the intracellular aggregation induced by the in vitro-created α Syn fibrils without compound treatment. It might be better to screen the efficacy of, i.e., the top 10 compounds from the ThT assay by this cell-based assay. At least a real cell-based assay should be performed on NS132 and NS163 after Fig. 4a, b, and c.

6. In Fig3a, differences in ThT fluorescence between monomeric aSyn and the group supplemented with the 20% of aSyn fibrils seem to be too subtle. Usually, although the ThT value of monomer aSyn is quite low, the presence of aSyn fibrils seeds should lead to a higher value of ThT fluorescence intensity. Why is there only a few-fold difference in fluorescence intensity with and without α Syn fibrils? In addition, important kinetic information would be available for the readers if the raw kinetics is shown here or as supplementary material, as the authors showed in the figure 2.

7. All nematode studies tested a single concentration of 50uM. How did the authors determine the optimal concentration? Concentration dependence needs to be evaluated in some experiments of *C.elegans*, such as in Fig4d and 4h. For motility assay, the evaluation on a specific day might be enough.

8. In figure 4e-f, the expression level of the aSyn seems to be affected by the treatment with compounds since the ubiquitous aSyn-YFP signal was also diminished. It is necessary to confirm if there is any difference in the expression of aSyn between groups, e.g., by western blot. Alternatively, it is desirable to present more images of each group at day8, for example, to see if this image is actually representative.

9. I have a serious concern about the method with the preparation for the locomotion assay. Worms are sensitive to changes in temperature and humidity, so worms need to be cultured in an incubator with constant temperature and controlled humidity. However, some protocols mention that the authors cultured worms at room temperature for 24 hours. It is required to describe what temperature RT means and whether it is constant. Also, if the nematodes are kept still during liquid culture, they are severely compromised due to a lack of oxygen. It is necessary to describe whether shaking is applied, and if so, at what RPM.

10. I also have questions about the motility of NL5901. When properly cultured, NL5901 shows a 10-20% decline in locomotion compared to WT in our laboratory. Other groups have shown a decrease in locomotion up to 50%. It is surprising that the NL5901 show as much as an 80% to 90% decrease in their locomotion. Since a similar phenotype of NL5901 has been observed in previous manuscripts from the authors, I am concerned that some factors might exacerbate a phenotype of the NL5901; e.g., due to infection or the higher temperature during the culture. Does this model manifest temperature sensitive phenotype?

In order to verify the strong phenotype, please submit raw data such as video or tracking data during the locomotion assay. If all data is not available, I would especially like to see the data from Day 4 and 8.

11. I am not so convinced with the experiment starting treatment from day5 for the UA196 model because the results are too dramatic. On day5, 28% of cells have degenerated, and we could imagine that a significant amount of α Syn aggregates have formed in the remaining cells, causing a certain level of cytotoxicity. The effect of NS163 can only be expected to prevent further growth of the aSyn aggregation, which might not be able to stop the degeneration cascade of the remaining cells completely. How can NS163 administration completely inhibit neurodegeneration? Please add some comments in the discussion section.

Minor

Legends in Fig. 2 should be "t" and "u", but they are "k" and "l".

Reviewer #2 (Remarks to the Author):

Stillman et al. describe the identification of a-synuclein (a-syn) protein aggregation inhibitors upon screening a library of synthetic protein mimetics developed using a fragment-based approach. They validate the ability of a first candidate (NS132) to act as an antagonist of a-syn aggregation in spontaneous and seeded reactions in vitro, using an array of biophysical techniques. They identify its binding to the monomeric protein occurring at the N-terminus and involving regions previously shown to be relevant for aggregation, with

a moderate affinity in the micromolar range. They also test whether the molecule's incubation interferes with fibril-induced seeding in a cellular model. Once these activities are validated, they test the permeability of this compound using PAMPA, and finding it moderate; they evolve a related molecule (NS163) which, together with the original one, they assay in two different models of worms. The compounds turn out to have high in vivo antiaggregational and neuroprotective activities.

The approach is novel, the results are overall sound, and the article will be of interest to those in the field. However, I think the authors should still address some critical aspects, which I delineate below.

- A strength of this work is the possibility to attain a structure-function relationship according to the lateral groups each of the developed molecules bears. It is true that in the case of NS132 they generate variants with altered activity, but still, the most relevant information for the reader remains hidden in the set of different molecules analyzed in Fig 2b and ad Fig 2g. I would like to see a detailed analysis of what makes a molecule active and another not. Is it just the presence of one or more aromatic rings together with and hydrogen donor? Is it the sequential order? Is it the nature of the non-polar and polar groups? Seeing and analyzing what these molecules have in common and in what they differ will likely allow for the rational design of optimized molecules, which I think is a powerful feature of this large-scale approach that remains undisclosed in the present form of the work.

- I miss a concentration-dependent toxicity assessment for both cells and *C. elegans*. This is especially important in the latter case since compounds are added apparently early in the development stage, and an accommodation effect might occur; in such a way that only those worms that survive to the treatment are further analyzed, and of course, in this subpopulation, the compound would work efficiently. We have observed such an effect in our lab when looking for drugs for different applications.

- A problem with the approach to bringing it to the clinic is that it targets the monomeric protein and not the toxic forms of the protein, oligomers, and fibrils, both in the unseeded

and seeded assay. Blocking the monomeric protein would also block its function and might have undesired side effects. The authors should elaborate on this. Moreover, it will be interesting to demonstrate whether the molecules bind to fibrils themselves and if they can disaggregate them eventually. These assays are easy to perform and would be informative on the therapeutic potential of the molecules.

- In the cellular assays, what I think is an important experiment is missing. Do cells treated with the compound previous to fibril insult become resistant to intracellular seeding? This is indeed the experiment that would recapitulate what occurs in vivo. The assays shown by the authors are excellent but only inform on the activity of the molecules pre-cell treatment.

- In seeding experiments, authors show the relative values of the Th-T signal at 24 h (Fig 3a). However, this representation is not very informative, and it should be shown the aggregation kinetics in the presence of seeds and the presence and absence of the compound, which are the conventional experiments to show, as the authors have done in panel 2 for unseeded reactions. In the same manner, TEM images of the endpoint of the reactions would be informative of the actual potency of the molecules.

- It turns out that NS163 is the most potent of the two assayed molecules in worms and likely a hit compound for further studies, this higher activity being attributed to a potential higher permeability. However, I found it strange that the binding affinity of this compound to α -synuclein is not shown, and the confirmation by NMR that it acts strictly as NS132, binding precisely the same N-terminal regions is not provided. In the absence of these data is hard to ascertain that both compounds act in the same manner and that the difference is only due to permeability issues. It will be really nice to have access to these studies.

- A last comment is that the *C.elegans* experiments should be explained in more detail. In the way they are described, it is difficult to decipher when and how the compounds are administered since there are no references to the development stages of the animals (commonly, it is referenced from stage L4), for an example of similar work with a more conventional description of the moment at which the drug is administered and the time at which the phenotype is analyzed, please see doi.org/10.1073/pnas.1610586114. Moreover,

the fact that the animals are cultured at 23-25 C instead of the conventional 20 C, makes it difficult to compare the development stages with those of related studies. These details are important and should be clarified since one of the authors' claims is that the compound acts in both early and late stages. How does the state of their animals at these time points compare with those of other compounds tested in the literature?

Reviewer #3 (Remarks to the Author):

In this study, the authors employ 2D-FAST to identify ligands that can reduce the aggregation of alpha-synuclein. They identified two potent antagonists (NS132, NS163; NS163 more effective than NS132) of de novo alpha-synuclein aggregation, and then used in vitro (HEK293 cells) and in vivo (C.elegans) models to test the efficacy of these antagonists. The authors use previously validated C elegans strains (NL5901 and UA196) to cause age-dependent alpha-synuclein aggregation in muscle cells (Fig. 4) and DA neurons (Fig. 5), respectively. The experiments in C. elegans appear convincing, but additional controls and analyses are needed to strengthen the authors conclusions.

Major concerns:

1. Essential information is missing on how the C. elegans strains were prepared for treatment. It is also not clear how the NS163 and NS132 antagonists were administered to the worms? Were worms maintained on agar plates or liquid cultures? Was NS163 and NS132 administered only at day 2 of adulthood and then the agonist was removed, or the agonist was provided continuously after day 2 until day 14? Did the authors conduct an egg preparation to synchronize the worm populations when they conducted the experiments? Along the same lines, how exactly and for how long NS163 was administered in the post-disease onset PD model?

2. I do recognize the thorough analysis of the effect of NS163 on NL5901 worms (fig. 4) and UA196 worms (fig. 5-6). However, in all these experiments a very critical control is missing. They authors need to test the effect of NS163 on N2 worms and evaluate motility rate, lifespan, and chemotaxis. This experiment is critical as it will strengthen the authors' conclusions and further support the therapeutic potential of the NS163 antagonist.

3. The authors have conducted a lot of experiments in Fig. 5 and 6. However, data presentation needs to be improved. For example, the Results state “% of loss of intact DA neurons”. However, there is not a single graph in Fig 5 showing the data as percentage (%). In Fig 5k-n, the authors show at the top of the y-axis as “60 neurons”. Not clear where this number is coming from. Hence, I am confused about the extent of “rescue” of DA degeneration that NS163 provides. In general, the labeling of the y axis (e.g., measurement units) can be improved in most graphs of this paper.

4. New confocal microscopy images are needed at higher resolution in Figures 4 and 5. In fig 4, it is unclear why the authors focus on head muscle only; a-Synuclein-YFP aggregation should be observed along the body-wall muscles throughout the A-P axis of the animal.

5. The flow of data presentation in Figs 5 and 6 can be improved and data need to be re-organized. It was hard to read that part of the Results as the reader needs to go back and forth between data in Fig 5, then in Fig 6, and the back to Fig 5.

Minor issues:

1. “Chemical space”: can the authors define this term for the non-expert reader?
2. In Fig 4h and other panels, the relative motility rate must be defined.
3. Statistical comparisons missing in Fig k, l, m.
4. In multiple places in the text, the authors use “subjective” language (e.g., the effect of NS163 was “better” than NS132; ...a “good” correlation between the inhibition of...It is a “remarkable” finding...; ...with “tremendous” therapeutic potential...”. The authors should replace these words with more accurate, less subjective language.
5. The authors should remove claims of novelty, such as “for the first time”, as such claims are hard to verify and do not add that much to an otherwise very interesting paper.
6. The claim in Discussion that the authors have developed a novel post-disease onset PD model is an overstatement and should be toned down.

Reviewers' comments:

Reviewer #1 (Remarks to the Author):

Nicholas H. Stillman et al. conducted a unique study using the 2D-FAST approach for the design and synthesis of oligopyridylamides (OPs), and an array of biophysical assays for the screening and evaluation of the activity of these OPs against the aggregation of alpha synuclein (aSyn). Although this study comprises a monumental work of synthesis and spectroscopic characterization of OPs, the authors have previously conducted similar study synthesizing OPs for aggregation inhibition of other amyloid proteins, and hence a lack of novelty for this screening system and study.

Response:

We agree with the reviewer that OPs have been utilized earlier to identify potent inhibitors of amyloid proteins. However, one of the main strengths of this manuscript is the development of a new approach (2D FAST), which is novel in many aspects. We apologize for not highlighting these novel aspects of our 2D FAST in the main manuscript. Here, we compare our 2D FAST method with the old method and highlight the novel aspect of our 2D FAST. We have modified the manuscript to highlight the main advantages of our 2D-FAST method over the old method and now it reads as follows in the manuscript:

“In the old method, we used only a small library of presynthesized tripyridyls with a very limited number of side chains on OPs (Figure 1 A, Table 1, old method). Additionally, the synthesis was very tedious, including several numbers of synthetic steps (Figure 1A, 14 steps, Table 1, old method), several column chromatography steps (Figure 1A, 11 steps, Table 1, old method), and very low total % yield to synthesize one tripyridyl (Table 1, old method). There was no systematic optimization of the antagonist activity of OPs against α S aggregation. Also, in the old approach, we never reported the optimization of cell permeability or any other pharmaceutical properties of the most potent antagonist OP. We have developed a 2D-FAST by combining fragment and structure-based techniques into the OP scaffold in order to systematically optimize the antagonist activity against α S aggregation (Fig. 1B,C). The fragment-based approach has emerged as a promising method for drug discovery to identify high-affinity ligands against various pathological targets, including aPPIs³⁹⁻⁴⁶. The OP is an ideal scaffold for the fragment-based approach because the antagonist activity of OPs against their biological targets has been shown to increase with increasing side chains (monopyridyl<dipyridyl<tripyridyl)³²⁻³⁵. In 2D-FAST, the 2D consists of the side chains and the number of pyridyls groups in OPs (Fig. 1C). There are several innovative features of our 2D-FAST for OPs, including (1) Use of common precursors for the elongation of OP from mono- to di- to tri-pyridyl synthesis; (2) A significant improvement in the synthetic procedure of OPs, including a smaller number of synthetic steps (8 steps), much higher % yield (>20 fold), and very few chromatography steps (Table 1, new method); (3) Introduction of a large chemically diverse library of side chains (21 side chains, two times more than the old method) on OPs that mimic to a higher number of amino acid side chains of proteins, which will aid in enhancing the affinity and specificity of OPs toward protein target; (4) Use of a fragment-based approach for systematic optimization of the antagonist activity of OPs against α S aggregation, (5) Enhancement of the cell

permeability of the most potent ligand without sacrificing its antagonist activity. To the best of our knowledge, this is the first report of the fragment-based approach that has been applied to a synthetic protein mimetic.

Properties	Old method	2D FAST
*Total % Yield	0.78%-3.61%	17.57%-78.15% (>20 fold higher)
Chromatography steps	11	0 [#]
Synthetic steps	14	8
Side chains mimicking natural amino acid side chains	11	21 (~ 2 fold higher)
Total tripyridyls	One tripyridyl	Tripyridyl Library

Table 1: Comparison of the properties to synthesize a tripyridyl with three distinct side chains using the old method and our new method. **The overall % yield was calculated by multiplying the yield of each synthetic step.* [#] *>90% of side chains did not require column chromatography.*

Fig. 1. A schematic for the comparison of the old method (A) and the 2D-FAST (B). A, Synthesis of the monopyrindyls with various side chains ($R_{i,j,k}$) and the number of synthetic steps. (Inset) The chemical structures

of the side chains on OPs. A flowchart for the synthesis of dipyriddyis and tripyriddyis and their testing against various biological targets. **a**, ROH, NaH or Na metal, toluene, 50 min. at 0 °C, then 5 h at r.t. **b**, Screening and identification of the potent antagonist monopyriddyis against α S aggregation using ThT aggregation assay. **c,g**, Pd/C, H₂ (g), EtOAc, 3 h at r.t. **d**, 6-chloro-5-nitropicolinic acid, DCM (anhydrous), triethylamine, thionyl chloride, 0 °C to r.t., 45 min. **e,i**, Primary amine/thiol, DIPEA, DCM, 3 h at r.t. **f**, Screening and identification of the potent antagonist dipyriddyis against α S aggregation using ThT aggregation assay. **h**, 6-chloro-5-nitropicolinoyl chloride, dichloroethane (DCE), saturated sodium bicarbonate (NaHCO₃), 10 min at 0 °C **j**, Screening and identification of the potent antagonist tripyriddyis against α S aggregation using ThT aggregation assay. **C**, The representation of two dimensions in the 2D-FAST.

In conclusion, we believe that we have introduced many novel aspects to our new method.

In addition to the lack of novelty, subsequent validation has not adequately demonstrated the rationale of their screening with ThT. The authors did not perform a secondary screening using, for example, a cell membrane permeability assay or a proper cell-based assay to acquire better compounds for the biological context; indeed, they noticed later that their compound was not suitable because of the lack of membrane permeability.

Response:

We used the ThT assay as a high throughout approach to identify potent antagonists of α S aggregation. For screening of library of ligands, ThT dye-based aggregation has been traditionally used as a high throughput screening method by various labs for amyloid proteins. As suggested by the reviewer, we have now used two additional assays, including a gel shift assay and a HEK cell-based assay to assess the antagonist activity of our ligands in the biological context.

We have included a secondary screening method to validate the results of the ThT screening method. We have used a sodium dodecyl sulphate–polyacrylamide gel electrophoresis (SDS-PAGE) assay in the absence and presence of the monopyriddyis, dipyriddyis, and tripyriddyis. In the SDS-PAGE assay, we used multiple monopyriddyis, dipyriddyis, and tripyriddyis with varying antagonist activity (from ThT assay) and we used exactly identical solution conditions of the ThT assay. The soluble and insoluble fractions of α S were separated in the presence of monopyriddyis, dipyriddyis, or tripyriddyis and quantified using SDS-PAGE assay. The antagonist activity of the monopyriddyis, dipyriddyis, and tripyriddyis determined using the SDS-PAGE assay corroborates well with the ThT screening assay. The detailed experimental procedure has been added to the materials and methods and the results have been included in the main manuscript and the supplementary information for monopyriddyis (Fig. 2a-d, Supplementary Fig. 4a), dipyriddyis (Fig. 2e-h, Supplementary Fig. 5a-e) and for tripyriddyis (Fig. 2k-q and Supplementary Fig. 7a-d). The explanation has been added in the main text and it reads as below:

Fig. 2. Identification and characterization of the potent antagonist of α S aggregation using the 2D-FAST.

The graphical representation of the ThT intensity of 100 μ M α S aggregation for four days in the absence and presence of monopyridyls (a), dipyrindyls (e), and tripyridyls (k) at an equimolar ratio. The arrow indicates the most potent antagonist of α S aggregation. The generic chemical structures of monopyridyls (b), dipyrindyls (f), and tripyridyls (l). The most potent monopyridyls (c), dipyrindyls (g), and tripyridyls (m) antagonists of α S aggregation. SDS-PAGE gel shift assay analysis of 100 μ M α S aggregation after four days in the absence and presence of monopyridyls (d) and dipyrindyls/tripyridyls (q) at the indicated molar ratios. The aggregation profile of 100 μ M α S in the absence and presence of NS55 (h) and NS132 (n) at an equimolar ratio. TEM images of 100 μ M α S aggregated for four days in the absence (i,o) and presence of NS55 (j) and NS132 (p) at an equimolar ratio. r, The ThT intensity of α S aggregation after four days in the absence and presence of NS132 derivatives. s, Chemical structures of NS132 derivatives. t, The ITC thermogram for the titration of a solution of NS132 into α S where heat burst curves and the corrected injection heats are represented by the upper panel and the lower panel, respectively. u, Graphical presentation of the relative volume changes of the backbone amide residue peaks of 15 N-labeled α S (70 μ M) in the presence of NS132 at an equimolar ratio. The colored sequences (blue and green) are the potential binding sites of NS132 on α S. The dashed line represents the reported volume changes in 15 N-labeled α S residues peaks in the presence of NS132 above the value of 5%. The ThT experiments were conducted three times and the reported change in the ThT intensity was an average of three separate experiments. The data were expressed as mean and the error bars report the S.D. The statistical analysis was performed using ANOVA with Tukey's multiple comparison test. * $p < 0.05$, ** $p < 0.01$, *** $p < 0.001$.

Supplementary Fig. 4. The graphical representation of the ThT intensity of 100 μ M α S aggregation for four days in the absence and presence of NS55 (a) and NS132 (b) in 1 \times PBS at substoichiometric ratio (for NS55, α S:ligand, 1:0.5, and for NS132, α S:ligand, 1:0.1).

“We used SDS-PAGE (Sodium dodecyl sulphate polyacrylamide gel electrophoresis) as a complementary assay to validate ThT results and to further characterize the antagonist activity of the monopyridyls against α S aggregation. We used a total of four monopyridyls for the SDS-PAGE assay with varying antagonist activity against α S aggregation. For the SDS-PAGE assay, a solution of 100 μ M α S was aggregated for four days in the aggregation buffer (1 \times PBS buffer, pH 6.5) in the absence and presence of the monopyridyls at an equimolar ratio. Subsequently, the α S solutions were centrifuged to separate α S aggregates from the soluble α S. Afterward, the samples were boiled at 95 $^{\circ}$ C for 5 min to disassemble α S aggregates and examined using the SDS-PAGE assay (Fig. 2d, Supplementary Fig. 4a). The band intensities of the gels in SDS-PAGE assay were quantified using ImageJ software. In the absence of ligands, ~12% of α S was found in the soluble fraction and the rest was found in the insoluble fraction (Fig. 2d, Supplementary Fig. 4a,b). In the presence of NS41 at molar ratios of 1:1 and 2:1 (NS41: α S), 48% and 73% of α S protein were found in the soluble fraction, respectively, and the rest of α S protein was found in the insoluble fraction (Fig. 2d, Supplementary Fig. 4a,b). The most potent ligand from ThT assay (NS41) also demonstrated the highest amount of α S in the soluble fraction. Similarly, the least effective monopyridyl ligand (RD247) has the highest ThT intensity and also demonstrated the highest amount of α S (87%)

in the insoluble fraction, a value very close to the untreated α S protein (87% insoluble, Fig. 2d, Supplementary Fig. 2a,b). The antagonist activity of the monopyridyls corroborated well from both ThT and the SDS-PAGE assays (Fig. 2a-d, Supplementary Fig. 2a,b). The higher the antagonist activity of monopyridyls against α S aggregation, the lower is the ThT intensity and the higher the amount of α S in the soluble fraction (Fig. 2a-d, Supplementary Fig. 4a,b). Together, NS41 was identified as the most potent monopyridyl antagonist of α S aggregation.”

Supplementary Fig. 5. The graphical representation of the ThT intensity (a), SDS-PAGE gel analysis (b), and SDS-PAGE gels (c-e) of 100 μ M α S aggregation for four days in the absence and presence of dipyrindyls at an equimolar ratio.

“Similar to monopyridyls, we also used the SDS-PAGE assay to further validate the results of the ThT assay to determine the antagonist activity of the dipyrindyls. We used six dipyrindyls with varying antagonist activity (from ThT assay) for the SDS-PAGE assay. In the absence of dipyrindyls, ~17% of α S was found in the soluble fraction and ~83% of α S was found in the insoluble fraction (Supplementary Fig. 5a-e). For the dipyrindyls, the amount of insoluble fraction of α S (from SDS-PAGE, Supplementary Fig. 5b-e) was in close agreement with the ThT intensity (from ThT assay, Supplementary Fig. 5a). We observed that the higher ThT intensity in the presence of dipyrindyls correlated with a higher amount of α S in the insoluble fraction. For example, in the presence of NS55 (most potent antagonist), the amount of α S the insoluble fraction (26%) and the ThT intensity (4%) were the lowest among the dipyrindyls (Fig. 2e-h, Supplementary Fig. 5a-e). On the contrary, in the presence of NS119 (a poor antagonist), the amount of α S in the insoluble fraction (83%) and the ThT intensity (>100%) were the highest among the dipyrindyls (Fig. 2e, Supplementary Fig. 5a-e). Clearly, we demonstrated that the antagonist activity of dipyrindyls determined from ThT assay was in close agreement with the SDS-PAGE assay.”

Supplementary Fig. 7. The graphical representation of the ThT intensity (a), SDS-PAGE gel analysis (b), and SDS-PAGE gels (c,d) of 100 μ M α S aggregation for four days in the absence and presence of tripyridyls at an equimolar ratio.

“Similar to monopyridyls and dipyridyls, we used the SDS-PAGE assay to further validate the antagonist activity of tripyridyls determined from the ThT assay. We used multiple tripyridyls with varying antagonist activity for the SDS-PAGE assay. Again, we observed that the higher the amount of the insoluble fraction of α S (gel shift assay) in the presence of tripyridyls, the higher their respective ThT intensities. For tripyridyls, the amount of insoluble α S (from SDS-PAGE, Supplementary Fig. 7b-d) was in close agreement with the ThT intensity (from ThT assay, Fig. 2k, Supplementary Fig. 7a). For example, in the presence of NS132 (potent antagonist), the amount of α S in the insoluble fraction (14%) and the ThT intensity (1%) were the lowest among the tripyridyls (Fig. 2k,m,n,q and Supplementary Fig. 7b-d). On the contrary, in the presence of NS169 (poor antagonist), the amount of insoluble α S (88%) and the ThT intensity (98%) were the highest among the tripyridyls (Fig. 2k and Supplementary Fig. 7b-d). Clearly, we demonstrated that the antagonist activity of tripyridyls determined from ThT assay was in close agreement with the SDS-PAGE assay.”

*In addition to the SDS-PAGE assay, we used a cell-based assay to test the antagonist ability of multiple tripyridyls with varying antagonist activity against α S aggregation in a cellular context. We picked four tripyridyls (NS127, NS131, NS132, and NS163) with varying antagonist activity and cell permeability for the cell-based assay. We determined the cell permeability of the tripyridyls using the PAMPA and also compared the antagonist activity of these tripyridyls to rescue intracellular α S aggregation in a recently developed HEK cell based assay in our lab²⁴. The detailed experimental procedure has been added to the materials and methods and the results have been included in the main manuscript (**Fig. 4a**) and the supplementary information (**Supplementary Fig. 18a-c**). The explanation has been added in the main text and it reads as below:*

Supplementary Fig. 18. a, Assessment of cell permeability of the indicated ligands using the PAMPA. Confocal images (**b**) and statistical analysis (**c**) of HEK cells treated with the aggregated solution of 5 μM αS , followed by the treatment with the indicated ligands (10 μM). Inclusions of $\alpha\text{S}_{\text{A53T-YFP}}$ = white arrows, Hoechst = blue, $\alpha\text{S-ps-129}$ = red, merge = Hoechst, $\alpha\text{S-ps-129}$, and $\alpha\text{S}_{\text{A53T-YFP}}$. The PAMPA assays were conducted three independent times with three technical replicates. The Proteostat assays were conducted with at least four biological replicates and three technical replicates for each biological replicate. The data were expressed as mean and the error bars report the s.e.m. ($n = 3\text{-}4$ independent experiments and each n consisted of three technical replicates). The statistical analysis was performed using ANOVA with Tukey's multiple comparison test. * $p < 0.05$, ** $p < 0.01$, *** $p < 0.001$, **** $p < 0.0001$.

“We also compared other tripyridyls with NS163 for their ability to inhibit intracellular αS aggregation using this assay. We used multiple tripyridyls (NS127, NS131, NS132, and NS163) with varying antagonist ability to inhibit αS aggregation. To test their antagonist activity against intracellular αS aggregation, we first determined the cell permeability of the tripyridyls using the PAMPA (Fig. 4a, Supplementary Fig. 18a). The increasing order of the cell permeability was $\text{NS131} < \text{NS127} < \text{NS132} < \text{NS163}$ (Fig. 4a, Supplementary Fig. 18a). The cell permeability of NS131 was the lowest, most likely because of the sulfonic acid side chain. The cell permeability of NS127 and NS132 was very similar because of the same COOH group and hydrophobic groups (Fig. 4a, Supplementary Fig. 18a). The cell permeability of NS163 was highest because the COOH group was replaced with a hydroxylamine group (Fig. 4a). We utilized the HEK293 cells-based assay as we used for NS163 and NS132 to determine the

antagonist activity of other tripyridyls against intracellular α S aggregation. To test the tripyridyls, we first incubated the HEK cells with preformed α S fibers (5 μ M, monomer concentration) for 8 h. Subsequently, we added tripyridyls (10 μ M) to the HEK cells and then we incubated the cells for additional 18 h, which is the total time (24 h) required for the complete maturation of α S inclusions in the HEK cells. We used confocal imaging and the ProteoStat dye-based assay to monitor the effect of tripyridyls on intracellular α S inclusions (Supplementary Fig. 18a-c). We observed a weak antagonist effect of NS131 (8% decrease in inclusions) on the intracellular α S aggregation even though it was a good antagonist of α S aggregation (from ThT, Fig. 2k, Supplementary Fig. 18b,c). The poor intracellular antagonist activity of NS131 is most likely because of very poor cell permeability (Supplementary Fig. 18a). We observed a noticeable inhibition (~20% decrease in inclusions) of the intracellular aggregation of α S in the presence of NS127 (Supplementary Fig. 18b,c). NS132 (47% decrease in inclusions) was a much better antagonist than NS127 of the intracellular α S aggregation (Fig. 4g,h, Supplementary Fig. 18b-c). It is worth noting that the cell permeability of NS127 was comparable to NS132 (Fig. 4a, Supplementary Fig. 18a). The ThT assay also validated these results that NS132 is a far more potent antagonist of α S aggregation (Fig. 2k). NS163 was the most potent antagonist of the intracellular α S aggregation (76% decrease in inclusions) among the tripyridyls (Fig. 4g,h, Supplementary Fig. 18b-c). Both NS163 and NS132 were comparable in the inhibition of α S aggregation from ThT and gel shift assays; however, the cell permeability of NS163 is better than NS132 (Fig. 4a). These results demonstrate that higher cell permeability and antagonist activity together allow for better antagonist activity against intracellular aggregation of α S.”

Moreover, there have been many reports of compounds that inhibit aSyn aggregation. It has also not been verified whether the hit compounds obtained by the authors' strategy are superior to existing compounds.

Response:

We agree with the reviewer that we did not compare the antagonist activity of OPs with the other compounds reported in the literature. In the revised manuscript, we compared the antagonist activity of OPs with multiple ligands, which were reported in literature (and commercially available) using both in vitro (ThT aggregation assay) and in vivo (UA196 worms) assays. Both OPs, NS132 and NS163 were a lot better antagonists of α S aggregation at an equimolar ratio under our conditions for the ThT aggregation assay. Also, we compared the antagonist activity of these ligands in rescuing the degeneration of DA neurons in UA196 worms at 50 μ M, similar concentration to NS132 and NS163. Under these conditions, NS163 (then NS132) was the most potent ligand in rescuing the degeneration of DA neurons in UA196 worms. Both assays demonstrate that our OPs (NS163 and NS132) were far better antagonists of α S aggregation than the reported ligands in literature, both in vitro and in vivo assays. The detailed experimental procedure has been added to the materials and methods and the results have been included in the main manuscript (Fig. 4f, 5l, 6l) and the supplementary information (Supplementary Fig. 27, 36). The explanation has been added in the main text and it reads as below:

Fig. 4f. The comparison of the aggregation profile of 100 μM αS in the absence and presence of the indicated ligands at an equimolar ratio. The ThT experiments were conducted three times and the reported change in the ThT intensity was an average of three independent experiments and the error bars report the s.d.

Supplementary Fig. 17. The graphical representation of the ThT intensity of 100 μM αS aggregation for four days in the absence and presence of the indicated ligands at an equimolar ratio. The aggregation assays were conducted three times and the reported change in the ThT intensity was an average of three independent experiments. The data were expressed as mean and the error bars report the s.d. (n = 3 independent experiments).

“We also compared the antagonist activity of NS163 and NS132 with various reported ligands using ThT aggregation assay. We used multiple reported ligand inhibitors of αS aggregation, which were commercially available, including Bexarotene⁶⁴, Tyrosol⁶⁵, Valporic acid⁶⁶, and EGCG⁶⁷. The ligands were screened against 100 μM αS aggregation (for four days) at an equimolar ratio using ThT aggregation assay. We identified that both NS163 and NS132 were the most potent antagonist of αS aggregation as they completely suppressed the ThT

intensity of the α S aggregation (Fig. 4f, Supplementary Fig. 17). Most of these ligands were not very effective antagonists of α S aggregation at an equimolar ratio, as reported by others as well. EGCG was the only ligand which demonstrated a moderate inhibition of α S aggregation as it decreased the overall ThT intensity by \sim 25% after 96 h. Clearly, NS163 and NS132 are far more potent antagonists than the reported antagonists of α S aggregation..”

Fig. 5l and Supplementary Fig. 27. Neuroprotective effect of various ligands on the degeneration of DA neurons. Statistics for the total number of neurons in UA196 worms on the indicated days and the effect of the indicated ligands (50 μ M, treatment on day two and four) on the DA neurons on day 15. The number of DA neurons were determined using the confocal imaging. For each confocal imaging experiment, 10 worms were used, and the healthy neurons were counted manually, and each condition (day) consisted of six independent experiments with freshly bleached worms. The data were expressed as mean and the error bars report the s.e.m. (n = 6 independent experiments and each experiment consisted of 10 technical replicates). The statistical analysis was performed using ANOVA with Tukey’s multiple comparison test. *p < 0.05, **p < 0.01, ***p < 0.001, ****p < 0.0001.

“We compared NS163 and NS132 with other reported ligand inhibitors of α S aggregation in the literature for their ability to rescue the degeneration of DA neurons in UA196 worms under matched conditions (50 μ M concentration). The number of intact DA neurons in UA196 worms on day 15 in the presence of Bexarotene, Tyrosol, Valporic acid, and EGCG was 19.3, 21.7, 18.7, and 30.5, respectively (Fig. 5l, Supplementary Fig. 27). Most of the ligands were ineffective in rescuing the degeneration of DA neurons in UA196 worms (Fig. 5l, Supplementary Fig. 27). EGCG was moderate in rescuing the degeneration of DA neurons (Fig. 5l, Supplementary Fig. 27). Our results corroborate well with the published data as some of these molecules demonstrate noticeable rescue effect at one mM concentration⁶⁴⁻⁶⁶. Under matched conditions, in the presence of NS132 (50 μ M), the total number of intact DA neurons on day 15 was 51, which confirms that NS132 also

effectively rescues the degeneration of DA neurons in UA196 worms (Supplementary Fig. 27, 28a-e). The neuroprotective effect of NS163 was better than NS132, indicated by a higher number of intact healthy neurons (Fig. 5k, Supplementary Fig. 28a-e), most likely due to the former's better ability to permeate the cell membrane."

Fig. 6l and Supplementary Fig. 36. Neuroprotective effect of various ligands on the degeneration of DA neurons in a post-disease PD model of UA196 worms. Statistics for the total number of intact DA neurons in UA196 worms in the absence and presence of the indicated ligands (at 50 μ M, day 5) on day 15. For each confocal imaging experiment at least 10 worms were used, and the healthy neurons were counted manually, and each condition (day) consisted of six independent experiments. The data were expressed as mean and the error bars report the s.e.m. ($n = 6$ independent experiments and each n consisted of 10 technical replicates). The statistical analysis was performed using ANOVA with Tukey's multiple comparison test. * $p < 0.05$, ** $p < 0.01$, *** $p < 0.001$, **** $p < 0.0001$.

"We also compared the antagonist activity of NS163 and NS132 with other reported inhibitors of α S aggregation for their ability to rescue the degeneration of DA neurons in UA196 worms in the late-stage onset PD model. The ligands (50 μ M) were added on day five and the number of intact DA neurons was counted on day 15. The number of intact DA neurons decreased from 58.7 (day 3) to 12.8 (day 15). At this dose, the number of intact DA neurons in UA196 worms on day 15 in the presence of NS163, Bexarotene, Tyrosol, Valporic acid, and EGCG was 41.3, 13.2, 14.3, 13.5, and 20.5, respectively (Fig. 6l). All other ligands except EGCG were not able to rescue the degeneration of DA neurons in UA196 worms (Fig. 6l). EGCG was moderate in rescuing the degeneration of DA neurons. NS132 was also effective (less than NS163) in rescuing degeneration of DA neurons in a post-disease onset PD model, characterized by confocal imaging and the number of healthy neurons (Supplementary Fig. 35a-f, 36). The number of intact DA neurons in UA196 worms on day 15 in the presence of NS132 was 32, which

indicates that it was a far better ligand than other reported ligands from literature in rescuing the degeneration of DA neurons (Supplementary Fig. 35a-f, 36). Overall, NS163 was the most potent ligand in rescuing degeneration of DA neurons in the post-disease onset PD model.”

In conclusion, our molecules (NS132 and NS163) were far better antagonists in rescuing α S aggregation (and PD phenotypes) in both in vitro and in vivo PD models.

Furthermore, although they have acquired numerous data using a nematode model system, there are concerns about their cultural methods and substantial questions about some of the data.

Response:

We have addressed the concerns raised by the reviewer in the specific major points below. We have revised the culture methods in detail and these methods have been included in the supplementary methods section.

Major points:

1. The authors started the evaluation of dipyridylamides with carboxylic acid functional group in the first side chain considering previous reports, and then they screened the second and third side chains by their capability of inhibiting the aggregation using the ThT assay. Although they synthesized and screened several compounds, the OPs seem to have been created randomly in a blinded manner, without a specific strategy and without, apparently, considering the specific structural characteristics of aSyn. If there is a strategy for the compound deployment, it should be described clearly.

Also, there is missing a deeper and rational discussion of the results that explains the relation between the structural properties of the side groups of the compounds with good activity against the fibril formation. For example, considering the compound NS132, there is a clear tendency, from the first to the third pyridyl side chains, to show an increase in the size of the side group. Is this tendency observed in the other compounds synthesized in this study? How do the authors explain the interaction between NS132, that possesses several hydrophobic moieties, with the regions in the N-terminus domain, positively charged, of aSyn?

In addition, it would be good to discuss from the result of ThT which kind of side chains of the second and third pyridyls effectively suppress the aggregation of aSyn. In other words, it might be useful to speculate the structure-efficacy correlations.

Response:

We apologize for not making it clear about the strategy to design and synthesize this library to screen against α S aggregation. We have multiple rationale for the design of OPs.

We have shown recently that a foldamer was very potent antagonist of α S aggregation and the antagonist activity of this foldamer was predominantly dependent on a negatively charged group (COOH) and an isopropyl (hydrophobic) group²⁴. We have also shown in that work that the foldamer interacts with the N-terminus

of α S, which is predominantly positively charged because of many lysine residues. Also, it has been suggested that the N-terminus of α S is important in facilitating the aggregation. Additionally, we have also shown that the

OPs with the COOH functional group are very effective inhibitors of aggregation of other amyloid proteins that contain lysine amino acid via the formation of salt bridge. Therefore, we chose the COOH as the first side chain on the OPs to interact predominantly with the lysine residues predominantly found on the N-terminus of α S. To further support our choice of using the COOH functional group as first side chain on the monopyridyl, we synthesized a library of monopyridyls with different functional groups (including COOH functional group). We tested all the monopyridyls against the aggregation of 100 μ M α S using the ThT aggregation assay at an equimolar ratio (in 1 \times PBS buffer, pH 6.5). The ThT assay identified NS41 as the most potent antagonist of α S aggregation as it reduced the ThT signal to 29% and 17% at binding stoichiometries of 1:1 and 2:1 (NS41/ α S), respectively. NS41 contains COOH functional group as a side chain. The detailed experimental procedure has been added to the materials and methods and the results have been included in the main manuscript (Fig. 2a-d) and the supplementary information (Supplementary Fig. 3, 4a,b). The explanation has been added in the main text and it reads as below:

“We used a novel 2D FAST approach in OPs to identify potent antagonists of the aggregation of α S. In this approach, we start with a library of monopyridyls with different functional groups. The library of monopyridyls was synthesized using 2-chloro-6-methyl-3-nitropyridine as a common precursor, which was treated with primary alcohols with diverse side chain functionalities via a one-pot reaction (Fig. 1B, a and see Supplementary information for synthetic details). The synthesis of the monopyridyls did not require any column chromatography as the pure monopyridyls were extracted via acid/base treatment or via lyophilization (see Supplementary information for synthetic details). We selected the side chains containing hydrophobic, polar, positively charged, and negatively charged functional groups, which mimic the side chains of the amino acids (Fig. 1C). We could not use a small selection of the monopyridyls for assays because of their poor solubility in solution conditions due to the side chains (in 1 \times PBS buffer, pH 6.5). The monopyridyl library was screened against the aggregation of 100 μ M α S at an equimolar ratio using Thioflavin T (ThT) dye-based aggregation assay (Fig. 1B, b). The most potent monopyridyl antagonist (OP1) of α S aggregation was used as the precursor for the synthesis of a chlorodipyridyl using a newly developed chromatography-free amide coupling in our lab (Fig. 1B, OP2). The chlorodipyridyl was reacted with a library of primary amines/thiols to synthesize a library of dipyrindyls (Fig 1B, e) using a one-pot reaction. All reactions went to completion and a large number of dipyrindyl products did not require column chromatography as the excess primary amines/thiols were evaporated on rotovap or lyophilizer. However, a few dipyrindyls required column chromatography to separate them from the starting material side chains because of their very high boiling point (7 out of 21 dipyrindyls required column, Supplementary Fig. 1). The dipyrindyls were screened against the aggregation of 100 μ M α S at an equimolar ratio using ThT assay (Fig. 1B, f), which identified the most potent dipyrindyl antagonist (OP3) of α S aggregation. Similarly, we synthesized a library of tripyridyls, screened, and identified the most potent tripyridyl against the aggregation of 100 μ M α S at an equimolar ratio (Fig. 1B, g-j).

Biophysical characterization of OPs against the aggregation of α S. The screening of the monopyridyls against 100 μ M α S aggregation (1 \times PBS buffer, pH 6.5) identified NS41 as the most potent antagonist as it reduced the

ThT signal to 29% and 17% at molar stoichiometries of 1:1 and 2:1 (NS41/ α S), respectively (Fig 2a-c). The inhibition of α S aggregation by NS41 was also confirmed by TEM images, which show an abundance (Supplementary Fig. 3a) and low amount (Supplementary Fig. 3b) of α S fibers in the absence and presence of NS41, respectively. We used SDS-PAGE (Sodium dodecyl sulphate polyacrylamide gel electrophoresis) as a complementary assay to validate ThT results and to further characterize the antagonist activity of the monopyridyls against α S aggregation. We used a total of four monopyridyls for the SDS-PAGE assay with varying antagonist activity against α S aggregation. For the SDS-PAGE assay, a solution of 100 μ M α S was aggregated for four days in the aggregation buffer (1 \times PBS buffer, pH 6.5) in the absence and presence of the monopyridyls at an equimolar ratio. Subsequently, the α S solutions were centrifuged to separate α S aggregates from the soluble α S. Afterward, the samples were boiled at 95 $^{\circ}$ C for 5 min to disassemble α S aggregates and examined using the SDS-PAGE assay (Fig. 2d, Supplementary Fig. 4a). The band intensities of the gels in SDS-PAGE assay were quantified using ImageJ software. In the absence of ligands, \sim 12% of α S was found in the soluble fraction and the rest was found in the insoluble fraction (Fig. 2d, Supplementary Fig. 4a,b). In the presence of NS41 at molar ratios of 1:1 and 2:1 (NS41: α S), 48% and 73% of α S protein were found in the soluble fraction, respectively, and the rest of α S protein was found in the insoluble fraction (Fig. 2d, Supplementary Fig. 4a,b). The most potent ligand from ThT assay (NS41) also demonstrated the highest amount of α S in the soluble fraction. Similarly, the least effective monopyridyl ligand (RD247) had the highest ThT intensity and demonstrated the highest amount of α S (87%) in the insoluble fraction, a value very close to the untreated α S protein (87% insoluble, Fig. 2d, Supplementary Fig. 4a,b). The antagonist activity of the monopyridyls corroborated well from both ThT and the SDS-PAGE assays (Fig. 2a-d, Supplementary Fig. 4a,b). The higher the antagonist activity of monopyridyls against α S aggregation, the lower the ThT intensity and the higher the amount of α S remained in the soluble fraction (Fig. 2a-d, Supplementary Fig. 4a,b). Together, NS41 was identified as the most potent monopyridyl antagonist of α S aggregation. It is important to note that NS41 contains a carboxyl (COOH) functional group side chain. We anticipated that the monopyridyl with the COOH functional group would be the most potent antagonist, because we have recently shown that a foldamer was a potent antagonist of α S aggregation and its antagonist activity predominantly relied on a negatively charged (COOH) side chain. We have also shown in that work that the foldamer binds to the N-terminus

of α S because of its negatively charged COOH functional group interaction with the positively charged lysine residues of α S. Also, it has been suggested that the N-terminus of α S is important in facilitating its aggregation. Therefore, ligands that interact with the N-terminus of α S will likely inhibit the aggregation of α S. Additionally, we have also shown that the OPs with the COOH functional group are very effective inhibitors of aggregation of other amyloid proteins that contain lysine amino acid via the formation of salt bridges³⁰⁻³⁴.”

For dipyridyls and tripyridyls, we have included an explanation detailing the structure activity relationship (SAR) analysis of the antagonist activity of ligands (dipyridyls and tripyridyls) against α S aggregation. The data has been included for dipyridyls (Fig. 2e) and tripyridyls (Fig. 2k) and the SAR explanation has been added to the main manuscript, which reads as follows:

SAR study for dipyridyls:

“We compared the antagonist activity of dipyridyls using the ThT assay (and gel shift assay) to carry out the SAR study between dipyridyls and α S. We observed various patterns between the antagonist activity and the chemical structure of the side chains on the second position of dipyridyls (Fig. 2e). We first compared the hydrophobicity of the side chains of the dipyridyls. For the most part, the antagonist activity of the dipyridyls was directly related to the hydrophobicity of the side chains. The propyl side chain has the lowest antagonist activity and the antagonist activity for the most part was increased with the increase in the hydrophobicity of the side chains. The cyclohexyl group had the highest antagonist activity (NS55, Fig. 2e-h). The antagonist activity decreased with a higher hydrophobicity than cyclohexyl group, as demonstrated by the indole group (NS72, Fig. 2e). We also observed that the side chains with amines (aliphatic or aromatic) were detrimental to the antagonist activity against α S aggregation. The polar non-aromatic hydroxyl groups as side chains were effective antagonists of α S aggregation; however, the phenol group as a side chain was a moderate antagonist of α S aggregation. The negatively charged (carboxylic and sulphonic acid) side chains were very effective inhibitors of α S aggregation. The effective antagonist activity of the hydrophobic, negatively charged, and primary hydroxyl groups is because these dipyridyls might be interacting with different regions of α S and inhibiting the aggregation. We will further explore these different regions of α S that are important for the aggregation and that study will be part of a separate manuscript. For the current study, we chose the cyclohexyl group as the second side chain to synthesize the tripyridyls because it demonstrated the highest antagonist activity against α S aggregation (Fig. 2e-h). We envision that the negatively charged (COOH group) and hydrophobic (cyclohexyl) side chains on the dimer interact with the positively charged (lysine) and hydrophobic groups on the N-terminus of α S.”

SAR study for tripyridyls:

“We compared the antagonist activity of tripyridyls using the ThT assay (and gel shift assay) to carry out a SAR study between tripyridyls and α S. We observed various patterns between the antagonist activity and the chemical structure of the side chains on the third position of tripyridyls (Fig. 2k). We first compared the antagonist activity of the hydrophobicity of the side chains of the tripyridyls. The tripyridyl with propyl side chain (NS123) demonstrated the lowest antagonist activity as it decreased the ThT signal from 100% to 72.7% (Fig. 2k). The tripyridyls with more hydrophobic groups, including NS127, NS126, and NS165 demonstrated moderate antagonist activity against α S aggregation as they decreased the ThT signal to 50.3%, 45.9%, 43.9%, respectively. The tripyridyl with the indole group (NS132) was the most potent antagonist as it decreased the ThT signal to 1%

(Fig. 2k). The antagonist activity, for the most part, was increased with an increase in the hydrophobicity of the side chains in the tripyridyls except the trimer with an alkyne side chain (NS174), which decreased the ThT signal to 17.9% (Fig. 2k). One of the reasons could be that NS174 might be interacting with a different α S region than NS132 and inhibiting α S aggregation. It has been shown recently by us and others that there are multiple α S sequences that facilitate α S aggregation. We also observed that the tripyridyls with side chains as amines (aliphatic or aromatic) did not demonstrate any noticeable effect on α S aggregation, a pattern similar to the dipyridyls (Fig. 2e,k). The tripyridyls with carboxylic (NS130) and sulphonic (NS131) acids were moderate antagonists of α S aggregation as they decreased the ThT signal to 59.6% and 36.1%, respectively (Fig. 2k). The tripyridyls with aliphatic hydroxyl (NS166) and phenol (NS168) groups were potent antagonists of α S aggregation as they decreased the ThT signal to 15.5% and 2.7%, respectively (Fig. 2k). The tripyridyl, NS168 was a potent antagonist of α S aggregation as it decreased the ThT signal to 2.5%, close to NS132 (Fig. 2k). We speculate that NS168 might be interacting with a different sequence of α S than NS132. We will investigate the interaction of these tripyridyls with α S in detail and it will be presented in the near future.”

“From the 2D HSQC NMR, we have identified two potential binding sites of NS132 on α S. NS132 contains a negatively charged side chain (COOH) and two hydrophobic side chains (cyclohexyl and indole). For the first binding site (from 10-20 residues of α S), we posit that the COOH and the two hydrophobic functional groups of NS132 likely interact with the lysine (K12) and the hydrophobic residues patch (14-16) of α S. Similarly, for the second binding site of NS132 on α S, we envision that the side chain functional groups of NS132 interact with the lysine (K43) and the hydrophobic residues patch (36-40) of α S. We have shown these interactions using 2D-NMR, where the overall volume intensities of these residues of α S were affected in the presence of NS132.”

2. The authors indicate that “the data suggest that NS132 interacts with α S and generates off-pathway fiber-incompetent structures”. From the experimental data of the Fig 3, we can only see the absence of fibrillar species. How the authors define the term “off-pathway structures”? Can the authors actually demonstrate the existence of these structures?

Response:

We apologize for the confusion. We meant that NS132 interacts with α S and generates fiber-incompetent structures, which are not able to template and convert the monomeric α S into fibers. To demonstrate that NS132 forms structures that are fiber-incompetent, we took the TEM images of the solution from the 5th cycle of the PMCA experiment of α S in the absence and presence of NS132 at an equimolar ratio (Supplementary Fig. 13a,b). We observed that there was an abundance of fibers in the solution from the 5th cycle of the PMCA experiment of α S (Supplementary Fig. 13a). In marked contrast, we observed a very small number of fibers from the 5th cycle of the PMCA experiment of α S in the presence of NS132 at an equimolar ratio (Supplementary Fig. 13b). The morphology of the fibers was different in the absence and presence of NS132 (Supplementary Fig. 13a,b). The α S fibers were amorphous in nature for the PMCA sample (5th cycle) of α S in the presence of NS132. We have also shown that α S treated with NS132 did not template α S monomer into fibers from the 1st cycle to the 5th cycle (Fig. 3f-h). Together, we surmise that the structure of α S in the presence of NS132 were fiber-incompetent structures.

We have added the TEM images (Supplementary Fig. 13a,b) and modified the language in the main manuscript, which now reads as below:

Supplementary Fig 13. The TEM images of the solution from the 5th cycle of the PMCA experiment of α S in the absence (a) and presence (b) of NS132 at an equimolar ratio in $1 \times$ PBS buffer (pH 6.5).

“We also used TEM images to validate the ThT results and characterize the morphology of α S fibers in the absence and presence of NS132. We took TEM images of the solution from the 5th cycle of the PMCA experiment of α S in the absence and presence of NS132 at an equimolar ratio (Supplementary Fig. 13a,b). We observed that there was an abundance of fibers in the solution from the 5th cycle of the PMCA experiment of α S (Supplementary Fig. 13a). In marked contrast, we observed a very small number of fibers from the 5th cycle of the PMCA experiment of α S in the presence of NS132 at an equimolar ratio (Supplementary Fig. 13b). The morphology of the fibers was amorphous in nature for the PMCA sample (5th cycle) of α S in the presence of NS132 (Supplementary Fig. 13b). We have also shown that α S treated with NS132 did not template α S monomer into fibers from the 1st cycle through the 5th cycle. Together, the data suggest that NS132 interacts with α S and generates fiber-incompetent structures in through all 5 cycles of the PMCA.”

3. In the Figs 2p and 3i-l, could the authors discuss why in one case they focus on the chemical shift perturbation (Fig. 2p), but on peak intensities in Fig 3i-l? Also, the manuscript would benefit from showing the data of peak intensities and chemical shift perturbation of the NMR spectra as supplementary material.

Response:

We apologize for the confusion. We reported the volume of the peak intensities in both Fig. 2p and Fig. 3i-l. We have now fixed the legends of the text for Fig. 2 and reported the change in the volume of the peak intensities for Fig. 2p (Now Fig. 2u) as well. We have now included the volume intensities of the amide peaks of α S in the absence and presence of NS132 (Fig. 2u) and we have included a writeup in the main manuscript to explain the volume intensity changes, which now reads as follows:

Fig. 2u, Graphical presentation of the relative volume changes of the backbone amide residue peaks of ¹⁵N-labeled αS (70 μM) in the presence of NS132 at an equimolar ratio. The colored sequences (blue and green) are the potential binding sites of NS132 on αS. The dashed line represents the reported volume changes in ¹⁵N-labeled αS residues peaks in the presence of NS132 above the value of 5%.

“We collected the HSQC NMR of 70 μM ¹⁵N-¹H-uniformly labeled αS in the absence (Supplementary Fig. 11, red) and presence of NS132 (Fig. 2u, Supplementary Fig. 11, blue) and compared the signal intensity (volume) of the amide peaks. In the presence of NS132, we observed noticeable volume changes in the amide peaks for specific residues toward the N-terminus, indicative of the interaction and binding site of NS132 on αS, especially continuous residue sequences from 10-20 and 36-43 (Fig. 2u, blue and green). NS132 also moderately or weakly interacted with a few distinct residues on the N-terminus of αS (up to 100 residues), represented by small changes in the peak volumes (Fig. 2u). These might be weak secondary binding sites of NS132 on αS.”

4. The screening is limited to the *in vitro* thioflavin T assay. Certainly, ThT is useful as a first screening, but a major issue is that the degree of reduction in ThT does not always correspond with the effect in cells or *in vivo* situations.

Response:

We used the ThT assay as a high throughput approach to identify potent antagonists of αS aggregation. For screening of library of ligands, ThT dye-based aggregation has been traditionally used as a high throughput screening method by various labs for amyloid proteins. As suggested by the reviewer, we have now used two additional assays, including a SDS PAGE gel shift assay and a HEK cell-based assay to assess the antagonist activity of our ligands in the biological context.

We have included a secondary screening method to validate the results of the ThT screening method. We have used a sodium dodecyl sulphate–polyacrylamide gel electrophoresis (SDS-PAGE) assay in the absence and

presence of the monopyridyls, dipyridyls, and tripyridyls. In the SDS-PAGE assay, we used multiple monopyridyls, dipyridyls, and tripyridyls with varying antagonist activity (from ThT assay) and we used exactly identical solution conditions of the ThT assay. The soluble and insoluble fractions of α S were separated in the presence of monopyridyls, dipyridyls, or tripyridyls and quantified using SDS-PAGE assay. The antagonist activity of the monopyridyls, dipyridyls, and tripyridyls determined using the SDS-PAGE assay corroborates well with the ThT screening assay. The detailed experimental procedure has been added to the materials and methods and the results have been included in the main manuscript and the supplementary information for monopyridyls (Fig. 2a-d, Supplementary Fig. 4a), dipyridyls (Fig. 2e-h, Supplementary Fig. 5a-e) and for tripyridyls (Fig. 2k-q and Supplementary Fig. 7a-d). The explanation has been added in the main text and it reads as below:

Supplementary Fig. 4. The graphical representation of the ThT intensity of 100 μ M α S aggregation for four days in the absence and presence of NS55 (a) and NS132 (b) in 1 \times PBS at substoichiometric ratio (for NS55, α S:ligand, 1:0.5, and for NS132, α S:ligand, 1:0.1).

“We used SDS-PAGE (Sodium dodecyl sulphate polyacrylamide gel electrophoresis) as a complementary assay to validate ThT results and to further characterize the antagonist activity of the monopyridyls against α S aggregation. We used a total of four monopyridyls for the SDS-PAGE assay with varying antagonist activity against α S aggregation. For the SDS-PAGE assay, a solution of 100 μ M α S was aggregated for four days in the

aggregation buffer (1 × PBS buffer, pH 6.5) in the absence and presence of the monopyridyls at an equimolar ratio. Subsequently, the α S solutions were centrifuged to separate α S aggregates from the soluble α S. Afterward, the samples were boiled at 95 °C for 5 min to disassemble α S aggregates and examined using the SDS-PAGE assay (Fig. 2d, Supplementary Fig. 4a). The band intensities of the gels in SDS-PAGE assay were quantified using ImageJ software. In the absence of ligands, ~12% of α S was found in the soluble fraction and the rest was found in the insoluble fraction (Fig. 2d, Supplementary Fig. 4a,b). In the presence of NS41 at molar ratios of 1:1 and 2:1 (NS41: α S), 48% and 73% of α S protein were found in the soluble fraction, respectively, and the rest of α S protein was found in the insoluble fraction (Fig. 2d, Supplementary Fig. 4a,b). The most potent ligand from ThT assay (NS41) also demonstrated the highest amount of α S in the soluble fraction. Similarly, the least effective monopyridyl ligand (RD247) had the highest ThT intensity and demonstrated the highest amount of α S (87%) in the insoluble fraction, a value very close to the untreated α S protein (87% insoluble, Fig. 2d, Supplementary Fig. 4a,b). The antagonist activity of the monopyridyls corroborated well from both ThT and the SDS-PAGE assays (Fig. 2a-d, Supplementary Fig. 4a,b). The higher the antagonist activity of monopyridyls against α S aggregation, the lower the ThT intensity and the higher the amount of α S remained in the soluble fraction (Fig. 2a-d, Supplementary Fig. 4a,b). Together, NS41 was identified as the most potent monopyridyl antagonist of α S aggregation”

Supplementary Fig. 5. The graphical representation of the ThT intensity (**a**), SDS-PAGE gel analysis (**b**), and SDS-PAGE gels (**c-e**) of 100 μ M α S aggregation for four days in the absence and presence of dipyrindyls at an equimolar ratio.

“Similar to monopyridyls, we also used the SDS-PAGE assay to further validate the results of the ThT assay to determine the antagonist activity of the dipyrindyls. We used six dipyrindyls with varying antagonist activity (from ThT assay) for the SDS-PAGE assay. In the absence of dipyrindyls, ~17% of α S was found in the soluble fraction and ~83% of α S was found in the insoluble fraction (Supplementary Fig. 5a-e). For the dipyrindyls, the amount of insoluble fraction of α S (from SDS-PAGE, Supplementary Fig. 5b-e) was in close agreement with the ThT intensity (from ThT assay, Supplementary Fig. 5a). We observed that the higher ThT intensity in the presence of dipyrindyls correlated with a higher amount of α S in the insoluble fraction. For example, in the presence of NS55 (most potent antagonist), the amount of α S the insoluble fraction (26%) and the ThT intensity (4%) were the lowest among the dipyrindyls (Fig. 2e-h, Supplementary Fig. 5a-e). On the contrary, in the presence of NS119 (a poor antagonist), the amount of α S in the insoluble fraction (83%) and the ThT intensity (>100%) were the highest

among the dipyridyls (Fig. 2e, Supplementary Fig. 5a-e). Clearly, we demonstrated that the antagonist activity of dipyridyls determined from ThT assay was in close agreement with the SDS-PAGE assay.”

Supplementary Fig. 7. The graphical representation of the ThT intensity (a), SDS-PAGE gel analysis (b), and SDS-PAGE gels (c,d) of 100 μM αS aggregation for four days in the absence and presence of tripyridyls at an equimolar ratio.

“Similar to monopyridyls and dipyridyls, we used the SDS-PAGE assay to further validate the antagonist activity of tripyridyls determined from the ThT assay. We used multiple tripyridyls with varying antagonist activity for the SDS-PAGE assay. Again, we observed that the higher the amount of the insoluble fraction of αS (gel shift assay) in the presence of tripyridyls, the higher their respective ThT intensities. For tripyridyls, the amount of insoluble αS (from SDS-PAGE, Supplementary Fig. 7b-d) was in close agreement with the ThT intensity (from ThT assay, Fig. 2k, Supplementary Fig. 7a). For example, in the presence of NS132 (potent antagonist), the amount of αS in the insoluble fraction (14%) and the ThT intensity (1%) were the lowest among the tripyridyls (Fig. 2k,m,n,q and Supplementary Fig. 7b-d). On the contrary, in the presence of NS169 (poor antagonist), the amount of insoluble αS (88%) and the ThT intensity (98%) were the highest among the tripyridyls (Fig. 2k and

Supplementary Fig. 7b-d). Clearly, we demonstrated that the antagonist activity of tripyridyls determined from ThT assay was in close agreement with the SDS-PAGE assay.”

In addition to the SDS-PAGE assay, we used a cell-based assay to test the antagonist ability of multiple tripyridyls with varying antagonist activity against α S aggregation in a cellular context. We picked four tripyridyls (NS127, NS131, NS132, and NS163) with varying antagonist activity and cell permeability for the cell-based assay. We determined the cell permeability of the tripyridyls using the PAMPA and also compared the antagonist activity of these tripyridyls to rescue intracellular α S aggregation in a recently developed HEK cell based assay in our lab²⁴. The detailed experimental procedure has been added to the materials and methods and the results have been included in the main manuscript (Fig. 4a) and the supplementary information (Supplementary Fig. 18a-c). The explanation has been added in the main text and it reads as below:

Supplementary Fig. 18. a, Assessment of cell permeability of the indicated ligands using the PAMPA. Confocal images (**b**) and statistical analysis (**c**) of HEK cells treated with the aggregated solution of 5 μ M α S, followed by the treatment with the indicated ligands (10 μ M). Inclusions of α SA53T-YFP = white arrows, Hoechst = blue, α S-ps-129 = red, merge = Hoechst, α S-ps-129, and α SA53T-YFP. The PAMPA assays were conducted three independent times with three technical replicates. The Proteostat assays were conducted with at least four biological replicates and three technical replicates for each biological replicate. The data were expressed as mean and the error bars report the s.e.m. (n = 3-4 independent experiments and each n consisted of three technical replicates). The statistical analysis was performed using ANOVA with Tukey’s multiple comparison test. *p < 0.05, **p < 0.01, ***p < 0.001, ****p < 0.0001.

“We used multiple tripyridyls (NS127, NS131, NS132, and NS163) with varying antagonist ability to inhibit α S aggregation. To test their antagonist activity against intracellular α S aggregation, we first determined the cell permeability of the tripyridyls using the PAMPA (Fig. 4a, Supplementary Fig. 18a). The increasing order of the cell permeability was NS131<NS127<NS132<NS163 (Fig. 4a, Supplementary Fig. 18a). The cell permeability of NS131 was the lowest, most likely because of the sulfonic acid side chain. The cell permeability of NS127 and NS132 was very similar because of the same COOH group and hydrophobic groups (Fig. 4a, Supplementary Fig. 18a). The cell permeability of NS163 was highest because the COOH group was replaced with a hydroxylamine group (Fig. 4a). We utilized the HEK293 cells-based assay as we used for NS163 and NS132 to determine the antagonist activity of other tripyridyls against intracellular α S aggregation. To test the tripyridyls, we first incubated the HEK cells with preformed α S fibers (5 μ M, monomer concentration) for 8 h. Subsequently, we added tripyridyls (10 μ M) to the HEK cells and then we incubated the cells for additional 18 h, which is the total time (24 h) required for the complete maturation of α S inclusions in the HEK cells. We used confocal imaging and the ProteoStat dye-based assay to monitor the effect of tripyridyls on intracellular α S inclusions (Supplementary Fig. 18a-c). We observed a weak antagonist effect of NS131 (8% decrease in inclusions) on the intracellular α S aggregation even though it was a good antagonist of α S aggregation (from ThT, Fig. 2k, Supplementary Fig. 18b,c). The poor intracellular antagonist activity of NS131 is most likely because of very poor cell permeability (Supplementary Fig. 18a). We observed a noticeable inhibition (~20% decrease in inclusions) of the intracellular aggregation of α S in the presence of NS127 (Supplementary Fig. 18b,c). NS132 (47% decrease in inclusions) was a much better antagonist than NS127 of the intracellular α S aggregation (Fig. 4g,h, Supplementary Fig. 18b-c). It is worth noting that the cell permeability of NS127 was comparable to NS132 (Fig. 4a, Supplementary Fig. 18a). The ThT assay also validated these results that NS132 is a far more potent antagonist of α S aggregation (Fig. 2k). NS163 was the most potent antagonist of the intracellular α S aggregation (76% decrease in inclusions) among the tripyridyls (Fig. 4g,h, Supplementary Fig. 18b-c). Both NS163 and NS132 were comparable in the inhibition of α S aggregation from ThT and gel shift assays; however, the cell permeability of NS163 is better than NS132 (Fig. 4a). These results demonstrate that higher cell permeability and antagonist activity together allow for better antagonist activity against intracellular aggregation of α S.”

In fact, the authors realized the importance of cell membrane permeability later and needed to modify the structure of the first pyridylamide, which was fixed at the very beginning of the screening. Changing the structure of the first pyridylamide raises suspicion throughout the library creation stage, subsequent screening, and the relevance of the hit compounds obtained.

Response:

We want to point out that the most potent antagonist (NS132) from our 2D FAST was also a potent antagonist of α S aggregation mediated PD phenotypes in two C. elegans PD models, even with low cell permeability. We have

shown that NS132 was able to rescue various PD phenotypes in two HEK cell models and two *C. elegans* PD models, including rescue of intracellular α S aggregation in HEK cells (Fig. 4g,h; Supplementary Fig. 18a-c, 19), NL5901 worms (Fig. 4i,l; Supplementary Fig. 20), rescue of motility in NL5901 (Supplementary Fig. 25), rescue of the degeneration of DA neurons in early stage (Supplementary Fig. 27, 28a-f) and the late-stage PD model of UA196 worms (Supplementary Fig. 35a-f, 36), rescue of motility in UA196 (Supplementary Fig. 29), decrease in the ROS level in UA196 worms in early (Supplementary Fig. 30) and late-stage (Supplementary Fig. 37) PD model of UA196 worms, and rescue of behavioral deficits in UA196 worms (Supplementary Fig. 32). Additionally, we compared NS132 with other reported ligands from the literature in various *in vitro* (Fig. 4f; Supplementary Fig. 17), cellular (Fig. 4g,h; Supplementary Fig. 18a-c, 19), and *in vivo* assays (Supplementary Fig. 27, 36) and identified that NS132 was a far better antagonist in inhibiting α S aggregation both *in vitro* and *in vivo* PD models. We have included this additional data in the Supplementary information and a writeup is included in the main manuscript, which reads as follows:

Fig. 4. The intracellular inhibition of α S aggregation by OPs in PD models. **a**, Assessment of cell permeability of the indicated ligands using the PAMPA. **b**, The chemical structures of NS132 and NS163. **c**, The aggregation profile of 100 μ M α S in the absence and presence of the indicated ligands at an equimolar ratio. **d**, The ITC thermogram for the titration of a solution of NS163 into α S where heat burst curves and the corrected injection heats are represented by the upper panel and the lower panel, respectively. **e**, The relative volume changes of the backbone amide peaks of 15 N-labeled α S (70 μ M) in the presence of NS163 at an equimolar ratio. The dashed line represents the reported volume changes in 15 N-labeled α S residues peaks in the presence of NS132 above the value of 5%. The colored sequences (blue and green) are the potential binding sites of NS163 on α S. **f**, The comparison of the aggregation profile of 100 μ M α S in the absence and presence of the indicated ligands at an equimolar ratio. Confocal images (**g**) and statistical analysis (**h**) of HEK cells treated with the aggregated solution of 5 μ M α S, followed by the treatment with the indicated ligands (10 μ M). Inclusions of α S_{A53T}-YFP = white arrows, Hoechst = blue, α S-ps-129 = red, merge = Hoechst, α S-ps-129, and α S_{A53T}-YFP. The representative confocal images of NL5901 worms with α S inclusions (white arrows) in the body wall muscle cells (Day 8) in the absence (**i**) and presence of 50 μ M NS132 (**j**) and 50 μ M NS163 (**k**). **l**, The number of inclusions for experiment in NL5901 in the absence (blue bar) and presence of 50 μ M NS132 (orange bar) and 50 μ M NS163 (red bar) from day four to day nine. **m**, The motility of N2 (green bar) and NL5901 and statistics (Day 8, **n**) in the absence (blue bar) and presence (red bar) of 50 μ M NS163 during the aging process. The ThT experiments were conducted three times and the reported change in the ThT intensity was an average of three independent experiments and the error bars report the s.d. For each confocal imaging experiment, at least 10 worms were used, and the inclusions were counted manually, and each condition (Each day) consisted of at least four independent experiments. For motility experiments, a total of 50 worms were used in duplicate for each experiment and each condition consisted of at least four independent experiments. The data were expressed as mean and the error bars report the s.e.m. (n = 3 or 4 independent experiments and each n consisted of a minimum of three technical replicates). The statistical analysis was performed using ANOVA with Tukey's multiple comparison test. *p < 0.05, **p < 0.01, ***p < 0.001.

“NS163 was able to reduce the intracellular inclusions of α S by 76% and 90% at 10 μ M and 25 μ M, respectively, measured using the ProteoStat dye-based assay (Fig. 4h, +NS163). NS163 inhibits the formation of α S inclusions in HEK cells in a dose dependent manner. NS132 was also able to inhibit the formation of inclusions by 47%, which is less than NS163 (Fig. 4g,h +NS132). It is likely to be a consequence of the better cell permeability of NS163 than NS132.”

“We observed a decrease in the number of inclusions in the presence of NS132 (lower effect than NS163) as the number of inclusions/worm were ~5 and ~6.6 on day eight and nine, respectively (Fig. 4j,l, orange bar, Supplementary Fig. 20). As we predicted earlier, the cell permeability of NS132 was lower than NS163, which is the likely reason for the lower effect of NS132 (Fig. 4l).”

Supplementary Fig. 25. The intracellular inhibition of α S aggregation by NS132 in an *in vivo* PD model. The comparison of the motility rate of N2 and NL5901 and statistics in the absence and presence of 50 μ M NS132 (treatment on day two and four). For motility rate experiment, a total of 50 worms were used in duplicate for each experiment and each condition consisted of four independent experiments. The data were expressed as mean and the error bars report the s.e.m. (n = 4 independent experiments and each n consisted of two technical replicates). The statistical analysis was performed using ANOVA with Tukey's multiple comparison test. *p < 0.05, **p < 0.01, ***p < 0.001.

Supplementary Fig. 27. Neuroprotective effect of various ligands on the degeneration of DA neurons.

Statistics for the total number of neurons in UA196 worms on the indicated days and the effect of the indicated ligands (50 μ M, treatment on day two and four) on the DA neurons on day 15. The number of DA neurons were determined using the confocal imaging. For each confocal imaging experiment, 10 worms were used, and the healthy neurons were counted manually, and each condition (day) consisted of six independent experiments with freshly bleached worms. The data were expressed as mean and the error bars report the s.e.m. ($n = 6$ independent experiments and each experiment consisted of 10 technical replicates). The statistical analysis was performed using ANOVA with Tukey's multiple comparison test. * $p < 0.05$, ** $p < 0.01$, *** $p < 0.001$, **** $p < 0.0001$.

“We compared NS163 and NS132 with other reported ligand inhibitors of α S aggregation in the literature for their ability to rescue the degeneration of DA neurons in UA196 worms under matched conditions (50 μ M concentration). The number of intact DA neurons in UA196 worms on day 15 in the presence of Bexarotene, Tyrosol, Valporic acid, and EGCG was 19.3, 21.7, 18.7, and 30.5, respectively (Fig. 5l, Supplementary Fig. 27). Most of the ligands were ineffective in rescuing the degeneration of DA neurons in UA196 worms (Fig. 5l, Supplementary Fig. 27). EGCG was moderate in rescuing the degeneration of DA neurons (Fig. 5l, Supplementary Fig. 27). Our results corroborate well with the published data as some of these molecules demonstrate noticeable rescue effect at one mM concentration⁶⁴⁻⁶⁶. Under matched conditions, in the presence of NS132 (50 μ M), the total number of intact DA neurons on day 15 was 51, which confirms that NS132 also effectively rescues the degeneration of DA neurons in UA196 worms (Supplementary Fig. 27, 28a-e). The neuroprotective effect of NS163 was better than NS132, indicated by a higher number of intact healthy neurons (Fig. 5k, Supplementary Fig. 28a-e), most likely due to the former's better ability to permeate the cell membrane.”

Supplementary Fig. 28. Neuroprotective effect of NS132 on the degeneration of DA neurons. **a**, Schematic of the aging process of UA196 worms and their treatment with the ligands. Representative confocal images of UA196 worms in the presence of 50 μ M NS132 (treatment on day two and four) on day 15 (**b,c**). The healthy DA neurons (red arrow) and neurites (yellow arrow) in UA196 worms on day 15. The relative number of neurons in UA196 worms during the aging process in the absence (**d**) and presence of 50 μ M NS132 (**e**). **f**, Statistics for the total number of neurons in UA196 worms during the aging process in the absence (blue) and presence (orange) of 50 μ M NS132. For each confocal imaging experiment (**b,c**), at least 10 worms were used, and the healthy neurons were counted manually, and each condition (day) consisted of six independent experiments with freshly bleached worms. The data were expressed as mean and the error bars report the s.e.m. ($n = 6$ independent experiments and each experiment consisted of 10 technical replicates). The statistical analysis was performed using ANOVA with Tukey's multiple comparison test. * $p < 0.05$, ** $p < 0.01$, *** $p < 0.001$, **** $p < 0.0001$.

Supplementary Fig. 29. The rescue of motility rate in UA196 worms by NS132. The comparison of the motility rate of N2 (green bar) and UA196 in the absence (blue bar) and presence (orange bar) of 50 μ M NS132 (treatment on day two and four). For motility rate experiment, a total of 50 worms were used in duplicate for each experiment and each condition consisted of four independent experiments. The data were expressed as mean and the error bars report the s.e.m. ($n = 4$ independent experiments and each n consisted of two technical replicates). The statistical analysis was performed using ANOVA with Tukey's multiple comparison test. * $p < 0.05$, ** $p < 0.01$, *** $p < 0.001$.

Supplementary Fig. 30. The effect of NS132 on the ROS level in UA196 worms. The comparison of the ROS level in UA196 worms in the absence and presence of 50 μM NS132 (treatment on day two and four) at the indicated time points. The UA196 worms were treated with NS132 on day two and four and the ROS level was measured on day eight. For ROS level quantification, at least 50 worms were used and each condition consisted of three independent experiments with freshly bleached worms. The data were expressed as mean and the error bars report the s.d. ($n = 3$ independent experiments and each n consisted of three technical replicates). The statistical analysis was performed using ANOVA with Tukey's multiple comparison test. * $p < 0.05$, ** $p < 0.01$, *** $p < 0.001$, **** $p < 0.0001$.

Supplementary Fig. 31. The CI graph for N2 worms in the absence (green bar) and presence (red bar) of 50 μ M NS163 (treatment on day two and four) under the indicated conditions on day 10. For chemotaxis assays, a total of 50 worms were used in duplicate for each experiment and each condition consisted of three independent experiments with freshly bleached worms. The data were expressed as mean and the error bars report the s.e.m. ($n = 3$ independent experiments and each n consisted of two technical replicates).

Supplementary Fig. 32. The effect of NS132 on the behavioral deficits mediated by α S aggregation in DA neurons in UA196 worms. **a**, The CI graph for UA196 worms treated with 50 μ M NS132 (treatment on day two and four) under the indicated conditions on day 10. **b**, The snapshots at 60 min. of the animated videos (Movie S8) collected for the CI for UA196+NS132 under the indicated conditions. For chemotaxis assays, a total of 50 worms were used in duplicate for each experiment and each condition consisted of three independent experiments with freshly bleached worms. The data were expressed as mean and the error bars report the s.e.m. ($n = 3$ independent experiments and each n consisted of two technical replicates).

Supplementary Fig. 35. Neuroprotective effect of NS132 on the preexisting PD *C. elegans* model. **a**, Schematic of the aging process of UA196 worms and their treatment with NS-132 in the late-stage disease onset PD model (day 5). Representative confocal images of UA196 worms on day 15 in the absence (**b,c**) and presence (**d,e**) of 50 μ M NS132 (treated on day 5). Filled red and yellow arrows = healthy neurons and neurites, empty red and yellow arrows = degenerated neurons and neurites. **f**, Statistics for the total number of healthy neurons in UA196 worms during the aging process in the absence and presence of 50 μ M NS132. **g**, The relative number of healthy DA neurons in UA196 worms during the aging process when treated with 50 μ M NS132 on day 5. For each confocal imaging experiment at least 10 worms were used, and the healthy neurons were counted manually, and each condition (day) consisted of six independent experiments. The data were expressed as mean and the error bars report the s.e.m. (n = 6 independent experiments and each n consisted of 10 technical replicates). The statistical analysis was performed using ANOVA with Tukey's multiple comparison test. *p < 0.05, **p < 0.01, ***p < 0.001, ****p < 0.0001.

Supplementary Fig. 36. Neuroprotective effect of various ligands on the degeneration of DA neurons in a post-disease PD model of UA196 worms. Statistics for the total number of intact DA neurons in UA196 worms in the absence and presence of the indicated ligands (at 50 μ M, day 5) on day 15. For each confocal imaging experiment at least 10 worms were used, and the healthy neurons were counted manually, and each condition (day) consisted of six independent experiments. The data were expressed as mean and the error bars report the s.e.m. ($n = 6$ independent experiments and each n consisted of 10 technical replicates). The statistical analysis was performed using ANOVA with Tukey's multiple comparison test. * $p < 0.05$, ** $p < 0.01$, *** $p < 0.001$, **** $p < 0.0001$.

“We also compared the antagonist activity of NS163 and NS132 with other reported inhibitors of α S aggregation for their ability to rescue the degeneration of DA neurons in UA196 worms in the late-stage onset PD model. The ligands (50 μ M) were added on day five and the number of intact DA neurons was counted on day 15. The number of intact DA neurons decreased from 58.7 (day 3) to 12.8 (day 15). At this dose, the number of intact DA neurons in UA196 worms on day 15 in the presence of NS163, Bexarotene, Tyrosol, Valporic acid, and EGCG was 41.3, 13.2, 14.3, 13.5, and 20.5, respectively (Fig. 6l). All other ligands except EGCG were not able to rescue the degeneration of DA neurons in UA196 worms (Fig. 6l). EGCG was moderate in rescuing the degeneration of DA neurons. NS132 was also effective (less than NS163) in rescuing degeneration of DA neurons in a post-disease onset PD model (Supplementary Fig. 35a-f, 36). The number of intact DA neurons in UA196 worms on day 15 in the presence of NS132 was 32, which indicates that it was a far better ligand than other reported ligands from

literature in rescuing the degeneration of DA neurons (Supplementary Fig. 35a-f, 36). Overall, NS163 was the most potent ligand in rescuing degeneration of DA neurons in the post-disease onset PD model.

Supplementary Fig. 37. The effect of NS132 on the ROS level in UA196 worms. The comparison of the ROS level in UA196 worms (blue) on day five and day eight when treated (orange) with 50 μ M NS132 on day five. For ROS level quantification, at least 50 worms were used and each condition consisted of three independent experiments with freshly bleached worms. The data were expressed as mean and the error bars report the s.d. ($n = 3$ independent experiments and each n consisted of three technical replicates). The statistical analysis was performed using ANOVA with Tukey's multiple comparison test. * $p < 0.05$, ** $p < 0.01$, *** $p < 0.001$, **** $p < 0.0001$.

Therefore, our data show that using our 2D FAST, we were able to identify NS132 as a potent antagonist of α S aggregation mediated PD phenotypes in two HEK cells models and two C. elegans PD models, without changing the chemical structure of the pyridyls. We understand that cell permeability is an important factor for testing the ligands in cellular and in vivo assays. However, if we go with that notion that we design and synthesize library of ligands with chemical modalities that have better ability to permeate cell membrane, then we might miss out on potent antagonists. Therefore, our approach relies on finding the most potent antagonists of α S aggregation and then modify the cell permeability of ligands without sacrificing their antagonist activity against α S aggregation, which is exactly we did in our current study.

We identified NS132 as a potent antagonist of α S aggregation. It was also able to rescue α S aggregation potentiated PD phenotypes in cellular and C. elegans PD models. However, its cell permeability was lower likely due to the COOH group as a side chain. Therefore, we anticipated that by enhancing the cell permeability of NS132, we will further enhance the antagonist activity of NS132 in cellular and C. elegans PD models. Therefore, we tuned the cell permeability of NS132 using medicinal chemistry approaches. It is a very common practice in medicinal chemistry to optimize the pharmaceutical properties of the most potent ligands; therefore, we decided

to optimize the cell permeability of NS132 using a medicinal chemistry approach. We synthetically changed the COOH functional group with one of its “isostere” group. The isostere groups are the functional groups that mimic the parental functional group (in our case COOH), enhance the cell permeability of the ligand, without sacrificing their antagonist activity. Here, in our study we selected hydroxylamine as an isostere of COOH functional group. The hydroxylamine group is a widely used isostere for the COOH group in medicinal chemistry to enhance the cell permeability (Lassalas, P. et al. J. Med. Chem. 2016). Using our synthetic approach, we replaced COOH group (NS132) with hydroxylamine group (NS163). As we anticipated, the cell permeability of NS163 was better than NS132 (Fig. 4a). Using a series of biophysical experiments, we showed that the antagonist activity of NS132 and NS163 ligands was very similar (Fig. 4c-f). Subsequently, we compared the antagonist activity of NS163 and NS132 in cellular and *C. elegans* PD models. NS163 was a better antagonist than NS132 in rescuing PD phenotypes in cellular (Fig. 4g-h) and *C. elegans* (Fig. 4i-l, Supplementary Fig. 25-32, 35-37) based PD models. Clearly, we were able to improve the cell permeability of an OP ligand without sacrificing its antagonist activity.

We consider this as a major strength of our approach, where we can tune the cell permeability of the ligands without sacrificing their antagonist activity. Here, we have shown that after the identification of a potent antagonist (NS132) of aS aggregation, the pharmaceutical properties of NS132 can be improved without sacrificing its antagonist activity against aS aggregation. This study also underpins the strength of our approach that we can conveniently enhance the pharmaceutical properties of a ligand without sacrificing their antagonist activity.

5. Although the author used a cellular system for the validation, they failed to show the compound efficacy in cells because the authors only treated cells with aSyn pretreated with the compounds during the fibril formations. It would be more important to see if the compound inhibits the intracellular aggregation induced by the in vitro-created α Syn fibrils without compound treatment. It might be better to screen the efficacy of, i.e., the top 10 compounds from the ThT assay by this cell-based assay. At least a real cell-based assay should be performed on NS132 and NS163 after Fig. 4a, b, and c.

Response:

We agree with reviewer that we did not perform an experiment to see if the compounds can inhibit the intracellular aggregation of aS templated by exogenously added aS fibers. As suggested by the reviewer, we have performed two HEK cell-based experiments to test multiple tripyridyls for their ability to inhibit the intracellular aS aggregation templated by exogenously added aS fibers. We have included these experiments and Figures (Fig. 4g,h; Supplementary Fig. 18a-c, 19) in the manuscript, which reads as below:

Fig. 4. The intracellular inhibition of α S aggregation by OPs in PD models. **a**, Assessment of cell permeability of the indicated ligands using the PAMPA. **b**, The chemical structures of NS132 and NS163. **c**, The aggregation profile of 100 μ M α S in the absence and presence of the indicated ligands at an equimolar ratio. **d**, The ITC thermogram for the titration of a solution of NS163 into α S where heat burst curves and the corrected injection heats are represented by the upper panel and the lower panel, respectively. **e**, The relative volume changes of the backbone amide peaks of 15 N-labeled α S (70 μ M) in the presence of NS163 at an equimolar ratio. The dashed line represents the reported volume changes in 15 N-labeled α S residues peaks in the presence of NS132 above the value of 5%. The colored sequences (blue and green) are the potential binding sites of NS163 on α S. **f**, The comparison of the aggregation profile of 100 μ M α S in the absence and presence of the indicated ligands at an equimolar ratio. Confocal images (**g**) and statistical analysis (**h**) of HEK cells treated with the aggregated solution of 5 μ M α S, followed by the treatment with the indicated ligands (10 μ M). Inclusions of α S_{A53T}-YFP = white arrows, Hoechst = blue, α S-ps-129 = red, merge = Hoechst, α S-ps-129, and α S_{A53T}-YFP. The representative confocal images of NL5901 worms with α S inclusions (white arrows) in the body wall muscle cells (Day 8) in the absence (**i**) and presence of 50 μ M NS132 (**j**) and 50 μ M NS163 (**k**). **l**, The number of inclusions for experiment in NL5901 in the absence (blue bar) and presence of 50 μ M NS132 (orange bar) and 50 μ M NS163 (red bar) from day four to day nine. **m**, The motility of N2 (green bar) and NL5901 and statistics (Day 8, **n**) in

the absence (blue bar) and presence (red bar) of 50 μ M NS163 during the aging process. The ThT experiments were conducted three times and the reported change in the ThT intensity was an average of three independent experiments and the error bars report the s.d. For each confocal imaging experiment, at least 10 worms were used, and the inclusions were counted manually, and each condition (Each day) consisted of at least four independent experiments. For motility experiments, a total of 50 worms were used in duplicate for each experiment and each condition consisted of at least four independent experiments. The data were expressed as mean and the error bars report the s.e.m. (n = 3 or 4 independent experiments and each n consisted of a minimum of three technical replicates). The statistical analysis was performed using ANOVA with Tukey's multiple comparison test. * $p < 0.05$, ** $p < 0.01$, *** $p < 0.001$.

Supplementary Fig. 18. **a**, Assessment of cell permeability of the indicated ligands using the PAMPA. Confocal images **(b)** and statistical analysis **(c)** of HEK cells treated with the aggregated solution of 5 μ M α S, followed by the treatment with the indicated ligands (10 μ M). Inclusions of α S_{A53T}-YFP = white arrows, Hoechst = blue, α S-ps-129 = red, merge = Hoechst, α S-ps-129, and α S_{A53T}-YFP. The PAMPA assays were conducted three independent times with three technical replicates. The Proteostat assays were conducted with at least four biological replicates and three technical replicates for each biological replicate. The data were expressed as mean and the error bars report the s.e.m. (n = 3-4 independent experiments and each n consisted of three technical replicates). The statistical analysis was performed using ANOVA with Tukey's multiple comparison test. * $p < 0.05$, ** $p < 0.01$, *** $p < 0.001$, **** $p < 0.0001$.

“To test the antagonist activity of NS163 against intracellular α S aggregation, we utilized HEK293 cells, which stably express α S-A53T-YFP^{23,24,50}. The endogenous monomeric α S-A53T-YFP template into fibers when transfected with exogenously added α S fibers in the presence of Lipofectamine 3000 (Fig. 3a)^{23,24,50}. In this assay,

preformed fibers of α S (5 μ M monomer concentration) were introduced to the HEK cells, followed by the introduction of NS163 to the cells after 8h, which is the time required for the internalization of α S fibers, as we have shown recently. The HEK cells were incubated for an additional 18 h, which is the total time (24 h) required for the complete maturation of α S inclusions in the HEK cells. To test the effect of the NS163 on the intracellular inclusions (α S-A53T-YFP), we used confocal imaging and a ProteoStat dye-based assay recently developed in our lab²⁴. In comparison to the control (Fig. 4g, control), there was an abundance of intracellular inclusions in the presence of α S fibers (Fig. 4g, + α S fibers). In contrast, there was a significant reduction in inclusions in the presence of 10 μ M NS163 (Fig. 4g, +NS163). The number of inclusions was even further reduced by NS163 at 25 μ M concentration, very close to the control (Fig. 4g,h, +NS163). The ProteoStat dye-based intensity of HEK cells treated with α S fibers was \sim 5 fold higher than the control (no fibers) (Fig. 4h, + α S fibers). NS163 was able to reduce the intracellular inclusions of α S by 76% and 90% at 10 μ M and 25 μ M, respectively, measured using the ProteoStat dye-based assay (Fig. 4h, +NS163). NS163 inhibits the formation of α S inclusions in HEK cells in a dose dependent manner. NS132 was also able to inhibit the formation of inclusions by 47%, which is less than NS163 (Fig. 4g,h +NS132). It is likely to be a consequence of the better cell permeability of NS163 than NS132. We also compared other tripyridyls with NS163 for their ability to inhibit intracellular α S aggregation using this assay. We used multiple tripyridyls (NS127, NS131, NS132, and NS163) with varying antagonist ability to inhibit α S aggregation. To test their antagonist activity against intracellular α S aggregation, we first determined the cell permeability of the tripyridyls using the PAMPA (Fig. 4a, Supplementary Fig. 18a). The increasing order of the cell permeability was NS131<NS127<NS132<NS163 (Fig. 4a, Supplementary Fig. 18a). The cell permeability of NS131 was the lowest, most likely because of the sulfonic acid side chain. The cell permeability of NS127 and NS132 was very similar because of the same COOH group and hydrophobic groups (Fig. 4a, Supplementary Fig. 18a). The cell permeability of NS163 was highest because the COOH group was replaced with hydroxylamine group (Fig. 4a). We utilized the HEK293 cells-based assay as we used for NS163 and NS132 to determine the antagonist activity of other tripyridyls against intracellular α S aggregation. To test the tripyridyls, we first incubated the HEK cells with preformed α S fibers (5 μ M, monomer concentration) for 8 h. Subsequently, we added tripyridyls (10 μ M) to the HEK cells and then we incubated the cells for additional 18 h, which is the total time (24 h) required for the complete maturation of α S inclusions in the HEK cells. We used confocal imaging and the ProteoStat dye-based assay to monitor the effect of tripyridyls on intracellular α S inclusions (Supplementary Fig. 18a-c). We observed a weak antagonist effect of NS131 (8% decrease in inclusions) on the intracellular α S aggregation even though it was a good antagonist of α S aggregation (from ThT, Fig. 2k, Supplementary Fig. 18b,c). The poor intracellular antagonist activity of NS131 is most likely because of very poor cell permeability (Supplementary Fig. 18a). We observed a noticeable inhibition (\sim 20% decrease in inclusions) of the intracellular aggregation of α S in the presence of NS127 (Supplementary Fig. 18b,c). NS132 (47% decrease in inclusions) was a much better antagonist than NS127 of the intracellular α S aggregation (Fig. 4g,h, Supplementary Fig. 18b-c). It is worth noting that the cell permeability of NS127 was comparable to NS132

(Fig. 4a, Supplementary Fig. 18a). The ThT assay also validated these results that NS132 is a far more potent antagonist of α S aggregation (Fig. 2k). NS163 was the most potent antagonist of the intracellular α S aggregation (76% decrease in inclusions) among the tripyridyls (Fig. 4g,h, Supplementary Fig. 18b-c). Both NS163 and NS132 were comparable in the inhibition of α S aggregation from ThT and gel shift assays; however, the cell permeability of NS163 is better than NS132 (Fig. 4a). These results demonstrate that higher cell permeability and antagonist activity together allow for better antagonist activity against intracellular aggregation of α S.”

Supplementary Fig. 19. The relative intensity of Proteostat dye-stained intracellular inclusions of α S_{A53T}-YFP after treatment with the indicated ligands (10 μ M) followed by the treatment with α S fibers (5 μ M monomer conc.) for 24 h. The Proteostat assays were conducted with at least four biological replicates and three technical replicates for each biological replicate. The data were expressed as mean and the error bars report the s.e.m. (n = 4 independent experiments and each n consisted of three technical replicates). The statistical analysis was performed using ANOVA with Tukey’s multiple comparison test. *p < 0.05, **p < 0.01, ***p < 0.001, ****p < 0.0001.

“We further tested NS163 against intracellular α S aggregation using another HEK cell-based assay. For this assay, we utilized the same HEK293 cells, which stably express YFP-labeled α S-A53T mutant (α S-A53T-YFP)^{23,24,50}. In this assay, we added NS163 (10 μ M) to the HEK cells and incubated the cells for 16 h to allow the internalization of NS163. Subsequently, the cells were washed with 1 \times PBS (3 times) and treated with the preformed fibers of α S (5 μ M monomer concentration). The HEK cells were incubated for an additional 24 h, which is the total time (24 h) required for the complete maturation of α S inclusions in the HEK cells. The ProteoStat dye-based intensity of HEK cells treated with α S fibers was ~4 fold higher than the control (no fibers) (Supplementary Fig. 19). NS163 was a very potent antagonist as it was able to reduce the intracellular inclusions by 80% at 10 μ M, measured using the ProteoStat dye-based assay (Supplementary Fig. 19). Under matched conditions, 10 μ M NS132 was able to reduce the intracellular inclusions by 57% (Supplementary Fig. 19). The higher antagonist activity of NS163 than NS132 is likely due to the better cell permeability of the former. Collectively, NS163 is a potent inhibitor of α S fiber-catalyzed aggregation of α S in the cellular milieu.”

In Fig3a, differences in ThT fluorescence between monomeric α Syn and the group supplemented with the 20% of α Syn fibrils seem to be too subtle. Usually, although the ThT value of monomer α Syn is quite low, the presence of α Syn fibrils seeds should lead to a higher value of ThT fluorescence intensity. Why is there only a few-fold difference in fluorescence intensity with and without α Syn fibrils? In addition, important kinetic information would be available for the readers if the raw kinetics is shown here or as supplementary material, as the authors showed in the figure 2.

Response:

*In Fig3a, the ThT intensity difference between α S monomer and α S fibers is ~ 4 fold. We agree with the reviewer that the ThT intensity should have been a few fold more in the case of α S fibers catalyzed aggregation. One of the likely reasons could be that after the complete aggregation of α S monomer (templated by α S fibers), some α S fibers crashed out of the solution, which lead to a decrease in the overall ThT intensity. Since, it was a single time point experiment (only 24 h reading), it was hard to assess the reason. As suggested by the reviewer, we have repeated the experiment and included the full kinetics of the α S fibers catalyzed aggregation of α S monomer in the absence and presence of NS132. The data has been added to the main manuscript (**Fig.3e**) and a writeup has been included in the main manuscript, which reads as follows:*

Fig. 3e. The aggregation profile of 100 μ M α S catalyzed by α S fibers (20% monomer concentration) in the absence and presence of NS132 at an equimolar ratio. The ThT experiments were conducted three times and the reported change in the ThT intensity was an average of three independent experiments and the error bars report the s.d.

“Therefore, we monitored the effect of NS132 on the α S fibers catalyzed aggregation of α S using ThT kinetic assay. The aggregation kinetics of 100 μ M α S in the presence of preformed α S fibers (20%, monomer concentration) resulted in the acceleration of monomeric α S aggregation (Fig. 3e, α S, blue line), which is reflected

by a significant increase in the ThT signal and the disappearance of the lag phase (Fig. 3e, + α S fibers, black line). The α S fiber-catalyzed aggregation of α S was wholly suppressed by NS132 at an equimolar ratio, as evidenced by a significantly lower ThT signal (Fig. 3e, + α S fibers+NS132, orange line).”

7. All nematode studies tested a single concentration of 50uM. How did the authors determine the optimal concentration? Concentration dependence needs to be evaluated in some experiments of C.elegans, such as in Fig4d and 4h. For motility assay, the evaluation on a specific day might be enough.

Response:

We agree that we have presented data for a single concentration (50 μ M) of NS163 with the nematode study. However, we had conducted a dose dependent study of NS163 against various nematode models and we presented the data just for a single concentration (50 μ M) where NS163 was able to completely rescue PD phenotypes. We had carried out the dose dependent study, which was a part of another manuscript. We have now included the dose dependent study of NS163 (10 μ M, 25 μ M, and 50 μ M) as suggested by the reviewer. We have tested effect of NS163 in a dose dependent manner (10 μ M, 25 μ M, and 50 μ M) in NL5901 for α S inclusions (day 8 and day 9, **Supplementary Fig. 21**) and motility (day 7,8, and 9, **Supplementary Fig. 23**). We also carried out a dose dependent study of NS163 to assess its effect on the degeneration of DA neurons in UA196 worms in a late-stage onset disease model (**Supplementary Fig. 34**). The Figures (**Supplementary Fig. 21, 23, 34**) and writeup for these experiments have been included in the main manuscript, which reads as follows:

Supplementary Fig. 21. The comparison of the number of intracellular α S inclusions in NL5901 worms in the absence and presence of the indicated doses of NS163 on day eight (blue bar) and nine (red bar). For each confocal imaging experiment, at least 5 worms were used and the inclusions were counted manually, and each condition (each day) consisted of three independent experiments. The data were expressed as mean and the error bars report

the s.e.m. (n = 3 independent experiments and each n consisted of a minimum of 5 technical replicates). The statistical analysis was performed using ANOVA with Tukey's multiple comparison test. *p < 0.05, **p < 0.01, ***p < 0.001, ****p < 0.0001.

“We also carried out a dose-dependent study of NS163 against the intracellular α S inclusions in NL5901 worms. The number of α S inclusions in NL5901 worms decreased from ~34/32 (day 8/9) to ~19/20 (day 8/9) and ~9/10 (day 8/9) at 10 and 25 μ M NS163, respectively (Supplementary Fig. 21). The data suggest that NS163 inhibits α S inclusions in NL5901 worms in a dose-dependent manner (Supplementary Fig. 21).”

Supplementary Fig. 23. The statistics for the relative motility rate of N2 and NL5901 in the absence and presence of various doses of NS163 on the indicated days. For motility experiments, a total of 50 worms were used in duplicate for each experiment and each condition consisted of at least four independent experiments. The data were expressed as mean and the error bars report the s.e.m. (n = 4 independent experiments and each n consisted of a minimum of two technical replicates). The statistical analysis was performed using ANOVA with Tukey's multiple comparison test. *p < 0.05, **p < 0.01, ***p < 0.001, ****p < 0.0001.

“We also carried out a dose-dependent study of NS163 against the motility of NL5901 worms. We observed a noticeable increase in the motility of NL5901 worms from 10 to 25 μ M NS163 (Supplementary Fig. 23). The data suggests that NS163 improves the motility of NL5901 worms in a dose-dependent manner (Supplementary Fig. 23).”

Supplementary Fig. 34. The relative number of healthy DA neurons in UA196 worms during the aging process when treated on day five with the indicated doses of NS163. For each confocal imaging experiment, at least 10 worms were used, and the healthy neurons (GFP signal) were counted manually, and each condition (day) consisted of six independent experiments (total of 60 DA neurons). The data were expressed as mean and the error bars report the s.e.m. (n = 6 independent experiments and each n consisted of 10 technical replicates).

“NS163 was also effective in rescuing the degeneration of DA neurons at lower doses as the number of intact healthy DA neurons was 34/27 (25 μ M/10 μ M) and 27/17 (25 μ M/10 μ M) on day 10 and day 15, respectively (Supplementary Fig. 34). Clearly, NS163 was potent in stopping the further degeneration of DA neurons when added on day five to UA196 worms as a post-disease onset PD model.”

8. In figure 4e-f, the expression level of the aSyn seems to be affected by the treatment with compounds since the ubiquitous aSyn-YFP signal was also diminished. It is necessary to confirm if there is any difference in the expression of aSyn between groups, e.g., by western blot. Alternatively, it is desirable to present more images of each group at day8, for example, to see if this image is actually representative.

Response:

We agree that the confocal representative images of NL5901 worms in the presence of NS163 were not of high quality as suggested by reviewer 3 as well. We have now included several high-resolution confocal images of NL5901 worms in the absence and presence of NS163 and NS132 (day 8). We have also included confocal images of full length NL5901 worms in the absence and presence of NS163 and NS132 as suggested by reviewer 3 (day 8). These confocal images suggest that NS163 and NS132 do not affect the expression level of aSyn-YFP. The new representative confocal images have been added to the main manuscript (Fig. 4i-k) and multiple images of NL5901 with and without ligands have been included in the supplementary information (Supplementary Fig. 20)

Fig. 4i-k. The representative confocal images of NL5901 worms with α S inclusions (white arrows) in the body wall muscle cells (Day 8) in the absence (i) and presence of 50 μ M NS132 (j) and 50 μ M NS163 (k). For each confocal imaging experiment, at least 10 worms were used, and the inclusions were counted manually, and each condition (Each day) consisted of at least four independent experiments.

Supplementary Fig. 20. The representative confocal images of α S-YFP inclusions in the body wall muscle cells of NL5901 (Days = 8) in the absence and presence of 50 μ M NS163 and 50 μ M NS132 (treated on day

two and four). These confocal images were collected from at least three independent experiments with random selection of the worms from each independent experiment.

9. I have a serious concern about the method with the preparation for the locomotion assay. Worms are sensitive to changes in temperature and humidity, so worms need to be cultured in an incubator with constant temperature and controlled humidity. However, some protocols mention that the authors cultured worms at room temperature for 24 hours. It is required to describe what temperature RT means and whether it is constant. Also, if the nematodes are kept still during liquid culture, they are severely compromised due to a lack of oxygen. It is necessary to describe whether shaking is applied, and if so, at what RPM.

Response:

*We apologize for not providing detailed method information about the culture of the *C. elegans* models. We cultured the worms at 20 °C and the temperature and humidity were constant in the incubator throughout the experiment. For various worm experiments (for both NL5901 and UA196 stains) that were carried out in the liquid media, the worm plates were shaken on a shaker (rpm = 100, orbital) throughout the course of the experiment. We have now modified the protocol for the worm experiments by including more detailed procedures. The modified protocol is included in the method section in the main manuscript and now it reads:*

C. elegans Experiments

Media and buffers for various worm assays.

The worms were maintained at standard conditions (20 °C at all times) on nematode growth media (NGM) agar in 60 mm plates (CytoOne, Ocala, FL) using *E. coli* OP50 strain as the food source following the previous protocols^{4,7,8}. To ensure that worm's colonies don't starve, they were always transferred on new plates with *E. coli* OP50 strain as the food source. All worm strains were replaced with new worm colony after every six months to avoid any genetic mutation, which might affect the disease phenotypes. NGM agar plates, M9 buffer (3 g KH₂PO₄, 6 g Na₂HPO₄, 5 g NaCl, 1 mL 1 M MgSO₄, milli-Q H₂O to 1L), Chemotaxis (CTX) media plates (2% Agar, 5 mM KH₂PO₄, 1 mM CaCl₂, and 1 mM MgSO₄), and CTX buffer (5 mM KH₂PO₄ 1 mM CaCl₂, and 1 mM MgSO₄) were prepared using previous protocols^{4,8-10}

Strains

The N2 (wild-type *C. elegans* Bristol strain), NL5901 (*C. elegans* model of PD), and *Escherichia coli* OP50 (*E. coli*, a uracil requiring mutant) strains were obtained from Caenorhabditis Genomics Center (CGC, Minneapolis, MN). The UA196 strain was generously donated by the laboratory of Dr. Guy Caldwell (Department of Biological Science, The University of Alabama, Tuscaloosa, AL, United States)⁶.

NL5901 strain (pkIs2386 [unc-54p::alphasynuclein::YFP+unc-119(+)]). In NL5901 strain, αS-YFP is expressed in the muscle cells of worms. The aggregates of αS-YFP are visible from day 4 and the maximum number of αS-YFP aggregates are visible round day 8 and day 9.

UA196 strain [P_{dat-1}::α-syn+P_{dat-1}::GFP]. In UA196 strain, human αS and GFP are expressed in DA neurons under control of a dopamine transporter-specific promoter [P_{dat-1}::α-syn+P_{dat-1}::GFP], which results in age-dependent neurodegeneration of six DA neurons.

Culture methods for *C. elegans* strains

The standard worm conditions were used for the culturing of various worm strains. Briefly, the worms were bleached and synchronized using hypochlorite solution, followed by the incubation of the eggs (at 20 °C) on NGM plates (35 mm, CellTreat Scientific, Pepperell, MA), which were seeded with OP50 (350 µL, 0.5 OD_{600nm}) as a food source, **referred to as PLATE 1**. The cultures of OP50 were prepared by incubating 50 mL of LB medium with OP50 18 h at 37 °C and the final OD value was adjusted to 0.5 at 600 nm. The NGM plates were prepared by treating them with 350 µL OP50 and leaving the plates at 20°C for 3 days. On day two, the worms were transferred (using M9 buffer) to NGM plates (35 mm, CellTreat Scientific, Pepperell, MA) containing 75 µM Fluorodeoxyuridine (FUDR; to prevent worm reproduction and ensure that equal ages of worms were used for the experiment)^{4,8} and OP50 as food source, **referred to as PLATE 2**. These NGM plates were prepared using autoclaved NGM media supplemented with 75 µM FUDR (2 mL total liquid). After 12 h, the NGM plates were seeded with 350 µL of OP50 (with OD = 0.5 at 600 nm) at 20 °C. For treating various worm strains with ligands, the worms were transferred on day 2 to the NGM plates treated with ligands (please see below to prepare the NGM plates treated with ligands).

Preparation of NGM plates with Ligands

The NGM plates (with NGM media) were used for the treatment of different worm strains with ligands. The NGM media containing 75 µM FUDR was autoclaved and poured in NGM plates (2 mL total liquid). After 12 h, the NGM plates were seeded with 350 µL of OP50 (with OD = 0.5 at 600 nm) at 20 °C. After 12 h, different doses of ligands (10-50 µM, stock conc. = 10 mM in DMSO, DMSO from 0.1-0.5%, v/v) were dissolved in M9 buffer (total volume = 300 µL) and were spotted atop the NGM plates at 20 °C, **referred to as PLATE 3**. The NGM plates treated with the ligands were placed in sterile laminar flow hood at 20 °C for 1 h. The NGM plates treated with ligands were prepared and used within 24 h. We used these plates to treat various disease strains for different assays.

Motility assay for *C. elegans* (N2, NL5901 and UA196)

Briefly, all worm strains were bleached at the same time and the eggs were incubated on **PLATE 1**. On day two, the worms were divided into two batches. One half of the worms were transferred (using M9 buffer) on **PLATE 2 (without ligand)** and the second half of the worms were transferred to **PLATE 3 (with ligand)**. The concentration of the ligands varied from 10-50 µM as described in the main manuscript. The worms were incubated on **PLATE 2 or PLATE 3** up to day 4 at 20 °C in an incubator with constant humidity. On day four, various worm strains (with and without ligands from PLATE 2 or PLATE 3) were transferred to sterile 24 well plate (CellTreat Scientific, Pepperell, MA) containing liquid media (500 µL/well), **referred to as PLATE 4A** and containing liquid media with ligands (500 µL/well with 10-50 µM ligand in M9 buffer with 0.1%-0.5% DMSO, v/v), **referred to as PLATE 4B**. The liquid media for **PLATE 4A/B** was prepared with 67.28% (v/v) of M9 buffer, 75 µM FUDR, 0.1% of 1 M magnesium sulfate (v/v), 0.1% of 1 M calcium chloride (v/v), 2.5% of 1 M potassium phosphate solution (pH 6, v/v), and 30% (v/v) of OP50 (0.5 OD_{600nm}). A total of 50 worms per well were manually transferred into **PLATE 4A/B** at 20 °C and a total of four wells (4 technical replicates) were used for each condition. Subsequently, the worms were incubated for 6 h at 20 °C with constant shaking (rpm = 100) to get acclimated with the solution conditions before starting the motility assay experiment.

To test the effect of each ligand on disease worm strains (NL5901 or UA196), there were four conditions were used in **PLATE 4A/B**: (1) N2 worms, (2) N2 worms treated with ligand, (3) NL5901 (or UA196) worms, and (4) NL5901 (or UA196) worms treated with the ligand (10-50 µM). The assay was started on day four using the WMicroTracker ARENA plate reader (Phylumtech, Santa Fe, Argentina) at 20 °C for 1 h per day over a 14-day period at intervals of 24 h. Each day, a total of 20 activity scores per well were collected in 1 h. After collecting the data each day, the **PLATE 4A/B** was again placed on the shaker (100 rpm) at 20 °C in the incubator. The **PLATE 4A/B** was on the on the shaker (100 rpm) at 20 °C in the incubator the whole duration of the experiment

except only when the reading was collected on the WMicroTracker ARENA plate reader for 1 h. For each condition, four biological replicates were performed, and each biological replicate consisted of four technical replicates (four wells). For each condition, the data were expressed as mean and the error bars report the s.e.m. (n = 4 independent experiments and each n consisted of a total of four technical replicates).

Motility assay for *C. elegans* (N2 and UA196) in the presence of dopamine and NS163

This motility assay was performed exactly similar to the previous assay with slight adjustment in the preparation of the plates for the treatment of UA196 worms. For this assay, the only difference is the 24 well plates were prepared with 2 mM dopamine (**referred to as PLATE 5**) and both 2 mM dopamine and 50 μ M NS163 (**referred to as PLATE 6**). To prepare **PLATE 5**, the NGM media containing 75 μ M FUDR was autoclaved and poured in NGM plates (2 mL total liquid). After 12 h, the NGM plates were seeded with 350 μ L of OP50 (with OD = 0.5 at 600 nm) at 20 °C. After 12 h, 2 mM dopamine dissolved in M9 buffer (total volume = 300 μ L) was spotted atop the NGM plates at 20 °C. To prepare **PLATE 6**, the NGM media containing 75 μ M FUDR was autoclaved and poured in NGM plates (2 mL total liquid). After 12 h, the NGM plates were seeded with 350 μ L of OP50 (with OD = 0.5 at 600 nm) at 20 °C. After 12 h, NS163 (50 μ M, stock conc. = 10 mM in DMSO) and dopamine (2 mM in M9 buffer) were dissolved in M9 buffer (total volume = 300 μ L) and were spotted atop the NGM plates at 20 °C. The NGM plates treated with dopamine and NS163 and dopamine were placed in sterile laminar flow hood at 20 °C for 1 h. The NGM plates treated with dopamine and NS163 and dopamine were prepared and used within 24 h.

Similar to the previous section, the motility assay was performed for N2 and UA196 worms in the absence and presence of NS163+dopamine in sterile 24 well plate (CellTreat Scientific, Pepperell, MA) containing liquid media (500 μ L/well), liquid media + 2 mM dopamine, and liquid media + 2 mM dopamine + 50 μ M NS163. There were six conditions for this experiment: (1) N2 worms, (2) N2 worms treated with 2 mM dopamine, (3) UA196 worms, (4) UA196 worms treated with 2 mM dopamine, (5) UA196 worms treated with 50 μ M NS163, and (6) UA196 worms treated with 2 mM dopamine and 50 μ M NS163. For each condition, four biological replicates were performed, and each biological replicate consisted of four technical replicates (four wells). For each condition, the data were expressed as mean and the error bars report the s.e.m. (n = 4 independent experiments and each n consisted of a total of four technical replicates).

Confocal microscopy imaging of early stage treated *C. elegans* (NL5901 and UA196) with ligands

This experiment was performed based on previously described protocols with slight modifications^{4,11,13}. Briefly, the worm strains (NL5901 or UA196) were bleached at the same time and the eggs were incubated on **PLATE 1**. On day two, the worms were divided into two batches. One half of the worms were transferred (using M9 buffer) on **PLATE 2 (without ligand)** and the second half of the worms were transferred to **PLATE 3 (with ligand)**. The concentration of the ligands varied from 10-50 μ M as described in the main manuscript. The worms were incubated on **PLATE 2 or PLATE 3** up to day 3 at 20 °C in an incubator with constant humidity. On day four, the ligand (10-50 μ M, stock conc. = 10 mM in DMSO, DMSO from 0.1-0.5%, v/v) was added again to **PLATE 3**. For confocal imaging, at least 10 worms per condition (from **PLATE 2 or PLATE 3**) were transferred to a cover slide containing an anesthetic (40 mM sodium azide), and mounted on glass microscope slide containing 2% agarose pads using a reported protocol¹³. The images of the worms were collected using an Olympus Fluoview FV3000 confocal/2-photon microscope (40 x Plan-Apo/1.3 NA objective with DIC capability) and processed using the OlympusViewer in ImageJ software⁴. For NL5901 worms, the inclusions of the aggregated α S (GFP puncta in the muscle cells) were manually counted (10 worms per condition) for the five-day imaging period (day 5 to day 9) for **PLATE 2 or PLATE 3**. For UA196 strain, the number of healthy DA neurons (fluorescence due to GFP in DA neurons) were counted (10 worms per condition) using confocal imaging of the worms on day 3, day 5, day 10, and day 15. For each condition, six biological replicates were performed and at least 10 technical

replicates were used for each biological replicate. The data were expressed as mean and the error bars report the s.e.m. (n = 6 independent experiments and each n consisted of a minimum of ten technical replicates).

Chemotaxis Assay for *C. elegans* (N2 and UA196) in the presence of ligands

For chemotaxis assay, the Chemotaxis (CTX) media plates (2% Agar, 5 mM KH₂PO₄, 1 mM CaCl₂, 1 mM MgSO₄, 75 μM FUDR), **referred to as PLATE 7**, and CTX buffer (5 mM KH₂PO₄, 1 mM CaCl₂, and 1 mM MgSO₄) were prepared using previous protocols^{4,8-10}. The **PLATE 7** (without ligand) were treated with ligand (50 μM, stock conc. = 10 mM in DMSO) that was dissolved in CTX buffer (total volume = 300 μL) and was spotted atop the CTX media plates at 20 °C, **referred to as PLATE 8**. For CTX assay, we prepared another plate **referred to as PLATE 9**, which was **PLATE 7** divided into four equal quadrants and designated as A and C (diagonally opposite), B and D (diagonally opposite). A solution of *E. coli* (50 μL of 0.5 OD at 600 nm, as an attractant) was placed at ~0.4 cm from the edge of quadrants B and D and the plate was allowed to dry for 1 h at 20 °C. Subsequently, ethanol (10 μL, repellent) was added at ~0.4 cm from the edge of quadrants A and C at 20 °C. To ensure that the ethanol did not dry, the worms (N2 or UA196) were transferred to this plate within 10 min and the lid of the plate was closed. The experiment on the WMicroTracker ARENA plate reader (Phylumtech, Santa Fe, Argentina) started within 1 h of the transfer of the worms at 20 °C. To carry out the experiment, the worms (N2 or UA196) were bleached at the same time and the eggs were incubated on **PLATE 1**. On day two, the worms were divided into two batches. One half of the worms were transferred (using CTX buffer) on **PLATE 7 (without ligand)** and the second half of the worms were transferred to **PLATE 8 (with 50 μM ligand)** and kept at 20 °C in an incubator. On day 3, a total of 50 worms were transferred to the center of the **PLATE 9** using CTX buffer (~20 μL) and the number of worms were counted under an Olympus microscope (SZ-6145, Waltham, MA). Subsequently, the lid of the plate was closed with parafilm (Bemis Company, Inc., Neenah, WI). The worm activity was monitored using the WMicroTracker ARENA plate reader (Phylumtech, Santa Fe, Argentina) at 20 °C for 2 h. We used exactly similar conditions for day 10 experiment except the ligand was added again (50 μM and 300 μL in CTX buffer, stock conc. = 10 mM in DMSO, DMSO = 0.5%, v/v) to **PLATE 8**. For day 10, 50 worms (N2 or UA196) from **PLATE 7 (without ligand)** or **PLATE 8 (with 50 μM ligand)** were transferred to **PLATE 9** using CTX buffer (~20 μL) and the number of worms were counted under an Olympus microscope (SZ-6145, Waltham, MA). Subsequently, the lid of the plate was closed with parafilm (Bemis Company, Inc., Neenah, WI). The worm activity was monitored using the WMicroTracker ARENA plate reader (Phylumtech, Santa Fe, Argentina) at 20 °C for 2 h. The report was generated using MapPlot option on the WMicroTracker ARENA plate reader. The experiment was conducted for both N2 and UA196 worms (in the absence and presence of 50 μM NS163) on day three and day 10 of the aging process. For each condition, three biological replicates were performed and at least two technical replicates were used for each biological replicate. The data were expressed as mean and the error bars report the s.e.m. (n = 3 independent experiments and each n consisted of two technical replicates).

Measurement of the ROS level in UA196 worms in the presence of ligands (Quantification of ROS and confocal imaging)

The worm strains were bleached at the same time and the eggs were incubated on **PLATE 1**. On day two, the worms were divided into two batches. One half of the worms were transferred (using M9 buffer) on **PLATE 2 (without ligand)** and the second half of the worms were transferred to **PLATE 3 (with ligand)**. The concentration of the ligands was 50 μM as described in the main manuscript. On day four, the ligand (10-50 μM in M9 buffer with 0.1%-0.5% DMSO, v/v) was spotted again atop the **PLATE 3** at 20 °C. The worms were incubated on **PLATE 2 or PLATE 3** up to day eight at 20 °C in an incubator with constant humidity. On day eight, the worms were transferred into 1.7 mL microcentrifuge tubes using M9 buffer (1 mL) and washed with M9 buffer (1 mL and three times) by centrifugation for 2 min at 2,500 rpm and 20 °C. Subsequently, a total of 100 worms/well were transferred to a Costar 96-well black plate (Corning, Kennebunk, ME) containing 100 μL solution, including

M9 buffer (89 μL), worm solution (10 μL and 100 worms), and 2',7'-dichlorofluorescein diacetate (H_2DCFDA) dye (1 μL , 50 μM , stock solution = 5 mM in cell culture grade DMSO, 99% pure). As a control, 99 μL of M9 buffer and 1 μL of 5 mM H_2DCFDA reaction solution was placed in the wells. Subsequently, the Costar 96-well plate was gently shaken for 30 sec at 20 $^\circ\text{C}$ and the fluorescence intensity was measured ($\lambda_{\text{ex}} = 485 \text{ nm}$ and $\lambda_{\text{em}} = 530 \text{ nm}$) at multiple time points (0 to 120 min) using the Infinite M200 Pro Plate Reader (Tecan, Männedorf, Switzerland)^{11,14,15}.

For confocal imaging, the same UA196 worms (in the absence and presence of ligands) used for the quantification of ROS, were transferred from the 96-well plate into **PLATE 2**, air-dried into the sterile laminar flow hood. The worms (~10 worms) were transferred to a cover slide containing an anesthetic (40 mM sodium azide), and mounted on a glass microscope slide containing 2% agarose pads for the confocal imaging as described in the previous sections. This experiment consisted of three biological replicates and each biological replicate consisted of three technical replicates. The data were expressed as mean and the error bars report the sd's (n = 3 independent experiments and each n consisted of three technical replicates).

Confocal imaging of a post-disease onset PD model of UA196 worms in the absence and presence of ligands

The UA196 worms were bleached, and the eggs were incubated on **PLATE 1**. On day two, the UA196 worms were transferred (using M9 buffer) on **PLATE 2 (without ligand)**. On day five, 10 UA196 worms were used to determine the number of healthy DA worms using confocal imaging (as described earlier) prior to the treatment with ligands on day five. In tandem, ~50 worms were transferred to **PLATE 3 (with ligand)** using M9 buffer (10-20 μL) and the plate was incubated at 20 $^\circ\text{C}$ in the incubator. On days 10 and 15, 10 UA196 worms from **PLATE 2 (without ligand)** and **PLATE 3 (with ligand)** were used to determine the number of healthy DA worms using confocal imaging (as described earlier). For each condition, six biological replicates were performed and at least 10 technical replicates were used for each biological replicate. The data were expressed as mean and the error bars report the s.e.m. (n = 6 independent experiments and each n consisted of a minimum of ten technical replicates).

Measurement of intracellular ROS level in a post-disease onset PD model of UA196 worms in the absence and presence of ligands

The UA196 worms were bleached, and the eggs were incubated on **PLATE 1**. On day two, the UA196 worms were transferred (using M9 buffer) on **PLATE 2 (without ligand)**. On day five, 100 UA196 worms were used to determine the ROS level (as described earlier) prior to the treatment with ligands on day five. In tandem, ~200 worms were transferred to **PLATE 3 (with ligand)** using M9 buffer (10-20 μL) and the plate was incubated at 20 $^\circ\text{C}$ in the incubator. On day 8, 100 UA196 worms from **PLATE 2 (without ligand)** and **PLATE 3 (with ligand)** were used to determine the ROS level (as described earlier). For each condition, three biological replicates were performed, and 3 technical replicates were used for each biological replicate. The data were expressed as mean and the error bars report the s.e.m. (n = 3 independent experiments and each n consisted of 3 technical replicates).

10. I also have questions about the motility of NL5901. When properly cultured, NL5901 shows a 10-20% decline in locomotion compared to WT in our laboratory. Other groups have shown a decrease in locomotion up to 50%. It is surprising that the NL5901 show as much as an 80% to 90% decrease in their locomotion. Since a similar phenotype of NL5901 has been observed in previous manuscripts from the authors, I am concerned that some factors might exacerbate a phenotype of the NL5901; e.g., due to infection or the higher temperature during the culture. Does this model manifest temperature sensitive phenotype? In order to verify the strong phenotype, please submit raw data such as video or tracking data during the locomotion assay. If all data is not available, I would especially like to see the data from Day 4 and 8.

Response:

Other groups generally started the locomotion experiment on day 3 in NL5901 worms; therefore, the motility (locomotion) was ~50% of the control worms (N2). In our study, we started the motility study on day 4 instead of day 3 because we treated the NL5901 worms on day 2 with our small molecules and we wanted to allow the NL5901 worms to settle before starting the motility study, which started on day 4. We carried out a comparison of the motility of NL5901 worms from day 3 and 4 and we found that the relative % motility of NL5901 was around ~60% and 32% of the N2 worms on day 3 and 4, respectively. The motility of NL5901 worms on day 3 from our experiment was in close vicinity to others reported in literature. We also compared the α S-YFP inclusions (day 7,8, and 9) of NL5901 worms from day 3 and 4. We did not observe any significant difference in the number of α S-YFP inclusions. Also, the number of α S-YFP inclusions in NL5901 from our study were in close vicinity to others reported in literature (Doherty C.P.A et al. *Nat. Struct. Mol. Biol.* 2020). We acknowledge that we might be using the term “motility rate or locomotion” loosely as we are measuring the overall speed of the worms using a WMicroTracker ARENA worm plate reader. The output value in this plater reader is the average movement of 50 worms in each well instead of the movement of individual worms as reported by others using other methods and instruments. So, if a few NL5901 worms develop PD phenotype quicker than the rest of the worms, the collective output of the motility rate will decrease significantly. We have included the relative motility of N2 and NL5901 worms from day 3/day 4 to day 17 and α S-YFP inclusions (day 7,8, and 9) in the supplementary information (**Supplementary Fig. 22**). We have also included a writeup in the main manuscript, which reads as follows:

“The decline in the motility of NL5901 worms was 70% on day four (Supplementary Fig. 22); however, it is worth noting that we started the motility experiment on day four because of the treatment of worms with ligands on day two and four. However, the decline in the motility of NL5901 worms was much slower (~40%) in comparison to the N2 worms on day three (Supplementary Fig. 22, Movie S1), which further decreased to ~13% on day eight (Supplementary Fig. 22, Movie S2).”

Supplementary Fig. 22. a, The comparison of the motility of N2 and NL5901 worms from day three (blue line) and day four (red line). **b,** The number of α S inclusions in NL5901 worms from day seven to day nine for NL5901 worms from experiments in **a**. For each confocal imaging experiment, at least 5 worms were used, and the inclusions were counted manually. Each condition (Each day) consisted of at least three independent experiments. For motility experiments, a total of 50 worms were used in duplicate for each experiment and each condition

consisted of four independent experiments. The data were expressed as mean and the error bars report the s.e.m. (n = 3 or 4 independent experiments and each n consisted of at least three technical replicates). The statistical analysis was performed using ANOVA with Tukey's multiple comparison test.

We want to verify that NL5901 worms do not manifest temperature sensitive phenotypes at the temperature conditions we are conducting our experiments (20 °C). Also, we are very careful in conducting worm experiments and we did not observe any infection during the culture of worms in our study. We also want to point out that the number of α S inclusions from our study are in good agreement with the earlier published work (Doherty C.P.A et al. Nat. Struct. Mol. Biol. 2020). As suggested by the reviewer, we have also included the videos of days 3 and 8 of N2 and NL5901 in the absence and presence of 50 μ M NS163. These videos have been included in the supplementary information (Raw File_NL5901 worms).

As a final note, for our current manuscript, we have also used another worm strain (UA196 worms) whose PD disease phenotypes are in close proximity to the published work by others (Pujols, P. et al. PNAS, 2018). Therefore, we are confident that our experimental methods are not compromised due to any infection or experimental errors.

11. I am not so convinced with the experiment starting treatment from day5 for the UA196 model because the results are too dramatic. On day5, 28% of cells have degenerated, and we could imagine that a significant amount of α Syn aggregates have formed in the remaining cells, causing a certain level of cytotoxicity. The effect of NS163 can only be expected to prevent further growth of the α Syn aggregation, which might not be able to stop the degeneration cascade of the remaining cells completely. How can NS163 administration completely inhibit neurodegeneration? Please add some comments in the discussion section.

Response:

We surmise two reasons for such a dramatic rescue of the degeneration of DA neurons by NS163 in a post-disease onset PD model. We have included the data (Supplementary Fig. 33, 34) and writeup in the main manuscript, which reads as follows:

“We surmise one main reason for such a dramatic rescue of the degeneration of DA neurons by NS163 in a post-disease onset PD model. Even though it has not been investigated thoroughly, we and others^{20,70-73} have reported that the degeneration level is not the same for all six DA neurons during the aging of UA196 worms. We envision that the manifestation of neurotoxicity in DA neurons due to α S aggregation varies in individual DA neurons, which is the reason for different time intervals for the degeneration of individual DA neurons. Therefore, we speculate that the neurotoxicity in individual DA neurons in UA196 worms might be starting at different time points. It is a stepwise process as some DA neurons degenerate earlier and the degeneration of the DA neurons slows down with the aging of worms, which is evidenced by the fact that the number of degenerated DA neurons is 16.2 in first two days (from day 3 to day 5), 23.5 in next five days (day 5 to day 10), and 10 in next five days (day 10 to day 15) (Fig. 6k, blue line). Since the degeneration of DA neurons is a sequential event, there is a possibility that the degeneration of DA neurons at the later stage is a consequence of both the *de novo* and fibers catalyzed aggregation of α S. Therefore, ligands that can inhibit both the aggregation pathways (*de novo* and fibers-catalyzed aggregation), could completely rescue the degeneration of DA neurons in a post-disease onset PD model. We have shown that NS163 is a potent antagonist of both the *de novo* and fibers-catalyzed aggregation

of α S. Therefore, we surmise that the sequential degeneration of DA neurons and the dual antagonist ability of NS163 are responsible for the effective inhibition of α S aggregation in DA neurons in the post-disease onset PD worm model.”

We repeated this experiment where we carried out a dose dependent study to demonstrate the effect of NS163 on the degeneration of DA neurons in UA196 worms in the late-stage disease model. We have included the writeup and Figure (Supplementary Fig. 34) in the main manuscript, which reads as follows:

“NS163 was also effective in rescuing the degeneration of DA neurons at lower doses as the number of intact healthy DA neurons was 34/27 (25 μ M/10 μ M) and 27/17 (25 μ M/10 μ M) on day 10 and day 15, respectively (Supplementary Fig. 34). Clearly, NS163 was potent in stopping the further degeneration of DA neurons when added on day five to UA196 worms as a post-disease onset PD model.”

Supplementary Fig. 34. The relative number of healthy DA neurons in UA196 worms during the aging process when treated on day five with the indicated doses of NS163. For each confocal imaging experiment, at least 10 worms were used, and the healthy neurons (GFP signal) were counted manually, and each condition (day) consisted of six independent experiments (total of 60 DA neurons). The data were expressed as mean and the error bars report the s.e.m. (n = 6 independent experiments and each n consisted of 10 technical replicates).

Also, we speculate that another factor that might contribute to the degeneration of DA neurons is the templating of the intraneuronal α S monomer by α S fibers generated in individual DA neurons; however, there is no reported study to demonstrate this phenomenon in the DA neurons in UA196 worms. Since the degeneration of all DA neurons is not simultaneous, there is a possibility that the aggregation of α S is not simultaneous in all DA neurons. Therefore, the α S fibers from aggregation of α S in some DA neurons could contribute to the toxicity by templating and accelerating the α S monomers into fibers via fibers catalyzed aggregation mechanism. Therefore, if NS163 can inhibit the fibers catalyzed aggregation of α S, then it will potentially effectively inhibit the degeneration of DA neurons in the late-stage disease model. Therefore, we decided to assess the ability of NS163 to inhibit the

fibers catalyzed aggregation of α S. We have included the writeup and Figure (Supplementary Fig. 34) in the main manuscript, which reads as follows:

“We tested the effect of NS163 on the α S fiber-catalyzed aggregation of α S using ThT kinetic assay. The aggregation kinetics of 100 μ M α S in the presence of preformed α S fibers (20%, monomer eq.) resulted in the acceleration of monomeric α S aggregation (Supplementary Fig. 33, blue line), which is reflected by a significant increase in the ThT signal and the disappearance of the lag phase (Supplementary Fig. 33, black line). The α S fibers catalyzed aggregation of α S was suppressed by 78% by NS163 at an equimolar ratio, as evidenced by a lower ThT fluorescence signal (Supplementary Fig. 33, red line). The data demonstrate that NS163 is a potent inhibitor of *de novo* and fiber-catalyzed aggregation of α S. Therefore, we envision that NS163 will be effective in rescuing the degeneration of DA neurons in the post-disease onset model of PD in UA196 worms.”

Supplementary Fig. 33. Effect of NS163 on the seed catalyzed aggregation of α S. The aggregation kinetic profile of the aggregation of α S monomer (100 μ M, blue line), α S monomer (100 μ M) + α S fibers (20% monomer, black line), and α S monomer (100 μ M) + α S fibers (20% monomer) + NS163 (100 μ M, red line) in the aggregation buffer conditions. The aggregation experiments were conducted three times and the reported change in the ThT intensity was an average of three independent experiments. The data were expressed as mean and the error bars report the s.d. (n = 3 independent experiments).

*Collectively, we surmise that the effective rescue of the degeneration of DA neurons in the late-stage model of UA196 worms is due to the slow degeneration of DA neurons and the multifaceted ability of NS163 to inhibit α S aggregation via *de novo* and fibers catalyzed aggregation.*

Minor

Legends in Fig. 2 should be "t" and "u", but they are "k" and "l".

Response:

We have fixed the legends of Fig. 2 in the revised Fig. 2.

Reviewer #2 (Remarks to the Author):

The approach is novel, the results are overall sound, and the article will be of interest to those in the field. However, I think the authors should still address some critical aspects, which I delineate below.

Response:

We really appreciate the comments from the reviewer that our approach is novel and the results are sound.

A strength of this work is the possibility to attain a structure-function relationship according to the lateral groups each of the developed molecules bears. It is true that in the case of NS132 they generate variants with altered activity, but still, the most relevant information for the reader remains hidden in the set of different molecules analyzed in Fig 2b and ad Fig 2g. I would like to see a detailed analysis of what makes a molecule active and another not. Is it just the presence of one or more aromatic rings together with and hydrogen donor? Is it the sequential order? Is it the nature of the non-polar and polar groups? Seeing and analyzing what these molecules have in common and in what they differ will likely allow for the rational design of optimized molecules, which I think is a powerful feature of this large-scale approach that remains undisclosed in the present form of the work.

Response:

We agree with the reviewer and really appreciate the observation from the reviewer. In the revised manuscript we have included an explanation detailing the structure activity relationship (SAR) analysis of the antagonist activity of ligands (dipyridyls and tripyridyls) against α S aggregation. We have divided the functional groups of the dipyridyls and tripyridyls into various categories, including hydrophobic, positively charged groups, negatively charged groups, and polar groups and linked them with their antagonist activity against α S aggregation. The data has been included for dipyridyls (**Fig. 2e**) and tripyridyls (**Fig. 2k**) and the SAR explanation has been added to the main manuscript, which reads as follows:

Fig. 2e. The graphical representation of the ThT intensity of 100 μ M α S aggregation for four days in the absence and presence of dipyridyls at an equimolar ratio. The arrow indicates the most potent antagonist of α S aggregation. The ThT experiments were conducted three times and the reported change in the ThT intensity was an average of three separate experiments. The data were expressed as mean and the error bars report the s.d.

SAR study for dipyriddyls:

“We compared the antagonist activity of dipyriddyls using the ThT assay (and gel shift assay) to carry out the SAR study between dipyriddyls and α S. We observed various patterns between the antagonist activity and the chemical structure of the side chains on the second position of dipyriddyls (Fig. 2e). We first compared the hydrophobicity of the side chains of the dipyriddyls. For the most part, the antagonist activity of the dipyriddyls was directly related to the hydrophobicity of the side chains. The propyl side chain has the lowest antagonist activity and the antagonist activity for the most part was increased with the increase in the hydrophobicity of the side chains. The cyclohexyl group had the highest antagonist activity (NS55, Fig. 2e-h). The antagonist activity decreased with a higher hydrophobicity than cyclohexyl group, as demonstrated by the indole group (NS72, Fig. 2e). We also observed that the side chains with amines (aliphatic or aromatic) were detrimental to the antagonist activity against α S aggregation. The polar non-aromatic hydroxyl groups as side chains were effective antagonists of α S aggregation; however, the phenol group as a side chain was a moderate antagonist of α S aggregation. The negatively charged (carboxylic and sulphonic acid) side chains were very effective inhibitors of α S aggregation. The effective antagonist activity of the hydrophobic, negatively charged, and primary hydroxyl groups is because these dipyriddyls might be interacting with different regions of α S and inhibiting the aggregation. We will further explore these different regions of α S that are important for the aggregation and that study will be part of a separate manuscript. For the current study, we chose the cyclohexyl group as the second side chain to synthesize the tripyridyls because it demonstrated the highest antagonist activity against α S aggregation (Fig. 2e-h). We envision that the negatively charged (COOH group) and hydrophobic (cyclohexyl) side chains on the dimer interact with the positively charged (lysine) and hydrophobic groups on the N-terminus of α S.”

Fig. 2k. The graphical representation of the ThT intensity of 100 μ M α S aggregation for four days in the absence and presence tripyridyls at an equimolar ratio. The arrow indicates the most potent antagonist of α S aggregation. The ThT experiments were conducted three times and the reported change in the ThT intensity was an average of three separate experiments. The data were expressed as mean and the error bars report the s.d.

SAR study for tripyridyls:

“We compared the antagonist activity of tripyridyls using the ThT assay (and gel shift assay) to carry out a SAR study between tripyridyls and α S. We observed various patterns between the antagonist activity and the chemical structure of the side chains on the third position of tripyridyls (Fig. 2k). We first compared the antagonist activity of the hydrophobicity of the side chains of the tripyridyls. The tripyridyl with propyl side chain (NS123) demonstrated the lowest antagonist activity as it decreased the ThT signal from 100% to 72.7% (Fig. 2k). The tripyridyls with more hydrophobic groups, including NS127, NS126, and NS165 demonstrated moderate antagonist activity against α S aggregation as they decreased the ThT signal to 50.3%, 45.9%, 43.9%, respectively. The tripyridyl with the indole group (NS132) was the most potent antagonist as it decreased the ThT signal to 1% (Fig. 2k). The antagonist activity, for the most part, was increased with an increase in the hydrophobicity of the side chains in the tripyridyls except the trimer with an alkyne side chain (NS174), which decreased the ThT signal to 17.9% (Fig. 2k). One of the reasons could be that NS174 might be interacting with a different α S region than NS132 and inhibiting α S aggregation. It has been shown recently by us and others that there are multiple α S sequences that facilitate α S aggregation. We also observed that the tripyridyls with side chains as amines (aliphatic or aromatic) did not demonstrate any noticeable effect on α S aggregation, a pattern similar to the dipyridyls (Fig. 2e,k). The tripyridyls with carboxylic (NS130) and sulphonic (NS131) acids were moderate antagonists of α S aggregation as they decreased the ThT signal to 59.6% and 36.1%, respectively (Fig. 2k). The tripyridyls with aliphatic hydroxyl (NS166) and phenol (NS168) groups were potent antagonists of α S aggregation as they decreased the ThT signal to 15.5% and 2.7%, respectively (Fig. 2k). The tripyridyl, NS168 was a potent antagonist of α S aggregation as it decreased the ThT signal to 2.5%, close to NS132 (Fig. 2k). We speculate that NS168 might be interacting with a different sequence of α S than NS132. We will investigate the interaction of these tripyridyls with α S in detail and it will be presented in the near future.”

- I miss a concentration-dependent toxicity assessment for both cells and *C. elegans*. This is especially important in the latter case since compounds are added apparently early in the development stage, and an accommodation effect might occur; in such a way that only those worms that survive to the treatment are further analyzed, and of course, in this subpopulation, the compound would work efficiently. We have observed such an effect in our lab when looking for drugs for different applications.

Response:

*We agree that we had presented data for a single concentration of 10 μ M for cellular study and 50 μ M of NS163 with the nematode study. However, we had conducted a dose dependent study of NS163 against various cellular and nematode models and we presented the data for a single concentration of NS163 (50 μ M). We had carried out the dose dependent study, which was a part of another manuscript. We have now included the dose dependent study of NS163 (10 μ M, 25 μ M, and 50 μ M) for both cellular and *C. elegans* models as suggested by the reviewer. We have tested the effect of NS163 in a dose dependent manner (10 μ M and 25 μ M) in HEK cells for intracellular*

aS inclusions. We have also tested the effect of NS163 in a dose dependent manner (10 μ M, 25 μ M, and 50 μ M) in NL5901 for *aS* inclusions (day 8 and day 9, **Supplementary Fig. 21**) and motility (day 7,8, and 9, **Supplementary Fig. 23**). We also carried out a dose dependent study of NS163 to assess its effect on the degeneration of DA neurons in UA196 worms in a late-stage onset disease model (**Supplementary Fig. 34**). The Figures (**Supplementary Fig. 21, 23, 34**) and writeup for these experiments have been included in the main manuscript, which reads as follows:

Supplementary Fig. 18. a, Assessment of cell permeability of the indicated ligands using the PAMPA. Confocal images (**b**) and statistical analysis (**c**) of HEK cells treated with the aggregated solution of 5 μ M α S, followed by the treatment with the indicated ligands (10 μ M). Inclusions of α SA53T-YFP = white arrows, Hoechst = blue, α S-ps-129 = red, merge = Hoechst, α S-ps-129, and α SA53T-YFP. The PAMPA assays were conducted three independent times with three technical replicates. The Proteostat assays were conducted with at least four biological replicates and three technical replicates for each biological replicate. The data were expressed as mean and the error bars report the s.e.m. (n = 3-4 independent experiments and each n consisted of three technical replicates). The statistical analysis was performed using ANOVA with Tukey's multiple comparison test. *p < 0.05, **p < 0.01, ***p < 0.001, ****p < 0.0001.

“To test the antagonist activity of NS163 against intracellular α S aggregation, we utilized HEK293 cells, which stably express α S-A53T-YFP^{23,24,50}. The endogenous monomeric α S-A53T-YFP template into fibers when transfected with exogenously added α S fibers in the presence of Lipofectamine 3000 (Fig. 3a)^{23,24,50}. In this assay, preformed fibers of α S (5 μ M monomer concentration) were introduced to the HEK cells, followed by the introduction of NS163 to the cells after 8h, which is the time required for the internalization of α S fibers, as we

have shown recently. The HEK cells were incubated for an additional 18 h, which is the total time (24 h) required for the complete maturation of α S inclusions in the HEK cells. To test the effect of the NS163 on the intracellular inclusions (α S- A_{53T} -YFP), we used confocal imaging and a ProteoStat dye-based assay recently developed in our lab²⁴. In comparison to the control (Fig. 4g, control), there was an abundance of intracellular inclusions in the presence of α S fibers (Fig. 4g, + α S fibers). In contrast, there was a significant reduction in inclusions in the presence of 10 μ M NS163 (Fig. 4g, +NS163). The number of inclusions was even further reduced by NS163 at 25 μ M concentration, very close to the control (Fig. 4g,h, +NS163). The ProteoStat dye-based intensity of HEK cells treated with α S fibers was \sim 5 fold higher than the control (no fibers) (Fig. 4h, + α S fibers). NS163 was able to reduce the intracellular inclusions of α S by 76% and 90% at 10 μ M and 25 μ M, respectively, measured using the ProteoStat dye-based assay (Fig. 4h, +NS163). NS163 inhibits the formation of α S inclusions in HEK cells in a dose dependent manner.”

Supplementary Fig. 21. The comparison of the number of intracellular α S inclusions in NL5901 worms in the absence and presence of the indicated doses of NS163 on day eight (blue bar) and nine (red bar). For each confocal imaging experiment, at least 5 worms were used and the inclusions were counted manually, and each condition (each day) consisted of three independent experiments. The data were expressed as mean and the error bars report the s.e.m. (n = 3 independent experiments and each n consisted of a minimum of 5 technical replicates). The statistical analysis was performed using ANOVA with Tukey’s multiple comparison test. *p < 0.05, **p < 0.01, ***p < 0.001, ****p < 0.0001.

“We also carried out a dose-dependent study of NS163 against the intracellular α S inclusions in NL5901 worms. The number of α S inclusions in NL5901 worms decreased from \sim 34/32 (day 8/9) to \sim 19/20 (day 8/9) and \sim 9/10 (day 8/9) at 10 and 25 μ M NS163, respectively (Supplementary Fig. 21). The data suggest that NS163 inhibits α S inclusions in NL5901 worms in a dose-dependent manner (Supplementary Fig. 21).”

Supplementary Fig. 23. The statistics for the relative motility rate of N2 and NL5901 in the absence and presence of various doses of NS163 on the indicated days. For motility experiments, a total of 50 worms were used in duplicate for each experiment and each condition consisted of at least four independent experiments. The data were expressed as mean and the error bars report the s.e.m. (n = 4 independent experiments and each n consisted of a minimum of two technical replicates). The statistical analysis was performed using ANOVA with Tukey’s multiple comparison test. *p < 0.05, **p < 0.01, ***p < 0.001, ****p < 0.0001.

“We also carried out a dose-dependent study of NS163 against the motility of NL5901 worms. We observed a noticeable increase in the motility of NL5901 worms from 10 to 25 μ M NS163 (Supplementary Fig. 23). The data suggests that NS163 improves the motility of NL5901 worms in a dose-dependent manner (Supplementary Fig. 23).”

Supplementary Fig. 34. The relative number of healthy DA neurons in UA196 worms during the aging process when treated on day five with the indicated doses of NS163. For each confocal imaging experiment, at least 10 worms were used, and the healthy neurons (GFP signal) were counted manually, and each condition (day) consisted of six independent experiments (total of 60 DA neurons). The data were expressed as mean and the error bars report the s.e.m. (n = 6 independent experiments and each n consisted of 10 technical replicates).

“NS163 was also effective in rescuing the degeneration of DA neurons at lower doses as the number of intact healthy DA neurons was 34/27 (25 μ M/10 μ M) and 27/17 (25 μ M/10 μ M) on day 10 and day 15, respectively (Supplementary Fig. 34). Clearly, NS163 was potent in stopping the further degeneration of DA neurons when added on day five to UA196 worms as a post-disease onset PD model.”

Overall, we have shown that NS163 is a potent antagonist of α S aggregation in a dose dependent manner in both cellular and C. elegans models.

- A problem with the approach to bringing it to the clinic is that it targets the monomeric protein and not the toxic forms of the protein, oligomers, and fibrils, both in the unseeded and seeded assay. Blocking the monomeric protein would also block its function and might have undesired side effects. The authors should elaborate on this. Moreover, it will be interesting to demonstrate whether the molecules bind to fibrils themselves and if they can disaggregate them eventually. These assays are easy to perform and would be informative on the therapeutic potential of the molecules.

*It has been suggested that α S exists in multiple native forms including monomeric, tetrameric, and a multimeric membrane bound state on the synaptic vesicles^{13,16,17}. Moreover, it has been suggested that the interaction between α S and the synaptic vesicles is a key feature to regulate its proposed biological functions, which are the neurotransmitter release and synaptic plasticity¹²⁻¹⁸. Also, the binding affinity between the synaptic vesicles and the monomeric α S protein (to regulate its function) is ~300-500 nM. We have shown that NS163 binds with the α S monomeric form with a K_d of 1.8 μ M. Therefore, the binding affinity between synaptic vesicles and α S is ~five fold higher than the binding affinity between NS163 and α S. Therefore, we surmise that NS163 will not disrupt the monomeric α S-synaptic vesicle interaction and, therefore NS163 may not disrupt the physiological function of α S. Also, we have shown in the revised manuscript that NS163 not only inhibit the *de novo* aggregation of α S, but also inhibits the fibers catalyzed aggregation of α S (Supplementary Fig. 33), which is another important mechanism to potentiate toxicity in neurons. Collectively, we surmise that NS163 inhibits the aggregation (*de novo* and fibers catalyzed) of α S via multiple mechanisms without disrupting the physiological function of α S. We have added this writeup in the discussion in the main manuscript, which reads as follows:*

“We surmise that NS163 will not likely disrupt the physiological function of α S and potently inhibit α S aggregation multiple mechanisms (*de novo* and fibers catalyzed). It has been suggested that α S exists in multiple native forms including monomeric, tetrameric, and a multimeric membrane-bound state on the synaptic vesicles^{13,16,17}. Moreover, it has been suggested that the interaction between α S and the synaptic vesicles is a key feature in regulating its proposed biological functions, which are neurotransmitter release and synaptic plasticity¹²⁻¹⁸. Also, the binding affinity between the synaptic vesicles and the monomeric α S protein (to regulate its function) is ~300-500 nM^{22,24,47}. We have shown that NS163 binds to the monomeric form of α S with a K_d of 1.8 μ M. Therefore, the binding affinity between synaptic vesicles and α S is ~five fold higher than the binding

affinity between NS163 and α S. Also, we have shown that NS163 not only inhibits the *de novo* aggregation of α S, but also inhibits the fiber-catalyzed aggregation of α S, which is another important mechanism to potentiate toxicity in neurons. Collectively, NS163 inhibits the aggregation (*de novo* and fibers catalyzed) of α S via multiple mechanisms. Therefore, we surmise that NS163 will not disrupt the monomeric α S-synaptic vesicle interaction nor disrupt the physiological function of α S.”

As suggested by the reviewer, we also tested the effect of NS163 on the preformed fibrils of α S.

*A solution of preformed fibers of α S (100 μ M, monomer concentration) in 1 \times PBS (pH 6.5) was treated with NS163 at an equimolar ratio. The solution was incubated for 24 h and then we added the ThT solution (50 μ M) and then checked the fluorescence intensity (ThT dye) of α S fibers in the absence and presence of NS163. We did not observe any significant difference in fluorescence intensity (ThT dye) of α S fibers in the absence and presence of NS163 at an equimolar ratio (**Supplementary Fig. 16a**). Additionally, we used TEM to further validate the results from the ThT assay. The TEM images demonstrated that the morphology and the amount of α S fibers were similar in the absence and presence of NS163 at an equimolar ratio (Supplementary Fig. 16b,c). We have added the data (**Supplementary Fig. 16a-c**) and the writeup to the main manuscript, which now reads as follows:*

“We also tested the effect of NS163 on the preformed fibers of α S. A solution of preformed fibers of α S (100 μ M, monomer concentration) in 1 \times PBS buffer (pH 6.5) was treated with NS163 at an equimolar ratio. The solution was incubated for 24 h, followed by the addition of ThT solution (50 μ M) and then checked the fluorescence intensity (ThT dye) of α S fibers in the absence and presence of NS163. We did not observe any significant difference in fluorescence intensity (ThT dye) of α S fibers in the absence and presence of NS163 at an equimolar ratio (Supplementary Fig. 16a). Additionally, we used TEM to further validate the results from the ThT assay. The TEM images of the preformed fibers of α S demonstrated that the morphology and the amount of α S fibers were similar in the absence (Supplementary Fig. 16b) and presence of NS163 (24 h incubation, Supplementary Fig. 16c) at an equimolar ratio. These results indicate that NS163 does not have any noticeable effect on the preformed fibers of α S.”

Supplementary Fig. 16. a, The comparison of the fluorescence intensity of ThT dye (50 μ M) of preformed 100 μ M α S fibers (1 \times PBS buffer, pH 6.5) in the absence and presence of NS163 at an equimolar ratio. The representative TEM images of preformed 100 μ M α S fibers (b) treated with NS163 for 24 h (c) at an equimolar

ratio. The ThT and TEM experiments were conducted three times and the reported change in the ThT intensity was an average of three separate experiments. The data were expressed as mean and the error bars report the s.d. ($n = 3$ independent experiments).

- In the cellular assays, what I think is an important experiment is missing. Do cells treated with the compound previous to fibril insult become resistant to intracellular seeding? This is indeed the experiment that would recapitulate what occurs in vivo. The assays shown by the authors are excellent but only inform on the activity of the molecules pre-cell treatment.

Response:

We agree with reviewer that we did not perform an experiment to see if the compounds can inhibit the intracellular aggregation of α S templated by exogenously added α S fibers. As suggested by the reviewer, we have performed two HEK cell-based experiments to test multiple tripyridyls for their ability to inhibit the intracellular α S aggregation templated by exogenously added α S fibers. We have included these experiments and Figures (Fig. 4g,h; Supplementary Fig. 18a-c, 19) in the manuscript, which reads as below:

Fig. 4. The intracellular inhibition of α S aggregation by OPs in PD models. a, Assessment of cell permeability of the indicated ligands using the PAMPA. **b**, The chemical structures of NS132 and NS163. **c**, The aggregation profile of 100 μ M α S in the absence and presence of the indicated ligands at an equimolar ratio. **d**, The ITC

thermogram for the titration of a solution of NS163 into α S where heat burst curves and the corrected injection heats are represented by the upper panel and the lower panel, respectively. **e**, The relative volume changes of the backbone amide peaks of ^{15}N -labeled α S ($70\ \mu\text{M}$) in the presence of NS163 at an equimolar ratio. The dashed line represents the reported volume changes in ^{15}N -labeled α S residues peaks in the presence of NS132 above the value of 5%. The colored sequences (blue and green) are the potential binding sites of NS163 on α S. **f**, The comparison of the aggregation profile of $100\ \mu\text{M}$ α S in the absence and presence of the indicated ligands at an equimolar ratio. Confocal images (**g**) and statistical analysis (**h**) of HEK cells treated with the aggregated solution of $5\ \mu\text{M}$ α S, followed by the treatment with the indicated ligands ($10\ \mu\text{M}$). Inclusions of $\alpha\text{SA}_{53\text{T}}\text{-YFP}$ = white arrows, Hoechst = blue, α S-ps-129 = red, merge = Hoechst, α S-ps-129, and $\alpha\text{SA}_{53\text{T}}\text{-YFP}$. The representative confocal images of NL5901 worms with α S inclusions (white arrows) in the body wall muscle cells (Day 8) in the absence (**i**) and presence of $50\ \mu\text{M}$ NS132 (**j**) and $50\ \mu\text{M}$ NS163 (**k**). **l**, The number of inclusions for experiment in NL5901 in the absence (blue bar) and presence of $50\ \mu\text{M}$ NS132 (orange bar) and $50\ \mu\text{M}$ NS163 (red bar) from day four to day nine. **m**, The motility of N2 (green bar) and NL5901 and statistics (Day 8, **n**) in the absence (blue bar) and presence (red bar) of $50\ \mu\text{M}$ NS163 during the aging process. The ThT experiments were conducted three times and the reported change in the ThT intensity was an average of three independent experiments and the error bars report the s.d. For each confocal imaging experiment, at least 10 worms were used, and the inclusions were counted manually, and each condition (Each day) consisted of at least four independent experiments. For motility experiments, a total of 50 worms were used in duplicate for each experiment and each condition consisted of at least four independent experiments. The data were expressed as mean and the error bars report the s.e.m. ($n = 3$ or 4 independent experiments and each n consisted of a minimum of three technical replicates). The statistical analysis was performed using ANOVA with Tukey's multiple comparison test. * $p < 0.05$, ** $p < 0.01$, *** $p < 0.001$.

Supplementary Fig. 18. **a**, Assessment of cell permeability of the indicated ligands using the PAMPA. Confocal images (**b**) and statistical analysis (**c**) of HEK cells treated with the aggregated solution of $5\ \mu\text{M}$ α S, followed by

the treatment with the indicated ligands (10 μ M). Inclusions of α S_{A53T}-YFP = white arrows, Hoechst = blue, α S-ps-129 = red, merge = Hoechst, α S-ps-129, and α S_{A53T}-YFP. The PAMPA assays were conducted three independent times with three technical replicates. The Proteostat assays were conducted with at least four biological replicates and three technical replicates for each biological replicate. The data were expressed as mean and the error bars report the s.e.m. (n = 3-4 independent experiments and each n consisted of three technical replicates). The statistical analysis was performed using ANOVA with Tukey's multiple comparison test. *p < 0.05, **p < 0.01, ***p < 0.001, ****p < 0.0001.

“To test the antagonist activity of NS163 against intracellular α S aggregation, we utilized HEK293 cells, which stably express α S-A53T-YFP^{23,24,50}. The endogenous monomeric α S-A53T-YFP template into fibers when transfected with exogenously added α S fibers in the presence of Lipofectamine 3000 (Fig. 3a)^{23,24,50}. In this assay, preformed fibers of α S (5 μ M monomer concentration) were introduced to the HEK cells, followed by the introduction of NS163 to the cells after 8h, which is the time required for the internalization of α S fibers, as we have shown recently. The HEK cells were incubated for an additional 18 h, which is the total time (24 h) required for the complete maturation of α S inclusions in the HEK cells. To test the effect of the NS163 on the intracellular inclusions (α S-A53T-YFP), we used confocal imaging and a ProteoStat dye-based assay recently developed in our lab²⁴. In comparison to the control (Fig. 4g, control), there was an abundance of intracellular inclusions in the presence of α S fibers (Fig. 4g, + α S fibers). In contrast, there was a significant reduction in inclusions in the presence of 10 μ M NS163 (Fig. 4g, +NS163). The number of inclusions was even further reduced by NS163 at 25 μ M concentration, very close to the control (Fig. 4g,h, +NS163). The ProteoStat dye-based intensity of HEK cells treated with α S fibers was ~5 fold higher than the control (no fibers) (Fig. 4h, + α S fibers). NS163 was able to reduce the intracellular inclusions of α S by 76% and 90% at 10 μ M and 25 μ M, respectively, measured using the ProteoStat dye-based assay (Fig. 4h, +NS163). NS163 inhibits the formation of α S inclusions in HEK cells in a dose dependent manner. NS132 was also able to inhibit the formation of inclusions by 47%, which is less than NS163 (Fig. 4g,h +NS132). It is likely to be a consequence of the better cell permeability of NS163 than NS132. We also compared other tripyridyls with NS163 for their ability to inhibit intracellular α S aggregation using this assay. We used multiple tripyridyls (NS127, NS131, NS132, and NS163) with varying antagonist ability to inhibit α S aggregation. To test their antagonist activity against intracellular α S aggregation, we first determined the cell permeability of the tripyridyls using the PAMPA (Fig. 4a, Supplementary Fig. 18a). The increasing order of the cell permeability was NS131<NS127<NS132<NS163 (Fig. 4a, Supplementary Fig. 18a). The cell permeability of NS131 was the lowest, most likely because of the sulfonic acid side chain. The cell permeability of NS127 and NS132 was very similar because of the same COOH group and hydrophobic groups (Fig. 4a, Supplementary Fig. 18a). The cell permeability of NS163 was highest because the COOH group was replaced with hydroxylamine group (Fig. 4a). We utilized the HEK293 cells-based assay as we used for NS163 and NS132 to determine the antagonist activity of other tripyridyls against intracellular α S aggregation. To test the tripyridyls, we first incubated the HEK cells with preformed α S fibers (5 μ M, monomer concentration) for 8 h. Subsequently, we added tripyridyls (10 μ M) to the HEK cells and then we incubated the cells for additional 18 h, which is the total

time (24 h) required for the complete maturation of α S inclusions in the HEK cells. We used confocal imaging and the ProteoStat dye-based assay to monitor the effect of tripyridyls on intracellular α S inclusions (Supplementary Fig. 18a-c). We observed a weak antagonist effect of NS131 (8% decrease in inclusions) on the intracellular α S aggregation even though it was a good antagonist of α S aggregation (from ThT, Fig. 2k, Supplementary Fig. 18b,c). The poor intracellular antagonist activity of NS131 is most likely because of very poor cell permeability (Supplementary Fig. 18a). We observed a noticeable inhibition (~20% decrease in inclusions) of the intracellular aggregation of α S in the presence of NS127 (Supplementary Fig. 18b,c). NS132 (47% decrease in inclusions) was a much better antagonist than NS127 of the intracellular α S aggregation (Fig. 4g,h, Supplementary Fig. 18b-c). It is worth noting that the cell permeability of NS127 was comparable to NS132 (Fig. 4a, Supplementary Fig. 18a). The ThT assay also validated these results that NS132 is a far more potent antagonist of α S aggregation (Fig. 2k). NS163 was the most potent antagonist of the intracellular α S aggregation (76% decrease in inclusions) among the tripyridyls (Fig. 4g,h, Supplementary Fig. 18b-c). Both NS163 and NS132 were comparable in the inhibition of α S aggregation from ThT and gel shift assays; however, the cell permeability of NS163 is better than NS132 (Fig. 4a). These results demonstrate that higher cell permeability and antagonist activity together allow for better antagonist activity against intracellular aggregation of α S.”

Supplementary Fig. 19. The relative intensity of Proteostat dye-stained intracellular inclusions of α S_{A53T}-YFP after treatment with the indicated ligands (10 μ M) followed by the treatment with α S fibers (5 μ M monomer conc.) for 24 h. The Proteostat assays were conducted with at least four biological replicates and three technical replicates for each biological replicate. The data were expressed as mean and the error bars report the s.e.m. (n = 4 independent experiments and each n consisted of three technical replicates). The statistical analysis was performed using ANOVA with Tukey’s multiple comparison test. *p < 0.05, **p < 0.01, ***p < 0.001, ****p < 0.0001.

“We further tested NS163 against intracellular α S aggregation using another HEK cell-based assay. For this assay, we utilized the same HEK293 cells, which stably express YFP-labeled α S-A53T mutant (α S-A53T-YFP)^{23,24,50}. In this assay, we added NS163 (10 μ M) to the HEK cells and incubated the cells for 16 h to allow the internalization

of NS163. Subsequently, the cells were washed with 1×PBS (3 times) and treated with the preformed fibers of α S (5 μ M monomer concentration). The HEK cells were incubated for an additional 24 h, which is the total time (24 h) required for the complete maturation of α S inclusions in the HEK cells. The ProteoStat dye-based intensity of HEK cells treated with α S fibers was \sim 4 fold higher than the control (no fibers) (Supplementary Fig. 19). NS163 was a very potent antagonist as it was able to reduce the intracellular inclusions by 80% at 10 μ M, measured using the ProteoStat dye-based assay (Supplementary Fig. 19). Under matched conditions, 10 μ M NS132 was able to reduce the intracellular inclusions by 57% (Supplementary Fig. 19). The higher antagonist activity of NS163 than NS132 is likely due to the better cell permeability of the former. Collectively, NS163 is a potent inhibitor of α S fiber-catalyzed aggregation of α S in the cellular milieu.”

- In seeding experiments, authors show the relative values of the Th-T signal at 24 h (Fig 3a). However, this representation is not very informative, and it should be shown the aggregation kinetics in the presence of seeds and the presence and absence of the compound, which are the conventional experiments to show, as the authors have done in panel 2 for unseeded reactions. In the same manner, TEM images of the endpoint of the reactions would be informative of the actual potency of the molecules.

We agree with the reviewer’s comment that the aggregation kinetics of the ThT assay should be more informative. We have now repeated and included the complete aggregation kinetics of the ThT assay of the fibers catalyzed aggregation of α S in the absence and presence of NS132 (Fig. 3e). We have included both the ThT aggregation kinetics (Fig. 3e) and TEM images (Supplementary Fig. 12) We have replaced Fig. 3a with the modified figure (Fig. 3e). We have included this data in the main manuscript (Fig. 3e) and supplementary information (Supplementary Fig. 12) and a writeup was included in the main manuscript as:

Fig. 3e. The ThT based aggregation kinetics of 100 μ M α S catalyzed by α S fibers (20% monomer conc.) in the absence and presence of NS132 at an equimolar ratio. The ThT experiments were conducted three times and the reported change in the ThT intensity was an average of three independent experiments and the error bars report the s.d. (n = 3 independent experiments and each n consisted of three technical replicates).

Supplementary Fig. 12. The TEM images of the preformed α S fibers (20%, monomer concentration) catalyzed aggregation of 100 μ M α S in the absence (a) and presence (b) of NS132 at an equimolar ratio in 1 \times PBS buffer (pH 6.5).

“Therefore, we monitored the effect of NS132 on the α S fiber-catalyzed aggregation of α S using ThT kinetic assay. The aggregation kinetics of 100 μ M α S in the presence of preformed α S fibers (20%, monomer concentration) resulted in the acceleration of monomeric α S aggregation (Fig. 3e, α S, blue line), which is reflected by a significant increase in the ThT signal and the disappearance of the lag phase (Fig. 3e, + α S fibers, black line). The α S fiber-catalyzed aggregation of α S was wholly suppressed by NS132 at an equimolar ratio, as evidenced by a significantly lower ThT signal (Fig. 3e, + α S fibers+NS132, orange line). The TEM data also corroborate well with the ThT assay. We used the solutions from the ThT assay at 60 h for the TEM experiment. We observed an abundance of α S fibers in the fiber-catalyzed aggregation of α S (Supplementary Fig. 12a). In marked contrast, significantly fewer α S fibers were observed in the presence of NS132 at an equimolar ratio (Supplementary Fig. 12b).

- It turns out that NS163 is the most potent of the two assayed molecules in worms and likely a hit compound for further studies, this higher activity being attributed to a potential higher permeability. However, I found it strange that the binding affinity of this compound to a-synuclein is not shown, and the confirmation by NMR that it acts strictly as NS132, binding precisely the same N-terminal regions is not provided. In the absence of these data is hard to ascertain that both compounds act in the same manner and that the difference is only due to permeability issues. It will be really nice to have access to these studies.

We agree that we did not include the binding characterization of NS163 with α S. To further confirm that the binding behavior of NS163 is similar to NS132 for α S, we carried out ITC titration and 2D NMR spectroscopy between NS163 and α S. We have included these experiments and Figures (Fig. 4d,e; Supplementary Fig. 15) in the manuscript and a writeup, which reads as below:

Fig. 4d. The ITC thermogram for the titration of a solution of NS163 into α S where heat burst curves and the corrected injection heats are represented by the upper panel and the lower panel, respectively. The fit for the curve (teal line) yielded the binding and thermodynamic parameters between NS163 and α S.

Fig. 4e. The relative volume changes of the backbone amide peaks of ^{15}N -labeled α S (70 μM) in the presence of NS163 at an equimolar ratio. The dashed line represents the reported volume changes in ^{15}N -labeled α S residues peaks in the presence of NS132 above the value of 5%. The colored sequences (blue and green) are the potential binding sites of NS163 on α S.

Supplementary Fig. 15. Overlay of 2D HSQC (^1H , ^{15}N) NMR spectra of 70 μM uniformly ^{15}N -labelled αS in the absence (red) and presence (blue) of NS163 at an equimolar ratio in $1 \times \text{PBS}$ buffer (pH 6.5).

“We used ITC and HSQC experiments to characterize and compare the binding affinity and binding site of NS163 with NS132 against αS . Under exact conditions to NS132, the K_d of NS163 for αS was $2.11 \pm 0.36 \mu\text{M}$ with a binding stoichiometry of 1.03 ± 0.04 , which was in close agreement of NS132 ($K_d = 1.81 \pm 0.33 \mu\text{M}$) (Fig. 4d). We collected the HSQC NMR of 70 μM ^{15}N - ^1H -uniformly labeled αS in the absence (Fig. 4e, Supplementary Fig. 15, red) and presence of 70 μM NS163 (Fig. 4e, Supplementary Fig. 15, blue) and compared the change in the signal intensity (volume) of the amide peaks of NS163 with NS132. In the presence of NS163, we observed noticeable volume changes in the amide peaks for specific residues toward the N-terminus, including 9-21 and 36-43 residues (Fig. 4e, blue and green). NS163 also demonstrated changes in the volume intensity of some residues toward the N-terminus of αS (up to 100 residues) (Fig. 4e). It is important to note that the binding site of

NS163 was in close proximity to the binding site of NS132 on α S (Fig. 2u and Fig. 4e). Clearly, the PAMPA assay, ThT assay, TEM, ITC, and HSQC NMR study demonstrate that we have improved the cell permeability of NS132 (in NS163) without sacrificing the antagonist activity, binding affinity, and binding site for α S.”

- A last comment is that the *C.elegans* experiments should be explained in more detail. In the way they are described, it is difficult to decipher when and how the compounds are administered since there are no references to the development stages of the animals (commonly, it is referenced from stage L4), for an example of similar work with a more conventional description of the moment at which the drug is administered and the time at which the phenotype is analyzed, please see doi.org/10.1073/pnas.1610586114. Moreover, the fact that the animals are cultured at 23-25 C instead of the conventional 20 C, makes it difficult to compare the development stages with those of related studies. These details are important and should be clarified since one of the authors' claims is that the compound acts in both early and late stages. How does the state of their animals at these time points compare with those of other compounds tested in the literature?

We apologize for not providing detailed method information about the culture of the C. elegans models. We cultured the worms at 20 °C and the temperature and humidity were constant in the incubator throughout the experiment. For various worm experiments (for both NL5901 and UA196 worms) that were carried out in the liquid media, the worm plates were shaken on a shaker (rpm = 100, orbital) throughout the course of the experiment. For all worm experiments, various ligands were added on day two and four before each study. We have now modified the protocol for the worm experiments by including more detailed procedures. We have revised the detailed experimental procedure for C. elegans experiments, and it is included in the supplementary information, which reads as below:

C. elegans Experiments

Media and buffers for various worm assays.

The worms were maintained at standard conditions (20 °C at all times) on nematode growth media (NGM) agar in 60 mm plates (CytoOne, Ocala, FL) using *E. coli* OP50 strain as the food source following the previous protocols^{4,7,8}. To ensure that worm's colonies don't starve, they were always transferred on new plates with *E. coli* OP50 strain as the food source. All worm strains were replaced with new worm colony after every six months to avoid any genetic mutation, which might affect the disease phenotypes. NGM agar plates, M9 buffer (3 g KH₂PO₄, 6 g Na₂HPO₄, 5 g NaCl, 1 mL 1 M MgSO₄, milli-Q H₂O to 1L), Chemotaxis (CTX) media plates (2% Agar, 5 mM KHPO₄, 1 mM CaCl₂, and 1 mM MgSO₄), and CTX buffer (5 mM KH₂PO₄ 1 mM CaCl₂, and 1 mM MgSO₄) were prepared using previous protocols^{4,8-10}

Strains

The N2 (wild-type *C. elegans* Bristol strain), NL5901 (*C. elegans* model of PD), and *Escherichia coli* OP50 (*E. coli*, a uracil requiring mutant) strains were obtained from Caenorhabditis Genomics Center (CGC, Minneapolis, MN). The UA196 strain was generously donated by the laboratory of Dr. Guy Caldwell (Department of Biological Science, The University of Alabama, Tuscaloosa, AL, United States)⁶.

NL5901 strain (pkIs2386 [unc-54p::alphasynuclein::YFP+unc-119(+)]). In NL5901 strain, α S-YFP is expressed in the muscle cells of worms. The aggregates of α S-YFP are visible from day 4 and the maximum number of α S-YFP aggregates are visible round day 8 and day 9.

UA196 strain [$P_{dat-1}::\alpha\text{-syn}+P_{dat-1}::\text{GFP}$]. In UA196 strain, human αS and GFP are expressed in DA neurons under control of a dopamine transporter-specific promoter [$P_{dat-1}::\alpha\text{-syn}+P_{dat-1}::\text{GFP}$], which results in age-dependent neurodegeneration of six DA neurons.

Culture methods for *C. elegans* strains

The standard worm conditions were used for the culturing of various worm strains. Briefly, the worms were bleached and synchronized using hypochlorite solution, followed by the incubation of the eggs (at 20 °C) on NGM plates (35 mm, CellTreat Scientific, Pepperell, MA), which were seeded with OP50 (350 μL , 0.5 OD_{600nm}) as a food source, **referred to as PLATE 1**. The cultures of OP50 were prepared by incubating 50 mL of LB medium with OP50 18 h at 37 °C and the final OD value was adjusted to 0.5 at 600 nm. The NGM plates were prepared by treating them with 350 μL OP50 and leaving the plates at 20°C for 3 days. On day two, the worms were transferred (using M9 buffer) to NGM plates (35 mm, CellTreat Scientific, Pepperell, MA) containing 75 μM Fluorodeoxyuridine (FUDR; to prevent worm reproduction and ensure that equal ages of worms were used for the experiment)^{4,8} and OP50 as food source, **referred to as PLATE 2**. These NGM plates were prepared using autoclaved NGM media supplemented with 75 μM FUDR (2 mL total liquid). After 12 h, the NGM plates were seeded with 350 μL of OP50 (with OD = 0.5 at 600 nm) at 20 °C. For treating various worm strains with ligands, the worms were transferred on day 2 to the NGM plates treated with ligands (please see below to prepare the NGM plates treated with ligands).

Preparation of NGM plates with Ligands

The NGM plates (with NGM media) were used for the treatment of different worm strains with ligands. The NGM media containing 75 μM FUDR was autoclaved and poured in NGM plates (2 mL total liquid). After 12 h, the NGM plates were seeded with 350 μL of OP50 (with OD = 0.5 at 600 nm) at 20 °C. After 12 h, different doses of ligands (10-50 μM , stock conc. = 10 mM in DMSO, DMSO from 0.1-0.5%, v/v) were dissolved in M9 buffer (total volume = 300 μL) and were spotted atop the NGM plates at 20 °C, **referred to as PLATE 3**. The NGM plates treated with the ligands were placed in sterile laminar flow hood at 20 °C for 1 h. The NGM plates treated with ligands were prepared and used within 24 h. We used these plates to treat various disease strains for different assays.

Motility assay for *C. elegans* (N2, NL5901 and UA196)

Briefly, all worm strains were bleached at the same time and the eggs were incubated on **PLATE 1**. On day two, the worms were divided into two batches. One half of the worms were transferred (using M9 buffer) on **PLATE 2 (without ligand)** and the second half of the worms were transferred to **PLATE 3 (with ligand)**. The concentration of the ligands varied from 10-50 μM as described in the main manuscript. The worms were incubated on **PLATE 2 or PLATE 3** up to day 4 at 20 °C in an incubator with constant humidity. On day four, various worm strains (with and without ligands from PLATE 2 or PLATE 3) were transferred to sterile 24 well plate (CellTreat Scientific, Pepperell, MA) containing liquid media (500 μL /well), **referred to as PLATE 4A** and containing liquid media with ligands (500 μL /well with 10-50 μM ligand in M9 buffer with 0.1%-0.5% DMSO, v/v), **referred to as PLATE 4B**. The liquid media for **PLATE 4A/B** was prepared with 67.28% (v/v) of M9 buffer, 75 μM FUDR, 0.1% of 1 M magnesium sulfate (v/v), 0.1% of 1 M calcium chloride (v/v), 2.5% of 1 M potassium phosphate solution (pH 6, v/v), and 30% (v/v) of OP50 (0.5 OD_{600nm}). A total of 50 worms per well were manually transferred into **PLATE 4A/B** at 20 °C and a total of four wells (4 technical replicates) were used for each condition. Subsequently, the worms were incubated for 6 h at 20 °C with constant shaking (rpm = 100) to get acclimated with the solution conditions before starting the motility assay experiment.

To test the effect of each ligand on disease worm strains (NL5901 or UA196), there were four conditions were used in **PLATE 4A/B**: (1) N2 worms, (2) N2 worms treated with ligand, (3) NL5901 (or UA196) worms, and (4) NL5901 (or UA196) worms treated with the ligand (10-50 μM). The assay was started on day four using the WMicroTracker ARENA plate reader (Phylumtech, Santa Fe, Argentina) at 20 °C for 1 h per day over a 14-day period at intervals of 24 h. Each day, a total of 20 activity scores per well were collected in 1 h. After collecting the data each day, the **PLATE 4A/B** was again placed on the shaker (100 rpm) at 20 °C in the incubator. The **PLATE 4A/B** was on the on the shaker (100 rpm) at 20 °C in the incubator the whole duration of the experiment except only when the reading was collected on the WMicroTracker ARENA plate reader for 1 h. For each condition, four biological replicates were performed, and each biological replicate consisted of four technical replicates (four wells). For each condition, the data were expressed as mean and the error bars report the s.e.m. (n = 4 independent experiments and each n consisted of a total of four technical replicates).

Motility assay for *C. elegans* (N2 and UA196) in the presence of dopamine and NS163

This motility assay was performed exactly similar to the previous assay with slight adjustment in the preparation of the plates for the treatment of UA196 worms. For this assay, the only difference is the 24 well plates were prepared with 2 mM dopamine (**referred to as PLATE 5**) and both 2 mM dopamine and 50 μM NS163 (**referred to as PLATE 6**). To prepare **PLATE 5**, the NGM media containing 75 μM FUDR was autoclaved and poured in NGM plates (2 mL total liquid). After 12 h, the NGM plates were seeded with 350 μL of OP50 (with OD = 0.5 at 600 nm) at 20 °C. After 12 h, 2 mM dopamine dissolved in M9 buffer (total volume = 300 μL) was spotted atop the NGM plates at 20 °C. To prepare **PLATE 6**, the NGM media containing 75 μM FUDR was autoclaved and poured in NGM plates (2 mL total liquid). After 12 h, the NGM plates were seeded with 350 μL of OP50 (with OD = 0.5 at 600 nm) at 20 °C. After 12 h, NS163 (50 μM , stock conc. = 10 mM in DMSO) and dopamine (2 mM in M9 buffer) were dissolved in M9 buffer (total volume = 300 μL) and were spotted atop the NGM plates at 20 °C. The NGM plates treated with dopamine and NS163 and dopamine were placed in sterile laminar flow hood at 20 °C for 1 h. The NGM plates treated with dopamine and NS163 and dopamine were prepared and used within 24 h.

Similar to the previous section, the motility assay was performed for N2 and UA196 worms in the absence and presence of NS163+dopamine in sterile 24 well plate (CellTreat Scientific, Pepperell, MA) containing liquid media (500 μL /well), liquid media + 2 mM dopamine, and liquid media + 2 mM dopamine + 50 μM NS163. There were six conditions for this experiment: (1) N2 worms, (2) N2 worms treated with 2 mM dopamine, (3) UA196 worms, (4) UA196 worms treated with 2 mM dopamine, (5) UA196 worms treated with 50 μM NS163, and (6) UA196 worms treated with 2 mM dopamine and 50 μM NS163. For each condition, four biological replicates were performed, and each biological replicate consisted of four technical replicates (four wells). For each condition, the data were expressed as mean and the error bars report the s.e.m. (n = 4 independent experiments and each n consisted of a total of four technical replicates).

Confocal microscopy imaging of early stage treated *C. elegans* (NL5901 and UA196) with ligands

This experiment was performed based on previously described protocols with slight modifications^{4,11,13}. Briefly, the worm strains (NL5901 or UA196) were bleached at the same time and the eggs were incubated on **PLATE 1**. On day two, the worms were divided into two batches. One half of the worms were transferred (using M9 buffer) on **PLATE 2 (without ligand)** and the second half of the worms were transferred to **PLATE 3 (with ligand)**. The concentration of the ligands varied from 10-50 μM as described in the main manuscript. The worms were incubated on **PLATE 2 or PLATE 3** up to day 3 at 20 °C in an incubator with constant humidity. On day four, the ligand (10-50 μM , stock conc. = 10 mM in DMSO, DMSO from 0.1-0.5%, v/v) was added again to **PLATE 3**. For confocal imaging, at least 10 worms per condition (from **PLATE 2 or PLATE 3**) were transferred to a cover slide containing an anesthetic (40 mM sodium azide), and mounted on glass microscope slide containing 2% agarose pads using a reported protocol¹³. The images of the worms were collected using an Olympus Fluoview

FV3000 confocal/2-photon microscope (40 x Plan-Apo/1.3 NA objective with DIC capability) and processed using the OlympusViewer in ImageJ software⁴. For NL5901 worms, the inclusions of the aggregated α S (GFP puncta in the muscle cells) were manually counted (10 worms per condition) for the five-day imaging period (day 5 to day 9) for **PLATE 2 or PLATE 3**. For UA196 strain, the number of healthy DA neurons (fluorescence due to GFP in DA neurons) were counted (10 worms per condition) using confocal imaging of the worms on day 3, day 5, day 10, and day 15. For each condition, six biological replicates were performed and at least 10 technical replicates were used for each biological replicate. The data were expressed as mean and the error bars report the s.e.m. (n = 6 independent experiments and each n consisted of a minimum of ten technical replicates).

Chemotaxis Assay for *C. elegans* (N2 and UA196) in the presence of ligands

For chemotaxis assay, the Chemotaxis (CTX) media plates (2% Agar, 5 mM KH₂PO₄, 1 mM CaCl₂, 1 mM MgSO₄, 75 μ M FUDR), **referred to as PLATE 7**, and CTX buffer (5 mM KH₂PO₄, 1 mM CaCl₂, and 1 mM MgSO₄) were prepared using previous protocols^{4,8-10}. The **PLATE 7** (without ligand) were treated with ligand (50 μ M, stock conc. = 10 mM in DMSO) that was dissolved in CTX buffer (total volume = 300 μ L) and was spotted atop the CTX media plates at 20 °C, **referred to as PLATE 8**. For CTX assay, we prepared another plate **referred to as PLATE 9**, which was **PLATE 7** divided into four equal quadrants and designated as A and C (diagonally opposite), B and D (diagonally opposite). A solution of *E coli* (50 μ L of 0.5 OD at 600 nm, as an attractant) was placed at ~0.4 cm from the edge of quadrants B and D and the plate was allowed to dry for 1 h at 20 °C. Subsequently, ethanol (10 μ L, repellent) was added at ~0.4 cm from the edge of quadrants A and C at 20 °C. To ensure that the ethanol did not dry, the worms (N2 or UA196) were transferred to this plate within 10 min and the lid of the plate was closed. The experiment on the WMicroTracker ARENA plate reader (Phylumtech, Santa Fe, Argentina) started within 1 h of the transfer of the worms at 20°C. To carry out the experiment, the worms (N2 or UA196) were bleached at the same time and the eggs were incubated on **PLATE 1**. On day two, the worms were divided into two batches. One half of the worms were transferred (using CTX buffer) on **PLATE 7 (without ligand)** and the second half of the worms were transferred to **PLATE 8 (with 50 μ M ligand)** and kept at 20 °C in an incubator. On day 3, a total of 50 worms were transferred to the center of the **PLATE 9** using CTX buffer (~20 μ L) and the number of worms were counted under an Olympus microscope (SZ-6145, Waltham, MA). Subsequently, the lid of the plate was closed with parafilm (Bemis Company, Inc., Neenah, WI). The worm activity was monitored using the WMicroTracker ARENA plate reader (Phylumtech, Santa Fe, Argentina) at 20°C for 2 h. We used exactly similar conditions for day 10 experiment except the ligand was added again (50 μ M and 300 μ L in CTX buffer, stock conc. = 10 mM in DMSO, DMSO = 0.5%, v/v) to **PLATE 8**. For day 10, 50 worms (N2 or UA196) from **PLATE 7 (without ligand)** or **PLATE 8 (with 50 μ M ligand)** were transferred to **PLATE 9** using CTX buffer (~20 μ L) and the number of worms were counted under an Olympus microscope (SZ-6145, Waltham, MA). Subsequently, the lid of the plate was closed with parafilm (Bemis Company, Inc., Neenah, WI). The worm activity was monitored using the WMicroTracker ARENA plate reader (Phylumtech, Santa Fe, Argentina) at 20°C for 2 h. The report was generated using MapPlot option on the WMicroTracker ARENA plate reader. The experiment was conducted for both N2 and UA196 worms (in the absence and presence of 50 μ M NS163) on day three and day 10 of the aging process. For each condition, three biological replicates were performed and at least two technical replicates were used for each biological replicate. The data were expressed as mean and the error bars report the s.e.m. (n = 3 independent experiments and each n consisted of two technical replicates).

Measurement of the ROS level in UA196 worms in the presence of ligands (Quantification of ROS and confocal imaging)

The worm strains were bleached at the same time and the eggs were incubated on **PLATE 1**. On day two, the worms were divided into two batches. One half of the worms were transferred (using M9 buffer) on **PLATE 2 (without ligand)** and the second half of the worms were transferred to **PLATE 3 (with ligand)**. The concentration

of the ligands was 50 μM as described in the main manuscript. On day four, the ligand (10-50 μM in M9 buffer with 0.1%-0.5% DMSO, v/v) was spotted again atop the **PLATE 3** at 20 $^{\circ}\text{C}$. The worms were incubated on **PLATE 2 or PLATE 3** up to day eight at 20 $^{\circ}\text{C}$ in an incubator with constant humidity. On day eight, the worms were transferred into 1.7 mL microcentrifuge tubes using M9 buffer (1 mL) and washed with M9 buffer (1 mL and three times) by centrifugation for 2 min at 2,500 rpm and 20 $^{\circ}\text{C}$. Subsequently, a total of 100 worms/well were transferred to a Costar 96-well black plate (Corning, Kennebunk, ME) containing 100 μL solution, including M9 buffer (89 μL), worm solution (10 μL and 100 worms), and 2',7'-dichlorofluorescein diacetate (H₂DCFDA) dye (1 μL , 50 μM , stock solution = 5 mM in cell culture grade DMSO, 99% pure). As a control, 99 μL of M9 buffer and 1 μL of 5 mM H₂DCFDA reaction solution was placed in the wells. Subsequently, the Costar 96-well plate was gently shaken for 30 sec at 20 $^{\circ}\text{C}$ and the fluorescence intensity was measured (λ_{ex} = 485 nm and λ_{em} = 530 nm) at multiple time points (0 to 120 min) using the Infinite M200 Pro Plate Reader (Tecan, Männedorf, Switzerland)^{11,14,15}.

For confocal imaging, the same UA196 worms (in the absence and presence of ligands) used for the quantification of ROS, were transferred from the 96-well plate into **PLATE 2**, air-dried into the sterile laminar flow hood. The worms (~10 worms) were transferred to a cover slide containing an anesthetic (40 mM sodium azide), and mounted on a glass microscope slide containing 2% agarose pads for the confocal imaging as described in the previous sections. This experiment consisted of three biological replicates and each biological replicate consisted of three technical replicates. The data were expressed as mean and the error bars report the sd's (n = 3 independent experiments and each n consisted of three technical replicates).

Confocal imaging of a post-disease onset PD model of UA196 worms in the absence and presence of ligands

The UA196 worms were bleached, and the eggs were incubated on **PLATE 1**. On day two, the UA196 worms were transferred (using M9 buffer) on **PLATE 2 (without ligand)**. On day five, 10 UA196 worms were used to determine the number of healthy DA worms using confocal imaging (as described earlier) prior to the treatment with ligands on day five. In tandem, ~50 worms were transferred to **PLATE 3 (with ligand)** using M9 buffer (10-20 μL) and the plate was incubated at 20 $^{\circ}\text{C}$ in the incubator. On days 10 and 15, 10 UA196 worms from **PLATE 2 (without ligand)** and **PLATE 3 (with ligand)** were used to determine the number of healthy DA worms using confocal imaging (as described earlier). For each condition, six biological replicates were performed and at least 10 technical replicates were used for each biological replicate. The data were expressed as mean and the error bars report the s.e.m. (n = 6 independent experiments and each n consisted of a minimum of ten technical replicates).

Measurement of intracellular ROS level in a post-disease onset PD model of UA196 worms in the absence and presence of ligands

The UA196 worms were bleached, and the eggs were incubated on **PLATE 1**. On day two, the UA196 worms were transferred (using M9 buffer) on **PLATE 2 (without ligand)**. On day five, 100 UA196 worms were used to determine the ROS level (as described earlier) prior to the treatment with ligands on day five. In tandem, ~200 worms were transferred to **PLATE 3 (with ligand)** using M9 buffer (10-20 μL) and the plate was incubated at 20 $^{\circ}\text{C}$ in the incubator. On day 8, 100 UA196 worms from **PLATE 2 (without ligand)** and **PLATE 3 (with ligand)** were used to determine the ROS level (as described earlier). For each condition, three biological replicates were performed, and 3 technical replicates were used for each biological replicate. The data were expressed as mean and the error bars report the s.e.m. (n = 3 independent experiments and each n consisted of 3 technical replicates).

We agree with the reviewer that we did not compare the antagonist activity of OPs with the other compounds reported in the literature. As suggested by the reviewer, in the revised manuscript, we compared the antagonist

activity of OPs with multiple ligands, which were reported in literature (and commercially available) using both *in vitro* (ThT aggregation assay) and *in vivo* (UA196 worms) assays. Both OPs, NS132 and NS163 were a lot better antagonists of α S aggregation at an equimolar ratio under our conditions for the ThT aggregation assay. Also, we compared the antagonist activity of these ligands in rescuing the degeneration of DA neurons in UA196 worms at 50 μ M, similar concentration to NS132 and NS163. Under these conditions, NS163 (then NS132) was the most potent ligand in rescuing the degeneration of DA neurons in UA196 worms. Both assays demonstrate that our OPs (NS163 and NS132) were far better antagonists of α S aggregation than the reported ligands in literature, both *in vitro* and *in vivo* assays. The detailed experimental procedure has been added to the materials and methods and the results have been included in the main manuscript (Fig. 4f, 5l, 6l) and the supplementary information (Supplementary Fig. 27, 36). The explanation has been added in the main text and it reads as below:

Fig. 5l and Supplementary Fig. 27. Neuroprotective effect of various ligands on the degeneration of DA neurons. Statistics for the total number of neurons in UA196 worms on the indicated days and the effect of the indicated ligands (50 μ M, treatment on day two and four) on the DA neurons on day 15. The number of DA neurons were determined using the confocal imaging. For each confocal imaging experiment, 10 worms were used, and the healthy neurons were counted manually, and each condition (day) consisted of six independent experiments with freshly bleached worms. The data were expressed as mean and the error bars report the s.e.m. (n = 6 independent experiments and each experiment consisted of 10 technical replicates). The statistical analysis was performed using ANOVA with Tukey's multiple comparison test. *p < 0.05, **p < 0.01, ***p < 0.001, ****p < 0.0001.

“We compared NS163 and NS132 with other reported ligand inhibitors of α S aggregation in the literature for their ability to rescue the degeneration of DA neurons in UA196 worms under matched conditions (50 μ M concentration). The number of intact DA neurons in UA196 worms on day 15 in the presence of Bexarotene, Tyrosol, Valporic acid, and EGCG was 19.3, 21.7, 18.7, and 30.5, respectively (Fig. 5l, Supplementary Fig. 27). Most of the ligands were ineffective in rescuing the degeneration of DA neurons in UA196 worms (Fig. 5l, Supplementary Fig. 27). EGCG was moderate in rescuing the degeneration of DA neurons (Fig. 5l,

Supplementary Fig. 27). Our results corroborate well with the published data as some of these molecules demonstrate noticeable rescue effect at one mM concentration⁶⁴⁻⁶⁶. Under matched conditions, in the presence of NS132 (50 μ M), the total number of intact DA neurons on day 15 was 51, which confirms that NS132 also effectively rescues the degeneration of DA neurons in UA196 worms (Supplementary Fig. 27, 28a-e). The neuroprotective effect of NS163 was better than NS132, indicated by a higher number of intact healthy neurons (Fig. 5k, Supplementary Fig. 28a-e), most likely due to the former's better ability to permeate the cell membrane.”

Fig. 6l and Supplementary Fig. 36. Neuroprotective effect of various ligands on the degeneration of DA neurons in a post-disease PD model of UA196 worms. Statistics for the total number of intact DA neurons in UA196 worms in the absence and presence of the indicated ligands (at 50 μ M, day 5) on day 15. For each confocal imaging experiment at least 10 worms were used, and the healthy neurons were counted manually, and each condition (day) consisted of six independent experiments. The data were expressed as mean and the error bars report the s.e.m. (n = 6 independent experiments and each n consisted of 10 technical replicates). The statistical analysis was performed using ANOVA with Tukey's multiple comparison test. *p < 0.05, **p < 0.01, ***p < 0.001, ****p < 0.0001.

“We also compared the antagonist activity of NS163 and NS132 with other reported inhibitors of α S aggregation for their ability to rescue the degeneration of DA neurons in UA196 worms in the late-stage onset PD model. The ligands (50 μ M) were added on day five and the number of intact DA neurons was counted on day 15. The number of intact DA neurons decreased from 58.7 (day 3) to 12.8 (day 15). At this dose, the number of intact DA neurons in UA196 worms on day 15 in the presence of NS163, Bexarotene, Tyrosol, Valporic acid, and EGCG was 41.3, 13.2, 14.3, 13.5, and 20.5, respectively (Fig. 6l). All other ligands except EGCG were not able to rescue the degeneration of DA neurons in UA196 worms (Fig. 6l). EGCG was moderate in rescuing the degeneration of DA neurons. NS132 was also effective (less than NS163) in rescuing degeneration of DA neurons in a post-disease onset PD model (Supplementary Fig. 35a-f, 36). The number of intact DA neurons in UA196 worms on day 15

in the presence of NS132 was 32, which indicates that it was a far better ligand than other reported ligands from literature in rescuing the degeneration of DA neurons (Supplementary Fig. 35a-f, 36). Overall, NS163 was the most potent ligand in rescuing degeneration of DA neurons in the post-disease onset PD model.”

In conclusion, our molecules (NS132 and NS163) were far better antagonists in rescuing α S aggregation (and PD phenotypes) in both in vitro and in vivo PD models.

Reviewer #3 (Remarks to the Author):

1. Essential information is missing on how the *C. elegans* strains were prepared for treatment. It is also not clear how the NS163 and NS132 antagonists were administered to the worms? Were worms maintained on agar plates or liquid cultures? Was NS163 and NS132 administered only at day 2 of adulthood and then the agonist was removed, or the agonist was provided continuously after day 2 until day 14? Did the authors conduct an egg preparation to synchronize the worm populations when they conducted the experiments? Along the same lines, how exactly and for how long NS163 was administered in the post-disease onset PD model?

*We apologize for not providing the details of the preparation of the *C. elegans* strains and their treatment with ligands. The worms were treated with ligands on day two and day four (two doses) for all the experiments. The ligands (10 mM stock in cell culture grade DMSO, 99% pure) were dissolved in M9 buffer (DMSO, 0.1-0.5% v,v; for different doses of ligands from 10-50 μ M) and then transferred to the agar NGM plates or added to the liquid media in plates. The worms were transferred and maintained on the agar plates for ROS level determination, chemotaxis assays, and confocal imaging. The worms were on the same NGM plates for the whole duration of the experiment. For the motility assays, the worms were transferred to the liquid media to measure the motility rate of various strains of worms using the WMicroTracker ARENA worm plate reader and the worms were in the same liquid media (with ligands) for the whole duration of the experiment. We used different conditions for experiments because of the suitability of the WMicroTracker ARENA worm plate reader for the liquid media. The ligands (NS163 and NS132) were administered on day two and day four to the worms on the plates and did not remove the worms from the plates for the whole course of the experiments. We did conduct an egg preparation by bleaching process and then synchronized the worms and treated them with FUDR to avoid any offsprings. For the post-disease onset model, the ligands (10 mM stock in cell culture grade DMSO, 99% pure) were dissolved in M9 buffer (DMSO, 0.1-0.5% v,v; for different doses of ligands from 10-50 μ M) and then transferred to the agar NGM plates. Subsequently, the worms were transferred to the NGM plates, and the worms were on the same NGM plates for the whole duration of the experiment. We have revised the detailed experimental procedure for *C. elegans* experiments, and it is included in the supplementary information, which reads as below:*

C. elegans Experiments

Media and buffers for various worm assays.

The worms were maintained at standard conditions (20 °C at all times) on nematode growth media (NGM) agar in 60 mm plates (CytoOne, Ocala, FL) using *E. coli* OP50 strain as the food source following the previous protocols^{4,7,8}. To ensure that worm's colonies don't starve, they were always transferred on new plates with *E.*

coli OP50 strain as the food source. All worm strains were replaced with new worm colony after every six months to avoid any genetic mutation, which might affect the disease phenotypes. NGM agar plates, M9 buffer (3 g KH_2PO_4 , 6 g Na_2HPO_4 , 5 g NaCl, 1 mL 1 M MgSO_4 , milli-Q H_2O to 1L), Chemotaxis (CTX) media plates (2% Agar, 5 mM KH_2PO_4 , 1 mM CaCl_2 , and 1 mM MgSO_4), and CTX buffer (5 mM KH_2PO_4 1 mM CaCl_2 , and 1 mM MgSO_4) were prepared using previous protocols^{4,8-10}

Strains

The N2 (wild-type *C. elegans* Bristol strain), NL5901 (*C. elegans* model of PD), and *Escherichia coli* OP50 (*E. coli*, a uracil requiring mutant) strains were obtained from Caenorhabditis Genomics Center (CGC, Minneapolis, MN). The UA196 strain was generously donated by the laboratory of Dr. Guy Caldwell (Department of Biological Science, The University of Alabama, Tuscaloosa, AL, United States)⁶.

NL5901 strain (pkIs2386 [unc-54p:: α synuclein::YFP+unc-119(+)]). In NL5901 strain, α S-YFP is expressed in the muscle cells of worms. The aggregates of α S-YFP are visible from day 4 and the maximum number of α S-YFP aggregates are visible round day 8 and day 9.

UA196 strain [P_{dat-1}:: α -syn+P_{dat-1}::GFP]. In UA196 strain, human α S and GFP are expressed in DA neurons under control of a dopamine transporter-specific promoter [P_{dat-1}:: α -syn+P_{dat-1}::GFP], which results in age-dependent neurodegeneration of six DA neurons.

Culture methods for *C. elegans* strains

The standard worm conditions were used for the culturing of various worm strains. Briefly, the worms were bleached and synchronized using hypochlorite solution, followed by the incubation of the eggs (at 20 °C) on NGM plates (35 mm, CellTreat Scientific, Pepperell, MA), which were seeded with OP50 (350 μL , 0.5 OD_{600nm}) as a food source, **referred to as PLATE 1**. The cultures of OP50 were prepared by incubating 50 mL of LB medium with OP50 18 h at 37 °C and the final OD value was adjusted to 0.5 at 600 nm. The NGM plates were prepared by treating them with 350 μL OP50 and leaving the plates at 20°C for 3 days. On day two, the worms were transferred (using M9 buffer) to NGM plates (35 mm, CellTreat Scientific, Pepperell, MA) containing 75 μM Fluorodeoxyuridine (FUDR; to prevent worm reproduction and ensure that equal ages of worms were used for the experiment)^{4,8} and OP50 as food source, **referred to as PLATE 2**. These NGM plates were prepared using autoclaved NGM media supplemented with 75 μM FUDR (2 mL total liquid). After 12 h, the NGM plates were seeded with 350 μL of OP50 (with OD = 0.5 at 600 nm) at 20 °C. For treating various worm strains with ligands, the worms were transferred on day 2 to the NGM plates treated with ligands (please see below to prepare the NGM plates treated with ligands).

Preparation of NGM plates with Ligands

The NGM plates (with NGM media) were used for the treatment of different worm strains with ligands. The NGM media containing 75 μM FUDR was autoclaved and poured in NGM plates (2 mL total liquid). After 12 h, the NGM plates were seeded with 350 μL of OP50 (with OD = 0.5 at 600 nm) at 20 °C. After 12 h, different doses of ligands (10-50 μM , stock conc. = 10 mM in DMSO, DMSO from 0.1-0.5%, v/v) were dissolved in M9 buffer (total volume = 300 μL) and were spotted atop the NGM plates at 20 °C, **referred to as PLATE 3**. The NGM plates treated with the ligands were placed in sterile laminar flow hood at 20 °C for 1 h. The NGM plates treated with ligands were prepared and used within 24 h. We used these plates to treat various disease strains for different assays.

Motility assay for *C. elegans* (N2, NL5901 and UA196)

Briefly, all worm strains were bleached at the same time and the eggs were incubated on **PLATE 1**. On day two, the worms were divided into two batches. One half of the worms were transferred (using M9 buffer) on **PLATE 2 (without ligand)** and the second half of the worms were transferred to **PLATE 3 (with ligand)**. The concentration of the ligands varied from 10-50 μM as described in the main manuscript. The worms were incubated on **PLATE 2 or PLATE 3** up to day 4 at 20 $^{\circ}\text{C}$ in an incubator with constant humidity. On day four, various worm strains (with and without ligands from PLATE 2 or PLATE 3) were transferred to sterile 24 well plate (CellTreat Scientific, Pepperell, MA) containing liquid media (500 μL /well), **referred to as PLATE 4A** and containing liquid media with ligands (500 μL /well with 10-50 μM ligand in M9 buffer with 0.1%-0.5% DMSO, v/v), **referred to as PLATE 4B**. The liquid media for **PLATE 4A/B** was prepared with 67.28% (v/v) of M9 buffer, 75 μM FUDR, 0.1% of 1 M magnesium sulfate (v/v), 0.1% of 1 M calcium chloride (v/v), 2.5% of 1 M potassium phosphate solution (pH 6, v/v), and 30% (v/v) of OP50 (0.5 OD_{600nm}). A total of 50 worms per well were manually transferred into **PLATE 4A/B** at 20 $^{\circ}\text{C}$ and a total of four wells (4 technical replicates) were used for each condition. Subsequently, the worms were incubated for 6 h at 20 $^{\circ}\text{C}$ with constant shaking (rpm = 100) to get acclimated with the solution conditions before starting the motility assay experiment.

To test the effect of each ligand on disease worm strains (NL5901 or UA196), there were four conditions were used in **PLATE 4A/B**: (1) N2 worms, (2) N2 worms treated with ligand, (3) NL5901 (or UA196) worms, and (4) NL5901 (or UA196) worms treated with the ligand (10-50 μM). The assay was started on day four using the WMicroTracker ARENA plate reader (Phylumtech, Santa Fe, Argentina) at 20 $^{\circ}\text{C}$ for 1 h per day over a 14-day period at intervals of 24 h. Each day, a total of 20 activity scores per well were collected in 1 h. After collecting the data each day, the **PLATE 4A/B** was again placed on the shaker (100 rpm) at 20 $^{\circ}\text{C}$ in the incubator. The **PLATE 4A/B** was on the on the shaker (100 rpm) at 20 $^{\circ}\text{C}$ in the incubator the whole duration of the experiment except only when the reading was collected on the WMicroTracker ARENA plate reader for 1 h. For each condition, four biological replicates were performed, and each biological replicate consisted of four technical replicates (four wells). For each condition, the data were expressed as mean and the error bars report the s.e.m. (n = 4 independent experiments and each n consisted of a total of four technical replicates).

Motility assay for *C. elegans* (N2 and UA196) in the presence of dopamine and NS163

This motility assay was performed exactly similar to the previous assay with slight adjustment in the preparation of the plates for the treatment of UA196 worms. For this assay, the only difference is the 24 well plates were prepared with 2 mM dopamine (**referred to as PLATE 5**) and both 2 mM dopamine and 50 μM NS163 (**referred to as PLATE 6**). To prepare **PLATE 5**, the NGM media containing 75 μM FUDR was autoclaved and poured in NGM plates (2 mL total liquid). After 12 h, the NGM plates were seeded with 350 μL of OP50 (with OD = 0.5 at 600 nm) at 20 $^{\circ}\text{C}$. After 12 h, 2 mM dopamine dissolved in M9 buffer (total volume = 300 μL) was spotted atop the NGM plates at 20 $^{\circ}\text{C}$. To prepare **PLATE 6**, the NGM media containing 75 μM FUDR was autoclaved and poured in NGM plates (2 mL total liquid). After 12 h, the NGM plates were seeded with 350 μL of OP50 (with OD = 0.5 at 600 nm) at 20 $^{\circ}\text{C}$. After 12 h, NS163 (50 μM , stock conc. = 10 mM in DMSO) and dopamine (2 mM in M9 buffer) were dissolved in M9 buffer (total volume = 300 μL) and were spotted atop the NGM plates at 20 $^{\circ}\text{C}$. The NGM plates treated with dopamine and NS163 and dopamine were placed in sterile laminar flow hood at 20 $^{\circ}\text{C}$ for 1 h. The NGM plates treated with dopamine and NS163 and dopamine were prepared and used within 24 h.

Similar to the previous section, the motility assay was performed for N2 and UA196 worms in the absence and presence of NS163+dopamine in sterile 24 well plate (CellTreat Scientific, Pepperell, MA) containing liquid media (500 μL /well), liquid media + 2 mM dopamine, and liquid media + 2 mM dopamine + 50 μM NS163. There were six conditions for this experiment: (1) N2 worms, (2) N2 worms treated with 2 mM dopamine, (3) UA196 worms, (4) UA196 worms treated with 2 mM dopamine, (5) UA196 worms treated with 50 μM NS163, and (6) UA196 worms treated with 2 mM dopamine and 50 μM NS163. For each condition, four biological

replicates were performed, and each biological replicate consisted of four technical replicates (four wells). For each condition, the data were expressed as mean and the error bars report the s.e.m. (n = 4 independent experiments and each n consisted of a total of four technical replicates).

Confocal microscopy imaging of early stage treated *C. elegans* (NL5901 and UA196) with ligands

This experiment was performed based on previously described protocols with slight modifications^{4,11,13}. Briefly, the worm strains (NL5901 or UA196) were bleached at the same time and the eggs were incubated on **PLATE 1**. On day two, the worms were divided into two batches. One half of the worms were transferred (using M9 buffer) on **PLATE 2 (without ligand)** and the second half of the worms were transferred to **PLATE 3 (with ligand)**. The concentration of the ligands varied from 10-50 μM as described in the main manuscript. The worms were incubated on **PLATE 2 or PLATE 3** up to day 3 at 20 °C in an incubator with constant humidity. On day four, the ligand (10-50 μM , stock conc. = 10 mM in DMSO, DMSO from 0.1-0.5%, v/v) was added again to **PLATE 3**. For confocal imaging, at least 10 worms per condition (from **PLATE 2 or PLATE 3**) were transferred to a cover slide containing an anesthetic (40 mM sodium azide), and mounted on glass microscope slide containing 2% agarose pads using a reported protocol¹³. The images of the worms were collected using an Olympus Fluoview FV3000 confocal/2-photon microscope (40 x Plan-Apo/1.3 NA objective with DIC capability) and processed using the OlympusViewer in ImageJ software⁴. For NL5901 worms, the inclusions of the aggregated αS (GFP puncta in the muscle cells) were manually counted (10 worms per condition) for the five-day imaging period (day 5 to day 9) for **PLATE 2 or PLATE 3**. For UA196 strain, the number of healthy DA neurons (fluorescence due to GFP in DA neurons) were counted (10 worms per condition) using confocal imaging of the worms on day 3, day 5, day 10, and day 15. For each condition, six biological replicates were performed and at least 10 technical replicates were used for each biological replicate. The data were expressed as mean and the error bars report the s.e.m. (n = 6 independent experiments and each n consisted of a minimum of ten technical replicates).

Chemotaxis Assay for *C. elegans* (N2 and UA196) in the presence of ligands

For chemotaxis assay, the Chemotaxis (CTX) media plates (2% Agar, 5 mM KH_2PO_4 , 1 mM CaCl_2 , 1 mM MgSO_4 , 75 μM FUDR), **referred to as PLATE 7**, and CTX buffer (5 mM KH_2PO_4 , 1 mM CaCl_2 , and 1 mM MgSO_4) were prepared using previous protocols^{4,8-10}. The **PLATE 7** (without ligand) were treated with ligand (50 μM , stock conc. = 10 mM in DMSO) that was dissolved in CTX buffer (total volume = 300 μL) and was spotted atop the CTX media plates at 20 °C, **referred to as PLATE 8**. For CTX assay, we prepared another plate **referred to as PLATE 9**, which was **PLATE 7** divided into four equal quadrants and designated as A and C (diagonally opposite), B and D (diagonally opposite). A solution of *E coli* (50 μL of 0.5 OD at 600 nm, as an attractant) was placed at ~0.4 cm from the edge of quadrants B and D and the plate was allowed to dry for 1 h at 20 °C. Subsequently, ethanol (10 μL , repellent) was added at ~0.4 cm from the edge of quadrants A and C at 20 °C. To ensure that the ethanol did not dry, the worms (N2 or UA196) were transferred to this plate within 10 min and the lid of the plate was closed. The experiment on the WMicroTracker ARENA plate reader (Phylumtech, Santa Fe, Argentina) started within 1 h of the transfer of the worms at 20°C. To carry out the experiment, the worms (N2 or UA196) were bleached at the same time and the eggs were incubated on **PLATE 1**. On day two, the worms were divided into two batches. One half of the worms were transferred (using CTX buffer) on **PLATE 7 (without ligand)** and the second half of the worms were transferred to **PLATE 8 (with 50 μM ligand)** and kept at 20 °C in an incubator. On day 3, a total of 50 worms were transferred to the center of the **PLATE 9** using CTX buffer (~20 μL) and the number of worms were counted under an Olympus microscope (SZ-6145, Waltham, MA). Subsequently, the lid of the plate was closed with parafilm (Bemis Company, Inc., Neenah, WI). The worm activity was monitored using the WMicroTracker ARENA plate reader (Phylumtech, Santa Fe, Argentina) at

20°C for 2 h. We used exactly similar conditions for day 10 experiment except the ligand was added again (50 μ M and 300 μ L in CTX buffer, stock conc. = 10 mM in DMSO, DMSO = 0.5%, v/v) to **PLATE 8**. For day 10, 50 worms (N2 or UA196) from **PLATE 7 (without ligand)** or **PLATE 8 (with 50 μ M ligand)** were transferred to **PLATE 9** using CTX buffer (~20 μ L) and the number of worms were counted under an Olympus microscope (SZ-6145, Waltham, MA). Subsequently, the lid of the plate was closed with parafilm (Bemis Company, Inc., Neenah, WI). The worm activity was monitored using the WMicroTracker ARENA plate reader (Phylumtech, Santa Fe, Argentina) at 20°C for 2 h. The report was generated using MapPlot option on the WMicroTracker ARENA plate reader. The experiment was conducted for both N2 and UA196 worms (in the absence and presence of 50 μ M NS163) on day three and day 10 of the aging process. For each condition, three biological replicates were performed and at least two technical replicates were used for each biological replicate. The data were expressed as mean and the error bars report the s.e.m. (n = 3 independent experiments and each n consisted of two technical replicates).

Measurement of the ROS level in UA196 worms in the presence of ligands (Quantification of ROS and confocal imaging)

The worm strains were bleached at the same time and the eggs were incubated on **PLATE 1**. On day two, the worms were divided into two batches. One half of the worms were transferred (using M9 buffer) on **PLATE 2 (without ligand)** and the second half of the worms were transferred to **PLATE 3 (with ligand)**. The concentration of the ligands was 50 μ M as described in the main manuscript. On day four, the ligand (10-50 μ M in M9 buffer with 0.1%-0.5% DMSO, v/v) was spotted again atop the **PLATE 3** at 20 °C. The worms were incubated on **PLATE 2 or PLATE 3** up to day eight at 20 °C in an incubator with constant humidity. On day eight, the worms were transferred into 1.7 mL microcentrifuge tubes using M9 buffer (1 mL) and washed with M9 buffer (1 mL and three times) by centrifugation for 2 min at 2,500 rpm and 20 °C. Subsequently, a total of 100 worms/well were transferred to a Costar 96-well black plate (Corning, Kennebunk, ME) containing 100 μ L solution, including M9 buffer (89 μ L), worm solution (10 μ L and 100 worms), and 2',7'-dichlorofluorescein diacetate (H₂DCFDA) dye (1 μ L, 50 μ M, stock solution = 5 mM in cell culture grade DMSO, 99% pure). As a control, 99 μ L of M9 buffer and 1 μ L of 5 mM H₂DCFDA reaction solution was placed in the wells. Subsequently, the Costar 96-well plate was gently shaken for 30 sec at 20 °C and the fluorescence intensity was measured (λ_{ex} = 485 nm and λ_{em} = 530 nm) at multiple time points (0 to 120 min) using the Infinite M200 Pro Plate Reader (Tecan, Männedorf, Switzerland)^{11,14,15}.

For confocal imaging, the same UA196 worms (in the absence and presence of ligands) used for the quantification of ROS, were transferred from the 96-well plate into **PLATE 2**, air-dried into the sterile laminar flow hood. The worms (~10 worms) were transferred to a cover slide containing an anesthetic (40 mM sodium azide), and mounted on a glass microscope slide containing 2% agarose pads for the confocal imaging as described in the previous sections. This experiment consisted of three biological replicates and each biological replicate consisted of three technical replicates. The data were expressed as mean and the error bars report the sd's (n = 3 independent experiments and each n consisted of three technical replicates).

Confocal imaging of a post-disease onset PD model of UA196 worms in the absence and presence of ligands

The UA196 worms were bleached, and the eggs were incubated on **PLATE 1**. On day two, the UA196 worms were transferred (using M9 buffer) on **PLATE 2 (without ligand)**. On day five, 10 UA196 worms were used to determine the number of healthy DA worms using confocal imaging (as described earlier) prior to the treatment with ligands on day five. In tandem, ~50 worms were transferred to **PLATE 3 (with ligand)** using M9 buffer (10-20 μ L) and the plate was incubated at 20 °C in the incubator. On days 10 and 15, 10 UA196 worms from **PLATE 2 (without ligand)** and **PLATE 3 (with ligand)** were used to determine the number of healthy DA worms using confocal imaging (as described earlier). For each condition, six biological replicates were performed

and at least 10 technical replicates were used for each biological replicate. The data were expressed as mean and the error bars report the s.e.m. (n = 6 independent experiments and each n consisted of a minimum of ten technical replicates).

Measurement of intracellular ROS level in a post-disease onset PD model of UA196 worms in the absence and presence of ligands

The UA196 worms were bleached, and the eggs were incubated on **PLATE 1**. On day two, the UA196 worms were transferred (using M9 buffer) on **PLATE 2 (without ligand)**. On day five, 100 UA196 worms were used to determine the ROS level (as described earlier) prior to the treatment with ligands on day five. In tandem, ~200 worms were transferred to **PLATE 3 (with ligand)** using M9 buffer (10-20 μ L) and the plate was incubated at 20 $^{\circ}$ C in the incubator. On day 8, 100 UA196 worms from **PLATE 2 (without ligand)** and **PLATE 3 (with ligand)** were used to determine the ROS level (as described earlier). For each condition, three biological replicates were performed, and 3 technical replicates were used for each biological replicate. The data were expressed as mean and the error bars report the s.e.m. (n = 3 independent experiments and each n consisted of 3 technical replicates).

2. I do recognize the thorough analysis of the effect of NS163 on NL5901 worms (fig. 4) and UA196 worms (fig. 5-6). However, in all these experiments a very critical control is missing. They authors need to test the effect of NS163 on N2 worms and evaluate motility rate, lifespan, and chemotaxis. This experiment is critical as it will strengthen the authors' conclusions and further support the therapeutic potential of the NS163 antagonist.

We agree with the reviewer that we did not include a critical experiment, which tested the effect of NS163 on the control worms (N2 worms). We have now included these experiments, which tested the effect of NS163 on the motility, lifespan, and the chemotaxis behavior of the N2 worms. We did not observe any significant effect of NS163 on the motility, lifespan, and the chemotaxis behavior of the N2 worms. We have included the data of the motility, lifespan, and the chemotaxis behavior of the N2 worms in the absence and presence of NS163 (Supplementary Fig. 24, 31) and a writeup in the main manuscript, which read as:

Supplementary Fig. 24. The effect of NS163 on the motility of N2 worms. The comparison of the motility rate of N2 in the absence and presence of 50 μ M NS163 (treatment on day two and four). For motility rate experiment, a total of 50 worms were used in duplicate for each experiment and each condition consisted of four independent experiments. The data were expressed as mean and the error bars report the s.e.m. ($n = 4$ independent experiments and each n consisted of two technical replicates). The statistical analysis was performed using ANOVA with Tukey’s multiple comparison test. * $p < 0.05$, ** $p < 0.01$, *** $p < 0.001$.

“As a control, we also checked the effect of NS163 on the motility of the N2 worms. Under matched conditions, NS163 (50 μ M) did not demonstrate any significant toxic effect on the motility of N2 worms (Supplementary Fig. 24). Clearly, NS163 did not display any inherent toxicity toward the N2 worms.”

Supplementary Fig. 31. The CI graph for N2 worms in the absence (green bar) and presence (red bar) of 50 μ M NS163 (treatment on day two and four) under the indicated conditions on day 10. For chemotaxis assays, a total of 50 worms were used in duplicate for each experiment and each condition consisted of three independent experiments with freshly bleached worms. The data were expressed as mean and the error bars report the s.e.m. ($n = 3$ independent experiments and each n consisted of two technical replicates).

“As a control, we also checked the effect of NS163 on the behavior of the N2 worms. Under matched conditions to the UA196 worms, we treated the N2 worms with NS163 (50 μ M) and used ARENA plate reader to measure the CI over time with values from -1.0 to +1.0 (Supplementary Fig. 31). The N2 worms did not display any behavioral deficit up to day 10 as the kinetic data for the CI suggest that all the worms spent most of their time in the *E. coli* quadrants (attractant) reflected by a value of ~ 1 and a value of ~ -1 for ethanol (repellent) during the course of 2h (Supplementary Fig. 31, light and dark green lines). Similarly, we did not observe any behavioral deficits in N2 worms after treatment with NS163 (50 μ M) for 10 days as their CI was in very close proximity to that of the N2 worms (Supplementary Fig. 31, light and dark red lines). Both motility and chemotaxis assays suggest that NS163 did not have any inherent toxicity toward the N2 worms.”

3. The authors have conducted a lot of experiments in Fig. 5 and 6. However, data presentation needs to be improved. For example, the Results state “% of loss of intact DA neurons”. However, there is not a single graph in Fig 5 showing the data as percentage (%). In Fig 5k-n, the authors show at the top of the y-axis as “60 neurons”. Not clear where this number is coming from. Hence, I am confused about the extent of “rescue” of DA degeneration that NS163 provides. In general, the labeling of the y axis (e.g., measurement units) can be improved in most graphs of this paper.

We apologize for not presenting the data in a satisfactory manner. We have now clearly indicated the origin of the values (e.g. 60 DA neurons) in the main manuscript (Fig. 5j,k,l; Fig. 6k,l; Supplementary Fig. 27, 28, 34,

35, 36). The maximum number of healthy DA neurons in one UA196 worm is six on day 3 (no degeneration) and we used 10 worms/biological replicate, which makes a total of 60 healthy neurons/biological replicate. Therefore, we used a total of 60 DA neurons as an average as the maximum value for each biological replicate for various experiments in the manuscript. We have clarified that in the main manuscript and changed the Y-axis values accordingly for the rest of the figures. We have revised figures in the manuscript (Fig. 5j,k,l; Fig. 6k,l) and in the supplementary information (Supplementary Fig. 27, 28, 34, 35, 36). We believe that the changes in the graphs in various Figures are much improved and easy to follow. We have included a writeup in the main manuscript, which read as follows:

“The maximum number of healthy DA neurons in one UA196 worm is six on day 3 (no degeneration) and we used 10 worms/biological replicate, which makes a total of 60 healthy DA neurons/biological replicate. Therefore, for each study, the maximum number of healthy DA neurons in UA196 worms is 60 as an average for each biological replicate.”

4. New confocal microscopy images are needed at higher resolution in Figures 4 and 5. In fig 4, it is unclear why the authors focus on head muscle only; a-Synuclein-YFP aggregation should be observed along the body-wall muscles throughout the A-P axis of the animal.

As suggested by the reviewer, we have incorporated high resolution images in both Figure 4 and 5. We did not have any specific reason to focus on the head muscle only, but we wanted to highlight the aggregation of α S closely, which is the reason to choose the head part. We have now replaced the confocal images in Fig. 4 (e-g), which demonstrate both the full length as well as the head region of the *C. elegans* (Fig. 4i-j). Also, we have included multiple high-resolution images of the NL5901 worms in the absence and presence of NS163 and NS132 (Supplementary Fig. S).

Fig. 4i-k. The representative confocal images of NL5901 worms with α S inclusions (white arrows) in the body wall muscle cells (Day 8) in the absence (i) and presence of 50 μ M NS132 (j) and 50 μ M NS163 (k). For each confocal imaging experiment, at least 10 worms were used, and the inclusions were counted manually, and each condition (Each day) consisted of at least four independent experiments.

NL5901

NL5901+NS132

NL5901+NS163

Supplementary Fig. 20. The representative confocal images of α S-YFP inclusions in the body wall muscle cells of NL5901 (Days = 8) in the absence and presence of 50 μ M NS163 and 50 μ M NS132 (treated on day two and four). These confocal images were collected from at least three independent experiments with random selection of the worms from each independent experiment.

We have also taken new confocal images for Fig. 5 and they have been incorporated in Fig. 5b-i.

We believe that the new confocal images (Fig 4f-g, Fig 5b-i) are high resolution images and look close to the other published images from literature.

5. The flow of data presentation in Figs 5 and 6 can be improved and data need to be re-organized. It was hard to read that part of the Results as the reader needs to go back and forth between data in Fig 5, then in Fig 6, and the back to Fig 5.

We apologize for the inconvenience. We have now modified and re-organized both Fig. 5 and Fig. 6 to improve the data presentation. The results in the modified Fig. 5 and Fig. 6 are now in the correct order and the readers don't have to go back and forth to understand the data. The data and results of Fig. 5 and Fig. 6 are in the sequential flow.

Fig. 5. Neuroprotective effect of OPs on the degeneration of DA neurons. **a**, Schematic of the aging process of UA196 worms and their treatment with the ligands on day 2 (arrow). Representative confocal images of UA196 worms on day 3 (**b,c**), day 5 (**d,e**), and day 15 (**f,g**) and in the presence (**h,i**) of 50 μ M NS163 on day 15. The healthy DA neurons (solid red arrow) and neurites (solid yellow arrow) and degenerated DA neurons (hollow red arrow) and neurites (hollow yellow arrow) on various days. **j**, Statistics of the total number of healthy DA neurons in UA196 worms during the aging process in the absence (**blue line**) and presence of 50 μ M NS163 (**red line**). **k**, Statistics for the total number of UA196 worms with six intact DA neurons during the aging process in the absence and presence of 50 μ M NS163. **l**, Statistics of the total number of healthy DA neurons in UA196 worms during the aging process in the absence (**blue line**) and presence of the indicated ligands (50 μ M). **m**, The comparison of the motility of N2 and UA196 for 13 days and statistics (for day eight, **n**) in the absence and presence of 50 μ M NS163. Representative confocal images of UA196 worms (day eight) treated with CM-H2DCFDA dye to quantify the ROS level (green) in the absence (**o**) and presence (**p**) of 50 μ M NS163 (treatment on day 2). **q**, Statistical analysis of the quantification of the ROS level for experiment **o-p**. For each confocal imaging experiment (**b-i**), at least 10 worms were used, and the healthy neurons (GFP signal) were counted

manually, and each condition (day) consisted of six independent experiments. For motility experiments, a total of 50 worms were used in duplicate for each experiment and each condition consisted of at least four independent experiments. For the ROS level quantification, at least 100 worms were used and each condition consisted of three independent experiments. For ROS assay, the data were expressed as mean and the error bars report the s.d. ($n = 3$ independent experiments and each n consisted of five technical replicates). The data were expressed as mean and the error bars report the s.e.m. ($n = 3$ or 4 independent experiments and each n consisted of a minimum of two technical replicates). The statistical analysis was performed using ANOVA with Tukey's multiple comparison test. * $p < 0.05$, ** $p < 0.01$, *** $p < 0.001$, **** $p < 0.0001$.

Fig. 6. Effect of OPs on the PD phenotypes in UA196 worms. **a**, Schematic to assess the behavioral deficits in UA196 worms in a petri dish in the presence of ethanol (red) and *E. coli* (green) as a function of time. The CI graph for N2, UA196 worms, and UA196 worms treated with 50 μ M NS163 under the indicated conditions on day three (**b**) and day 10 (**c**). Snapshots at 60 min. of the animated videos collected for the CI for UA196 (**d**), UA196+50 μ M NS163 (**e**), and N2 (**f**) under the indicated conditions on day 10. The comparison of the motility for UA196 worms treated with 2 mM dopamine (**g**), UA196 worms treated with 50 μ M NS163+ 2mM dopamine (**h**), and N2 worms treated with 2 mM dopamine (**i**). **j**, Schematic of the aging process of UA196 worms and their treatment with the ligands in a late-onset disease model (day five). **k**, Statistical analysis of the relative number of healthy DA neurons in UA196 worms (blue) during the aging process when treated on day five with 50 μ M NS163 (red). **l**, Statistical analysis of the ROS level in UA196 worms when 50 μ M NS163 was added on day five (arrow). **l**, Statistical analysis of the total number of healthy DA neurons in UA196 worms during the aging

process in the absence (**blue line**) and presence of the indicated ligands (50 μ M) when treated on day 5. For motility assay experiment, a total of 50 worms were used in duplicate for each experiment and each condition consisted of four independent experiments. For each confocal imaging experiment (**k,l**), at least 10 worms were used, and the healthy neurons (GFP signal) were counted manually, and each condition (day) consisted of six independent experiments. For chemotaxis assays, a total of 50 worms were used for each experiment and each condition consisted of three independent experiments. For the ROS level quantification, at least 100 worms were used and each condition consisted of three independent experiments. The data were expressed as mean and the error bars report the s.e.m. (n = 3 or 4 independent experiments and each n consisted of a minimum of two technical replicates). For ROS assay, the data were expressed as mean and the error bars report the s.d. (n = 3 independent experiments and each n consisted of five technical replicates). The statistical analysis was performed using ANOVA with Tukey's multiple comparison test. *p < 0.05, **p < 0.01, ***p < 0.001, ****p < 0.0001.

Minor issues:

1. "Chemical space": can the authors define this term for the non-expert reader?

We have now replaced the term "chemical space" with "array of non-proteinogenic side chains" in the whole manuscript. We believe that the term "array of non-proteinogenic side chains" is suitable for the non-expert readers.

2. In Fig 4h and other panels, the relative motility rate must be defined.

We apologize for not defining the term "the relative motility rate". We meant to use the term "the relative motility" in the manuscript. We have now changed the term to "the relative motility" in the whole manuscript. We have also explained the term "the relative motility" in the main manuscript as it now reads as:

"We utilized the WMicroTracker ARENA plate reader^{24,69} to measure the motility of NL5901 worms in the absence and presence of 50 μ M NS163. The WMicroTracker ARENA plate reader uses a large array of infrared light microbeams in each well of the plate to detect the interference caused by the movement of the worms^{24,69}. The output by the ARENA plate reader is an average of the overall movement of all worms present in each well, which is denoted as the motility of the worms in each well. The relative motility of the worms (under various conditions; see methods for detailed procedure) is calculated by using the first data point of the well of the control worms (N2, healthy *C. elegans* strain, Fig. 4m, green bar) as the highest value of one."

3. Statistical comparisons missing in Fig k, l, m.

We have now calculated the statistical comparisons for Fig. 6k,l,m (now Fig. 6g,h,i) and have incorporated them in the main manuscript.

Fig. 6g-i. The motility of UA196 worms (+2 mM Dopamine) in the absence (**g**) and presence (**h**, +2 mM Dopamine) of 50 μ M NS163 and N2 worms (**i**, +2 mM Dopamine).

4. In multiple places in the text, the authors use “subjective” language (e.g., the effect of NS163 was “better” than NS132; ...a “good” correlation between the inhibition of...It is a “remarkable” finding...; ...with “tremendous” therapeutic potential...). The authors should replace these words with more accurate, less subjective language.

We really appreciate the reviewer for pointing it out. We have now modified the whole manuscript by removing subjective language and replaced it with more accurate language by incorporating comparison values for ligands in various assays. For comparisons between different conditions, we have used numerical values from the data sets. The modified language in the manuscript clearly states and compares the values for different conditions.

5. The authors should remove claims of novelty, such as “for the first time”, as such claims are hard to verify and do not add that much to an otherwise very interesting paper.

We have modified the whole manuscript and have removed the claims of novelty, including the claims “for the first time”. We have removed all the novelty claims from the manuscript and also removed the word “novel” from the whole manuscript.

6. The claim in Discussion that the authors have developed a novel post-disease onset PD model is an overstatement and should be toned down.

We have removed the word “novel” from the whole manuscript. Also, we have removed the claim of novelty for the post-disease onset PD model from the manuscript. We have modified the statement in the discussion and it now reads as follows:

“Moreover, we have applied a post-disease onset PD model that partly resembles and templates the current therapeutic models to identify potent disease-modifying ligands for PD.”

REVIEWER COMMENTS

Reviewer #1 (Remarks to the Author):

Nicholas H. Stillman et al conducted an impressive and innovative study applying the 2D-FAST approach for the design and synthesis of oligopyridylamides (OPs) with different number of pyrimidyl groups and a variety of side chains. This 2D-FAST method has substantial improvements compared with previous work reported by the authors, attributed to the smart use of structural pyrimidylamide scaffolds as starting point, the introduction of different side chains with specific physicochemical properties, and a reduction of the synthetic steps with the concomitant increase in the final product yield compared with their previous reports. The properties of the synthesized OPs were thoroughly evaluated employing biophysical and biological assays for the screening and evaluation of the activity of these OPs against the aggregation of alpha synuclein (aSyn). The article will be of interest to those in the field of amyloid fibril inhibition. The manuscript is well written and organization of the manuscript is clear.

I only have one concern about the final modification to increase the permeability of NS132 to obtain NS163: since the hydroxylamine group has already been reported in the medicinal chemistry field to improve the cell permeability, as the authors cite (Lassalas P., et al, J. Med. Chem. 2016), why the authors did not include the hydroxylamine variant in the initial OPs scaffolds? Alternatively, have the authors evaluated the possibility of developing a “3D-FAST” approach with the permeability as a third component in the future?

I have a few minor comments:

No. 1: In the SDS-PAGE evaluation of fibrillation in the presence and absence of the evaluated compounds, can the authors demonstrate that a 5 min boiling is enough to completely disassemble the fibrils? In all the figures containing SDS-PAGE gels (for example Fig. 2d and Fig Fig. 3g), at the upper part of the gels, it can be seen a fraction of the samples did not enter the gel, especially for the Total aS and Insoluble aS.

No. 2: Although the NMR experiments show that the interaction of both NS132 and NS163

with aS seems to take place at the initial and middle part of the N-terminus domain of WT-aS, this might be different in the case of the mutant A53T-aS. Since the authors have already demonstrated the benefits of NS132 and NS163 in the A53T-aSyn HEK-cells system, a brief comment to discuss the point could be useful.

Reviewer #3 (Remarks to the Author):

The revised manuscript is very much improved. The authors have addressed all my concerns.

Suggestion to the authors regarding manuscript text:

The Results section is unnecessarily long. This reviewer does recognize that a lot of data have been added to this work. However, the data can be presented more succinctly, otherwise the reader will struggle to identify the key conclusion of each experiment. For example, in the *C. elegans* part, extraneous details for each *C. elegans* experiment are included in the Results. Such details are important, but they have to be moved to Materials and Methods.

Abstract, Introduction and Discussion read very well.

Reviewer #1 (Remarks to the Author):

Nicholas H. Stillman et al conducted an impressive and innovative study applying the 2D-FAST approach for the design and synthesis of oligopyridylamides (OPs) with different number of pyrimidyl groups and a variety of side chains. This 2D-FAST method has substantial improvements compared with previous work reported by the authors, attributed to the smart use of structural pyrimidylamide scaffolds as starting point, the introduction of different side chains with specific physicochemical properties, and a reduction of the synthetic steps with the concomitant increase in the final product yield compared with their previous reports. The properties of the synthesized OPs were thoroughly evaluated employing biophysical and biological assays for the screening and evaluation of the activity of these OPs against the aggregation of alpha synuclein (aSyn). The article will be of interest to those in the field of amyloid fibril inhibition. The manuscript is well written and organization of the manuscript is clear.

Response:

We really appreciate comments from the reviewer acknowledging that our work is impressive and innovative, and it is a substantial improvement over the previous synthetic methods. We would also like to thank the reviewer for the comments, which helped us in improving our manuscript.

I only have one concern about the final modification to increase the permeability of NS132 to obtain NS163: since the hydroxylamine group has already been reported in the medicinal chemistry field to improve the cell permeability, as the authors cite (Lassalas P., et al, J. Med. Chem. 2016), why the authors did not include the hydroxylamine variant in the initial OPs scaffolds? Alternatively, have the authors evaluated the possibility of developing a “3D-FAST” approach with the permeability as a third component in the future?

Response:

We agree with the reviewer that we should have used the hydroxylamine variant and other variants with good cell permeability in the OP scaffolds. The reason we did not use the hydroxylamine and other scaffolds is that in our initial study, we used all the variants on the OP scaffolds that can mimic the side chains of the natural amino acids, commercially available, easily available, and cheap. However, the introduction of some of the variants on OPs with better cell permeability, including hydroxylamine (N,2-dihydroxyacetamide for synthesis) was not easily available and was very expensive at commercial sources. The other option was to make them in our lab, which would have further introduced a large body of synthetic work to generate the library. Therefore, in our first installment of 2D-FAST, we introduced various variants on OP scaffolds that were commercially available, easily available, and cheap.

In the future, as the reviewer mentioned, we are working on a second generation of a library of OP scaffolds (e.g. 3D-FAST) where we will further expand the chemical diversity of variants with more improved pharmaceutical properties, including cell permeability. This work is underway and will be presented in the near future.

I have a few minor comments:

No. 1: In the SDS-PAGE evaluation of fibrillation in the presence and absence of the evaluated compounds, can the authors demonstrate that a 5 min boiling is enough to completely disassemble the fibrils? In all the figures containing SDS-PAGE gels (for example Fig. 2d and Fig Fig. 3g), at the upper part of the gels, it can be seen a fraction of the samples did not enter the gel, especially for the Total aS and Insoluble aS.

Response:

We apologize for not being entirely clear on the methodology used for the SDS-PAGE experiment. For SDS-PAGE gel experiments, the protein samples under different conditions were incubated in a 2×Laemmli protein sample loading buffer (containing 2.1% SDS) and then boiled at 95 °C for 5 minutes before running on a 12% Mini-PROTEAN precast protein gel. These details have been added to the methods section of the method section in the Supplemental Information. For the SDS-PAGE experiment, we followed the instructions of the protocol recommended by the supplier (BIO-RAD Laboratories) and by using a published protocol in the literature (Kurien, et al. Methods Mol Biol. 2012, 633-640; Jemil et. al. Nature Commun. 2022, 1-17).

To further support the results of our SDS-PAGE analysis, we performed an SDS-PAGE analysis of the aggregated solution of α -synuclein (70 μ M, 1×PBS buffer, 4 days), where we boiled the samples for 5, 10, and 15 minutes under the SDS-PAGE conditions as described above. We did not observe any noticeable difference between the total amount, the amount of soluble, and insoluble α -synuclein when the solutions were boiled at 5, 10, and 15 minutes (**Figure 1a,b**). Clearly, this experiment shows that the boiling time of the α -synuclein solutions did not make any noticeable difference in the amount of α -synuclein and the outcome of the experiment.

Figure 1. The representative SDS-PAGE gel image (a) and the graphical representation (b) of the SDS-PAGE gel analysis of the solutions of 70 μ M α S (aggregated for four days in 1×PBS buffer) when boiled for 5, 10, and 15 minutes.

No. 2: Although the NMR experiments show that the interaction of both NS132 and NS163 with α S seems to take place at the initial and middle part of the N-terminus domain of WT- α S, this might be different in the case of the mutant A53T- α S. Since the authors have already demonstrated the benefits of NS132 and NS163 in the A53T- α S HEK-cells system, a brief comment to discuss the point could be useful.

Response:

We apologize if we did not make it clear that in our HEK cells based assay, we used the preformed fibers of WT α -synuclein to template the intracellular aggregation of the monomeric A53T mutant of α -synuclein. We used our ligands to inhibit the aggregation of WT α -synuclein fibers templated aggregation of monomeric A53T mutant of α -synuclein. It has been shown that the WT α -synuclein fibers templated aggregation of various mutants of α -synuclein is predominantly dependent on the interaction of the fibers with their N-terminus. Therefore, we envision that the inhibition of WT α -synuclein fibers templated aggregation of monomeric A53T mutant of α -synuclein is likely due to the interaction of our ligands toward the N-terminus of A53T mutant of α -synuclein (similar to WT α -synuclein). We have added a brief description to the main manuscript, which now reads as follows.

“It is important to point out that our potent ligands (NS163 and NS132) were able to inhibit α S preformed fibers templated aggregation of both monomeric WT α S and A53T α S mutant. It has been shown that the WT α S fibers templated aggregation of WT α S and its mutants is predominantly dependent on the interaction of the fibers with the N-terminus of the monomeric WT α S and its mutants^{62,68,69}. Therefore, we surmise that the inhibition of WT α S fibers templated aggregation of intracellular monomeric A53T α S mutant is likely due to the interaction of our ligands (NS163 and NS132) toward the N-terminus of A53T α S mutant (similar to WT α S).”

Reviewer #3 (Remarks to the Author):

The revised manuscript is very much improved. The authors have addressed all my concerns.

Response:

We really appreciate comments from the reviewer acknowledging that the revised manuscript is a much improved version of the original submitted manuscript. We would also like to thank the reviewer for the comments, which helped us in improving our manuscript.

Suggestion to the authors regarding manuscript text:

The Results section is unnecessarily long. This reviewer does recognize that a lot of data have been added to this work. However, the data can be presented more succinctly, otherwise the reader will struggle to identify the key conclusion of each experiment. For example, in the C. elegans part, extraneous details for each C. elegans experiment are included in the Results. Such

details are important, but they have to be moved to Materials and Methods.
Abstract, Introduction and Discussion read very well.

Response:

We agree with the reviewer that the Results section is much longer, and the reason is because of the addition of a lot of data during the revision of the paper. As suggested by the reviewer, we have now shortened the Results section (highlighted in grey color in the main manuscript) by moving parts of it to the Materials and Methods section.

REVIEWERS' COMMENTS

Reviewer #1 (Remarks to the Author):

no more comment. I think the manuscript is now worth acceptance.

Reviewer #1 (Remarks to the Author):

no more comment. I think the manuscript is now worth acceptance.

Response:

We appreciate that the reviewer thinks the manuscript is ready for acceptance.